# ROBUST TRAINING THROUGH ADVERSARIALLY SELECTED DATA SUBSETS

## ABSTRACT

Robustness to adversarial perturbations often comes at the cost of a drop in accuracy on unperturbed or clean instances. Most existing defense mechanisms attempt to defend the learner from attack on all possible instances, which often degrades the accuracy on clean instances significantly. However, in practice, an attacker might only select a small subset of instances to attack, *e.g.*, in facial recognition systems an adversary might aim to target specific faces. Moreover, the subset selection strategy of the attacker is seldom known to the defense mechanism a priori, making it challenging to attune the mechanism beforehand. This motivates designing defense mechanisms which can (i) defend against attacks on subsets instead of all instances to prevent degradation of clean accuracy and, (ii) ensure good overall performance for attacks on any selected subset. In this work, we take a step towards solving this problem. We cast the training problem as a min-max game involving worst-case subset selection along with optimization of model parameters, rendering the problem NP-hard. To tackle this, we first show that, for a given learner's model, the objective can be expressed as a difference between a $\gamma$-weakly submodular and a modular function. We use this property to propose ROGET, an iterative algorithm, which admits approximation guarantees for a class of loss functions. Our experiments show that ROGET obtains better overall accuracy compared to several state-of-the-art defense methods for different adversarial subset selection techniques.

## 1 INTRODUCTION

Recent years have witnessed a dramatic improvement in the predictive power of the machine learning models across several applications such as computer vision, natural language processing, speech processing, *etc.* This has led to their widespread usage in several safety critical systems like autonomous car driving (Janai et al., 2020; Alvarez et al., 2010; Sallab et al., 2017), face recognition (Hu et al., 2015; Kemelmacher-Shlizerman et al., 2016; Wang & Deng, 2021), voice recognition (Myers, 2000; Yuan et al., 2018), *etc.*, which in turn requires the underlying models to be security complaint. However, most existing machine learning models suffer from significant vulnerability in the face of *adversarial attacks* (Szegedy et al., 2014; Carlini & Wagner, 2017; Goodfellow et al., 2015; Baluja & Fischer, 2018; Xiao et al., 2018; Kurakin et al., 2017; Xie & Yuille, 2019; Kannan et al., 2018; Croce & Hein, 2020; Yuan et al., 2019; Tramèr et al., 2018), where instances are contaminated with small and often indiscernible perturbations to delude the model at the test time. This may result in catastrophic consequences when the underlying ML model is deployed in practice.

Driven by this motivation, a flurry of recent works (Madry et al., 2017; Zhang et al., 2019b; 2021b; Athalye et al., 2018; Andriushchenko & Flammarion, 2020; Shafahi et al., 2019; Rice et al., 2020) have focused on designing adversarial training methods, whose goal is to maintain the accuracy of ML models in presence of adversarial attacks. In principle, they are closely connected to robust machine learning methods that seek to minimize the worst-case performance of the ML models with adversarial perturbations. In general, these approaches assume equal likelihood of adversarial attack across each instance. However, in several applications, an adversary might selectively wish to attack a specific subset of instances, which may be unknown to the learner. For example, an adversary can only be interested in perturbing images of specific persons to evade facial recognition systems (Xiao et al., 2021; Vakhshiteh et al., 2021; Zhang et al., 2021b; Sarkar et al., 2021; Venkatesh et al., 2021); in traffic signs classification, the adversary may like to perturb only the stop signs, which can have more adverse impact during deployment. Therefore, the existing adversarial training methods can be

overly pessimistic in terms of their predictive power, since they consider adversarial perturbation for each instance. We discuss the related works in more detail in Appendix B.

## 1.1 OUR CONTRIBUTIONS

Responding to the above limitations, we propose a novel robust learning framework, which is able to defend adversarial attacks targeted at any chosen subset of examples. Specifically, we make the following contributions.

**Learning in presence of perturbation on adversarially selected subset.** We consider an attack model, where the adversary *selectively* perturbs a subset of instances, rather than drawing them uniformly at random. However, the exact choice of the subset or its property remains unknown to the learner during training and validation. Consequently, a learner cannot adapt to such specific attack well in advance through training or cross-validation. To defend these attacks, we introduce a novel adversarial training method, where the learner aims at minimizing the worst-case loss across all the data subsets. Our defense strategy is agnostic to any specific selectivity of the attacked subset. Its key goal is to maintain high accuracy during attacks on any selected subset, rather than providing optimal accuracy for any specific subset.

To this end, we posit our adversarial training task as an instance of min-max optimization problem, where the inner optimization problem seeks the data subset that maximizes the training loss and, the outer optimization problem then minimizes this loss with respect to the model parameters. While training the model, the outer problem also penalizes the loss on the unperturbed instances. This allows us to optimize for the overall accuracy across both perturbed and unperturbed instances.

**Theoretical characterization of our defense objective.** Existing adversarial training methods (Madry et al., 2017; Zhang et al., 2019b; Robey et al., 2021) involve only continuous optimization variables— the model parameters and the amount of perturbation. In contrast, the inner maximization problem in our proposal searches over the worst-case data subset. This translates our optimization task into a parameter estimation problem in conjunction with a subset selection problem, which renders it NP-hard. We provide a useful characterization of the underlying training objective that would help us design approximation algorithm to solve the problem. Given a fixed ML model, we show that the training objective can be expressed as the difference between a monotone $\gamma$-weakly submodular function and a modular function (Theorem 2). This allows us to leverage distorted greedy algorithm (Harshaw et al., 2019) to optimize the underlying objective.

**Approximation algorithms.** We provide ROGET (RObust aGainst adversarial subsETs), a family of algorithms to solve our optimization problem, by building upon the proposal of (Adibi et al., 2021), that admits approximation guarantees. In each iteration, ROGET first applies gradient descent (GD) or stochastic gradient descent step to update the estimate of the model parameters and then applies distorted greedy algorithm to update the estimate of attacked subset of instances. We show that ROGET admits approximation guarantees for convex and non-convex training objective (Thoerem 5), where in the latter case we require that the objective satisfies Polyak-Lojasiewicz (PL) condition (Theorem 4). Our analysis can be applied in any min-max optimization setup where the inner optimization problem seeks to maximize the difference between a monotone $\gamma$-weakly submodular and a modular function and therefore, is of independent interest.

Finally, we provide a comprehensive experimental evaluation of ROGET, by comparing them against seven state-of-the-art defense methods. Here, in addition to hyperparameter set by the baselines in their papers, we also use a new hyperparameter selection method, which is more suited in our setup. Unlike our proposal, the baselines are not trained to optimize for the worst case accuracy. To reduce this gap between the baselines and our method, we tune the hyperparameters of the baselines, which would maximize the minimum accuracy across a large number of subsets chosen for attack. We observe that, ROGET is able to outperform the state-of-the-art defense methods in terms of the overall accuracy across different hyperparameter selection and different subset selection strategies.

## 2 PROBLEM FORMULATION

**Instances, learner's model and the loss function.** We consider a classification setup where $x \in \mathcal{X} = \mathbb{R}^d$ are the features, $y \in \mathcal{Y}$ are the discrete labels. We denote $\{(x_i, y_i)\}_{i \in D}$ to be the training instances where $D$ denotes the training dataset. We use $h_{\boldsymbol{\theta}} \in \mathcal{H}$ to indicate the learner's model,

where $\mathcal{H}$ is the hypothesis class and $\boldsymbol{\theta} \in \Theta$ is the parameter vector of the model. We use the cross entropy loss $\ell(h_{\boldsymbol{\theta}}(\boldsymbol{x}), y)$ in the paper.

**Adversarial perturbation on selected subset.** We assume that the adversary's goal is to *selectively* attack a specific subset of instances $S^{\text{latent}}$— instead of every possible instances in the data or drawing instances uniformly at random. The adversary then uses an adversarial perturbation method to generate $\boldsymbol{x}_i^{\text{adv}}$ using $\boldsymbol{x}_i$ for all for $i \in S^{\text{latent}}$ such that $\boldsymbol{x}_i^{\text{adv}}$ is close to $\boldsymbol{x}_i$, but the model misclassifies $\boldsymbol{x}_i^{\text{adv}}$. Now, it is important to note that neither the strategy behind selecting $S^{\text{latent}}$ nor the adversarial perturbation method is known to the learner. Hence, during training, we use $a_{\boldsymbol{\phi}} : \mathcal{X} \to \mathcal{X}$ as the learner's belief about the adversarial network or the perturbation method with parameter vector $\boldsymbol{\phi} \in \Phi$ similar to (Baluja & Fischer, 2018; Xiao et al., 2018; Mopuri et al., 2018), where $\Phi$ is domain of $\boldsymbol{\phi}$. Many popular attacks like FGSM (Goodfellow et al., 2015), PGD (Madry et al., 2017) are un-parameterized and they induce perturbation of one point independently of others. Still, we assume a parameterized adversary model (Baluja & Fischer, 2018; Xiao et al., 2018; Mopuri et al., 2018) to make the formulation more generalized as one can always overparameterize such a model to induce enough capacity and mimic pointwise attacks.

**Proposed adversarial training problem.** Let us assume, for instance, that the subset selection strategy of the adversary is revealed to the learner. Following this strategy, the learner can easily compute the underlying subset $S \subset D$ to mimic the adversary and minimize the sum of the loss on the perturbed instances $i \in S$ and the unperturbed instances $j \in D \backslash S$. However, in practice, the learner may not have any knowledge about the underlying subset selection strategy. In such a case, the goal of a defense algorithm should be to ensure high overall accuracy in the face of attacks on all possible subsets. To this end, we design an adversarial training framework, which attempts to minimize the worst case loss across all subsets, as described below.

Given a set of training instances $\{(\boldsymbol{x}_i, y_i)\}_{i \in D}$, we defend the attacks on selected subset of instances by training a new model $h_{\boldsymbol{\theta}}$ which minimizes the highest possible loss on the perturbed instances $a_{\boldsymbol{\phi}}(\boldsymbol{x}_i)$ with $i \in S$ across subsets $S \subset D$ of size at most $b$, while ensuring that the new predictions $h_{\boldsymbol{\theta}}(\boldsymbol{x}_j)$ and the labels $y_j$ remain close on the unperturbed instances $j \notin S$. Given the the learner's belief about the adversarial network $a_{\boldsymbol{\phi}}$, we define the learner's loss function as follows:

$$F(h_{\boldsymbol{\theta}}, S \,|\, a_{\boldsymbol{\phi}}) = \frac{1}{|D|} \left[ \sum_{i \in S} \ell(h_{\boldsymbol{\theta}}(a_{\boldsymbol{\phi}}(\boldsymbol{x}_i)), y_i) + \sum_{j \in D \backslash S} \rho \, \ell(h_{\boldsymbol{\theta}}(\boldsymbol{x}_j), y_j) \right]. \tag{1}$$

The parameter $\rho$ is a regularization parameter which gives additional flexibility to control the trade-off between accuracy on clean examples and perturbed examples. Then, we formulate our adversarial training problem as the following bi-level discrete-continuous optimization task.

$$\underset{\boldsymbol{\theta} \in \Theta}{\text{minimize}} \ \underset{S : |S| \leq b}{\text{maximize}} \ \ F(h_{\boldsymbol{\theta}}, S \,|\, a_{\boldsymbol{\phi}^*(\boldsymbol{\theta}, S)}) \tag{2}$$

$$\text{where,} \ \ \boldsymbol{\phi}^*(\boldsymbol{\theta}, S) = \underset{\boldsymbol{\phi} \in \Phi}{\text{argmax}} \sum_{i \in S} [\ell(h_{\boldsymbol{\theta}}(a_{\boldsymbol{\phi}}(\boldsymbol{x}_i)), y_i) - \mu C(a_{\boldsymbol{\phi}}(\boldsymbol{x}_i), \boldsymbol{x}_i)]. \tag{3}$$

The optimization problem (2) is a min-max game where the inner optimization problem aims to find the subset $S$ of size which provides the highest loss and the outer minimization problem aims to find the model $h_{\boldsymbol{\theta}}$ that minimizes this loss.

The optimization problem (3) is the learner's *belief about the adversary's strategy*. It need not be true in practice. In Section 4, we perform experiments when the true adversary's models differ from $a_{\boldsymbol{\phi}}$ during test. Eq. (3) provides the learner's estimate about the parameters of the adversarial network. Here, $C(a_{\boldsymbol{\phi}}(\boldsymbol{x}_i), \boldsymbol{x}_i)$ is cost of perturbing $\boldsymbol{x}_i$ to $a_{\boldsymbol{\phi}}(\boldsymbol{x}_i)$ often measured using different notions of distances, *e.g.*, normed differences or their squares, etc. In such a case, this optimization problem (3) can also be seen as the dual of the constrained optimization problem (Madry et al., 2017; Robey et al., 2021) given by $\max_{\boldsymbol{\phi}} \sum_{i \in S} \ell(h_{\boldsymbol{\theta}}(a_{\boldsymbol{\phi}}(\boldsymbol{x}_i)), y_i)$ such that $C(a_{\boldsymbol{\phi}}(\boldsymbol{x}_i), \boldsymbol{x}_i) \leq \xi(\mu)$ where $\xi$ is dependent on $\mu$.

Note that, the adversary can select the subset $S$ in both deterministic or probabilistic manner. For example, it can attack images of specific persons in facial recognition systems, perturb the instances with stop-signs in traffic signs classification system, etc. On the other hand, it can select instances with probability proportional to the uncertainty of a classifier. Similar to the optimization problems (2)–(3), one can derive a continuous min-max optimization problem when $S$ is drawn from a probability

distribution. We show the connection between this continuous optimization problem and our discrete continuous optimization problem in Appendix C.

**Hardness analysis.** The inner optimization problem in Eq. (2) involves combinatorial search over $S$ for a fixed $\boldsymbol{\theta}$. However, while doing so, it requires to compute $\phi^*(\boldsymbol{\theta}, S)$. This makes our adversarial training problem NP-Hard (see Apppendix C for a proof).

## 3 PROPOSED APPROACH

In this section, we provide algorithms to solve the optimization problem (2). We first characterize it as a difference between a $\gamma$-weakly submodular function and a modular function. Next, we design ROGET, a family of algorithms to solve our adversarial training problem (2).

### 3.1 SET FUNCTION THEORETIC CHARACTERIZATION OF $F$

Here, we provide a characterization of the objective function $F(h_{\boldsymbol{\theta}}, S \,|\, a_{\boldsymbol{\phi}})$ using the notions of monotonicity and $\gamma$-weak submodularity, which would lead us to design an approximation algorithm to solve our training problem (2). To do so, we first formally state the definitions of these notions.

**Definition 1.** *Given a set function $Q : 2^D \to \mathbb{R}$, we define the marginal gain of $Q$ as $Q(k \,|\, S) := Q(S \cup \{k\}) - Q(S)$. The function $Q$ is monotone (non-decreasing) if $Q(k \,|\, S) \geq 0$ for all $k \in D \backslash S$. The function $Q$ is called $\gamma$-weakly submodular if for some $\gamma \in (0, 1]$, we have $\sum_{k \in T \backslash S} Q(k \,|\, S) \geq \gamma[Q(T \cup S) - Q(S)]$ whenever $S \subset T \subseteq D$. Here, $\gamma$ is called the submodularity ratio of $Q$. The function $Q$ is modular if $Q(k \,|\, S) = Q(k \,|\, T)$ for all $S \subset T \subset D$ and $k \in D \backslash T$.*

**Alternative representation of $F(h_{\boldsymbol{\theta}}, S \,|\, a_{\phi^*(\boldsymbol{\theta}, S)})$.** Given a regularization function $R(\boldsymbol{\theta})$ and a regularization parameter $\lambda > 0$, we define two set functions $G(\boldsymbol{\theta}, S)$ and $m(\boldsymbol{\theta}, S)$ as follows:

$$G_\lambda(\boldsymbol{\theta}, S) = \frac{1}{|D|} \left[ \sum_{i \in S} [\lambda R(\boldsymbol{\theta}) + \ell(h_{\boldsymbol{\theta}}(a_{\boldsymbol{\phi}}(\boldsymbol{x}_i)), y_i)] + \sum_{j \in D} \rho\, \ell(h_{\boldsymbol{\theta}}(\boldsymbol{x}_j), y_j) \right] \tag{4}$$

$$m_\lambda(\boldsymbol{\theta}, S) = \frac{1}{|D|} \sum_{i \in S} [\lambda R(\boldsymbol{\theta}) + \rho\, \ell(h_{\boldsymbol{\theta}}(\boldsymbol{x}_j), y_j)]. \tag{5}$$

Then, we represent $F(h_{\boldsymbol{\theta}}, S \,|\, a_{\phi^*(\boldsymbol{\theta}, S)})$ as the difference between the above functions, *i.e.*,

$$F(h_{\boldsymbol{\theta}}, S \,|\, a_{\phi^*(\boldsymbol{\theta}, S)}) = G_\lambda(\boldsymbol{\theta}, S) - m_\lambda(\boldsymbol{\theta}, S). \tag{6}$$

Here, $m_\lambda$ is a modular function. Note that the above equality holds for any value of $\lambda > 0$. However, as we shall see next, the submodularity ratio of $G_\lambda$ depends on $\lambda$ which affects the performance of the approximation algorithm designed in Section 3.2. Next, we present some assumptions which would be used to characterize the above representation of $F$.

**Assumptions 1.** **(1) Lipschitz continuity**: (a) The loss function $\ell(h(\boldsymbol{x}), y)$ is $L_h$-Lipschitz with respect to $h$, (b) $h_{\boldsymbol{\theta}}(\boldsymbol{x})$ is $L_x$-Lipschitz with respect to $\boldsymbol{x}$, (c) the adversarial network $a_{\boldsymbol{\phi}}(.)$ is $L_\phi$-Lipschitz with respect to $\phi$. **(2) Stability of $\phi^*(\boldsymbol{\theta}, S)$:** The learner's estimate about the parameter for the adversarial network is stable (Bousquet & Elisseeff, 2002; Charles & Papailiopoulos, 2018; Hardt et al., 2016), *i.e.*, the solution $\phi^*(\boldsymbol{\theta}, S)$ of the optimization (3) satisfies $\|\phi^*(\boldsymbol{\theta}, S \cup k) - \phi^*(\boldsymbol{\theta}, S)\| \leq \beta/|S|$ for some $\beta > 0$ and all $\boldsymbol{\theta}$. Stability holds for a wide variety of loss functions including convex losses Bousquet & Elisseeff (2002), a class of neural models called Polyak-Lojasiewicz (PL) losses Charles & Papailiopoulos (2018), *etc.*. **(3) Metric property of the cost of perturbation $C$:** The cost of perturbation $C$ used in Eq. (3) is a distance metric. Specifically, it follows the triangle inequality, *i.e.*, $C(\boldsymbol{x}', \boldsymbol{x}) \leq C(\boldsymbol{x}', \boldsymbol{x}'') + C(\boldsymbol{x}'', \boldsymbol{x})$. **(4) Norm-boundedness of $C$:** The cost of perturbation $C$ is a bounded by an $\ell_2$ norm, *i.e.*, $C(\boldsymbol{x}', \boldsymbol{x}) \leq q\|\boldsymbol{x}' - \boldsymbol{x}\|$. In the context of several prior works (Goodfellow et al., 2015; Madry et al., 2017) use $\ell_\infty$ distance which is bounded by $\ell_2$ norm. **(5) Boundeness of $\Theta$ and $\Phi$:** We assume that the parameter space of both the learning model and adversarial network are bounded, *i.e.*, $\|\theta\|_2 \leq \theta_{\max}$ and $\|\phi\|_2 \leq \phi_{\max}$.

**Monotonicity and $\gamma$-weakly submodularity of $G_\lambda$.** We now present our results on monotonicity and $\gamma$-weak submodularity of $G_\lambda$ (see Appendix E.1 for a proof).

**Theorem 2.** *Given Assumptions 1 let there be a value of minimum non-zero constant $\lambda_{\min} > 0$ such that $\ell^* = \min_{\boldsymbol{x} \in \mathcal{X}, y \in \mathcal{Y}} \min_{\boldsymbol{\theta}} [\lambda_{\min} R(\boldsymbol{\theta}) + \ell(h_{\boldsymbol{\theta}}(\boldsymbol{x}), y) - 2q\mu\beta L_\phi] > 0$ where $q$ and $\beta$ are given in Assumptions 1. Then, for $\lambda > \lambda_{\min}$, we have the following statements:*

*(1)* $G_\lambda(S)$ *is monotone in* $S$.

*(2)* $G_\lambda(S)$ $\gamma$-*weakly submodular with* $\gamma > \gamma^* = \ell^*[\ell^* + 2L_h L_x L_\phi \phi_{\max} + 3q\mu\beta L_\phi]^{-1}$.

### 3.2 ROGET: PROPOSED ALGORITHM TO SOLVE THE ADVERSARIAL TRAINING PROBLEM (2)

In this section, we develop ROGET to solve our optimization problem (2) by building upon the proposal of Adibi et al. (2021). However, they design algorithms to solve the min-max problem on those functions that are submodular in $S$ and convex in $\boldsymbol{\theta}$. In contrast, ROGET applies to the proposed objective $F(h_{\boldsymbol{\theta}}, S \mid a_{\boldsymbol{\phi}^*(\boldsymbol{\theta}, S)})$ (1) which is the difference between a $\gamma$-weakly submodular function and a modular function, and may also be non-convex in $\boldsymbol{\theta}$. In the following, we describe them in details beginning with an outline of the proposed algorithm.

**Development of ROGET.** Our key goal is to optimize $\min_\theta \max_S F(h_{\boldsymbol{\theta}}, S \mid a_{\boldsymbol{\phi}^*(\boldsymbol{\theta}, S)})$. Now, we aim to develop an algorithm which would iterate over the inner and outer optimization problem and gradually refine $S$ and $\theta$.

*Iterative method for the inner optimization on* $S$: Now, given a fixed $\boldsymbol{\theta}$, the inner maximization problem becomes a set function optimization problem. If $F$ were a monotone submodular function, then we could have applied the well known greedy algorithm (Nemhauser et al., 1978). At each step, such a greedy algorithm would seek for an element $e$ that would maximize the marginal gain $F(h_{\boldsymbol{\theta}}, S \cup e \mid a_{\boldsymbol{\phi}^*(\boldsymbol{\theta}, S \cup e)}) - F(h_{\boldsymbol{\theta}}, S \mid a_{\boldsymbol{\phi}^*(\boldsymbol{\theta}, S)})$ and update $S \to S \cup \{e\}$.

However, in our context, the function $F$ may neither be monotone nor submodular and thus, we cannot apply the greedy algorithm for iterating over the inner optimization loop Nevertheless, we note that it can be expressed as difference between the $\gamma$-weakly submodular function $G_\lambda(\boldsymbol{\theta}, S)$ and the modular function $m_\lambda(\boldsymbol{\theta}, S)$, as suggested by Eq. (6). As a result, we can adopt the stochastic distorted greedy algorithm (Harshaw et al., 2019) which, instead

---

**Algorithm 1** ROGET Algorithm

**Require:** Training set $D$, regularization parameter $\lambda$, budget $b$, # of iterations $T$, learning rate $\eta$.
1: INIT($h_{\boldsymbol{\theta}}$), $\widehat{S}_0 \leftarrow \emptyset$
2: **for** $t = 0$ to $T - 1$ **do**
3:     $\widehat{\boldsymbol{\theta}}_{t+1} \leftarrow \text{TRAIN}(\widehat{\boldsymbol{\theta}}_t, \widehat{S}_t, \eta_t)$
4:     $\widehat{S}_{t+1} \leftarrow \text{SDG}(G_\lambda, m_\lambda, \widehat{\boldsymbol{\theta}}_{t+1}, b)$.
5: $\widehat{\boldsymbol{\theta}} = \boldsymbol{\theta}_T$
6: **return** $\widehat{\boldsymbol{\theta}}, \widehat{S}_T$

---

1: **procedure** SDG $(G_\lambda, m_\lambda, \widehat{\boldsymbol{\theta}}, b)$
2:     $S \leftarrow \emptyset$
3:     **for** $s \in [b]$ **do**
4:         $\gamma' \leftarrow (1 - \gamma/b)^{b-s-1}$
5:         Randomly draw a subset $B$ from $D$
6:         $\boldsymbol{M} \leftarrow [\gamma' G_\lambda(\boldsymbol{\theta}, e \mid S) - m_\lambda(\boldsymbol{\theta}, \{e\})]_{e \in B}$
7:         $e^* = \underset{e \in B}{\text{argmax}} \, \boldsymbol{M}[e]$
8:         **if** $\gamma' G_\lambda(\boldsymbol{\theta}, e^* \mid S) - m_\lambda(\boldsymbol{\theta}, \{e^*\}) \geq 0$ **then**
9:             $S \leftarrow S \cup \{e\}$
10:     **return** $S$

---

1: **procedure** TRAIN($\widehat{\boldsymbol{\theta}}, S, \eta$)
2:     //SGD for $k$ steps
3:     **for** $i \in [k]$ **do**
4:         Draw $i \sim D$ uniformly at random
5:         **if** $i \in S$ **then**
6:             $\widehat{\boldsymbol{\theta}} \leftarrow \widehat{\boldsymbol{\theta}} - \eta \frac{\partial \ell(h_{\boldsymbol{\theta}}(a_{\boldsymbol{\phi}}(\boldsymbol{x}_i)), y_i)}{\partial \boldsymbol{\theta}} \big|_{\boldsymbol{\theta} = \widehat{\boldsymbol{\theta}}}$
7:         **else**
8:             $\widehat{\boldsymbol{\theta}} \leftarrow \widehat{\boldsymbol{\theta}} - \eta \frac{\rho \, \partial \ell(h_{\boldsymbol{\theta}}(\boldsymbol{x}_i), y_i)}{\partial \boldsymbol{\theta}} \big|_{\boldsymbol{\theta} = \widehat{\boldsymbol{\theta}}}$
9:     **return** $\widehat{\boldsymbol{\theta}}$

---

of maximizing exact marginal gain, maximizes a distorted marginal gain $(1 - \gamma)^{b-s-1} G_\lambda(\boldsymbol{\theta}, e \mid S) - m_\lambda(\boldsymbol{\theta}, \{e\})$ for all $e \notin S$ at each step (procedure SDG in Algorithm 1). If this distorted marginal gain is positive, we update $S \to S \cup \{e\}$. This method gives an approximation guarantee for the inner optimization problem (due to Harshaw et al. (2019)).

**Theorem 3** (Harshaw et al. (2019))**.** *Given a fixed* $\boldsymbol{\theta}$ *and that the size of the set* $B$ *is* $|B| = (|D|/b) \log(1/\delta)$ *in line 5 of the procedure* $SDG$. *If the procedure SDG returns* $S$ *as the output, then we have*

$$\mathbb{E}[G_\lambda(\boldsymbol{\theta}, S) - m_\lambda(\boldsymbol{\theta}, S)] \geq [1 - \exp(-\gamma)]G_\lambda(\boldsymbol{\theta}, OPT) - m_\lambda(\boldsymbol{\theta}, OPT). \quad (7)$$

*where* $OPT$ *is the optimal solution of the inner optimization problem. Here, expectation is carried out over the randomness of selection of* $B$.

*Iterative routine for outer optimization on* $\boldsymbol{\theta}$: Having updated $S$ using the distorted greedy algorithm described above, we minimize the $F(h_{\boldsymbol{\theta}}, S \mid a_{\boldsymbol{\phi}^*(\boldsymbol{\theta}, S)})$ with respect to $\boldsymbol{\theta}$ using few steps of gradient descent (called as $k$-SGD) per each round of update.

*Outline of* ROGET: We sketch the pseudocode of ROGET in Algorithm 1. It updates $\boldsymbol{\theta}$ and $S$ in an iterative manner. During iteration $t$, ROGET updates $\widehat{\boldsymbol{\theta}}_t \to \widehat{\boldsymbol{\theta}}_{t+1}$ by running few steps of gradient descent (line 3)— where we fix $S$ at $\widehat{S}_t$ and attempt to reduce the loss $F(h_{\boldsymbol{\theta}}, \widehat{S}_t \mid a_{\boldsymbol{\phi}^*(\boldsymbol{\theta}, \widehat{S}_t)})$ with

respect to $\boldsymbol{\theta}$. In the next step, we fix $\widehat{\boldsymbol{\theta}}_t$ and update $\widehat{S}_t \to \widehat{S}_{t+1}$ using stochastic distorted greedy algorithm (SDG, line 4). Note that during each time $t$, we compute $S_t$ for a fixed $\widehat{\boldsymbol{\theta}}$ sequentially in $b$ steps. Having obtained $S$ at $s < b$, we update it such that it has highest positive *distorted* marginal gain $(1-\gamma)^{b-s-1} G_\lambda(\boldsymbol{\theta}, e \,|\, S) - m_\lambda(\boldsymbol{\theta} \,|\, \{e\})$ and then, it is included into the set $S$ to have $S \cup \{e\}$.

**Approximation guarantees.** In general, $F$ is a highly non-convex function of $\theta$ and therefore, obtaining an approximation guarantee for any general $F$ is extremely challenging. Hence, we derive approximation guarantee for a restrictive class of loss functions, called Polyak-Lojasiewicz (PL) loss functions. A function $f$ is Polyak-Lojasiewicz if $||\nabla_\theta f(\boldsymbol{\theta})|| \geq \sigma[f(\boldsymbol{\theta}) - \min_{\theta'} f(\boldsymbol{\theta}')]$ In Appendix E.2, we also present our results when $F$ is convex in $\theta$. Next, we initiate our discussion with a few more assumptions in addition to Assumptions 1.

**Assumptions 2. (1) L-smoothness of $F$.** For all $\boldsymbol{\theta} \in \Theta$ and for all $S \in 2^V$, we have $||\nabla_\theta F(h_\theta, S \,|\, a_{\phi^*(\boldsymbol{\theta},S)}) - \nabla_{\theta'} F(h_{\theta'}, S \,|\, a_{\phi^*(\boldsymbol{\theta}',S)})|| \leq L||\boldsymbol{\theta} - \boldsymbol{\theta}'||$. **(2) Boundedness of gradients of $F$.** We have $||\nabla_\theta F(h_\theta, S \,|\, a_{\phi^*(\boldsymbol{\theta},S)})|| \leq \nabla_{\max}$ for all $\boldsymbol{\theta} \in \Theta$ and for all $S \in 2^V$. **(3) Boundedness of loss $\ell$ and model $h_\theta$.** For all $\boldsymbol{\theta} \in \Theta$, $|\ell(h_\theta(\boldsymbol{x}_i), y_i)| < \ell_{\max}$ and $||h_\theta|| < h_{\max}$. **(4) Size of $B$ in SDG procedure.** The size of the set $B_k$ is $|B_k| = (|D|/b) \log(1/\delta)$ (line 5 in procedure SDG used in Algorithm 1). **(5) Adversarial network always perturbs a feature.** The cost of perturbation $C(a_{\phi^*(\boldsymbol{\theta},S)}(\boldsymbol{x}_i), \boldsymbol{x}_i) > C_{\min} > 0$. Moreover, $C_{\min} > \lambda\theta_{\max}^2(e^{-\gamma^*} + \delta)/(1 - e^{-\gamma^*} - \delta)$ where $\gamma^*$ is the submodularity ratio (Theorem 2).

We provide justification for all the assumptions in Appendix D. Now we state the approximation guarantee of Algorithm 1 for Polyak-Lojasiewicz (PL) loss (see Appendix E.2 for proof).

**Theorem 4.** *Given the conditions of Theorem 2 and Assumptions 2, let $F(h_\theta, S \,|\, a_{\phi^*(\boldsymbol{\theta},S)})$ be a PL function in $\boldsymbol{\theta}$ for all $S$,* i.e., $||\nabla_\theta F(h_\theta, S \,|\, a_{\phi^*(\boldsymbol{\theta},S)})|| \geq \sigma[F(h_\theta, S \,|\, a_{\phi^*(\boldsymbol{\theta},S)}) - \min_{\theta'} F(h_{\theta'}, S \,|\, a_{\phi^*(\boldsymbol{\theta}',S)})]$ *for some constant $\sigma$. If we set the learning rate $\eta = 1/kT$ in Algorithm 1, then for $T = O(1/k\epsilon)$ iterations, $\delta < (1 - e^{-\gamma^*})$ and $\rho < [C_{\min}((e^{-\gamma^*} + \delta)^{-1} - 1) - \lambda\theta_{\max}^2]/\ell_{\max}$, we have the following approximation guarantee for Algorithm 1:*

$$\min_t \mathbb{E}[F(h_{\widehat{\boldsymbol{\theta}}_t}, \widehat{S}_T \,|\, a_{\phi^*(\widehat{\boldsymbol{\theta}}_t, \widehat{S}_T)})] \leq \max_S \min_t \mathbb{E}[F(h_{\widehat{\boldsymbol{\theta}}_t}, S \,|\, a_{\phi^*(\widehat{\boldsymbol{\theta}}_t, S)})]$$

$$\leq \left[1 - (e^{-\gamma^*} + \delta)/\kappa\right]^{-1}\left(OPT + 2L^2 h_{\max}/\sigma + \epsilon\right) \qquad (8)$$

*where $\boldsymbol{\theta}_t$ is the iterate in Line 3 in Algorithm 1, $OPT$ is the value at the optimum solution of our adversarial training problem* (2)*, $\kappa = C_{\min}/(\lambda\theta_{\max}^2 + \rho\ell_{\max} + C_{\min})$. Here $\theta_{\max}$ is defined in Assumption 1 and $\ell_{\max}, C_{\min}$ are defined in Assumptions 2.*

Note that, due to non-convexity, we do not provide guarantee on final $\widehat{\boldsymbol{\theta}} = \boldsymbol{\theta}_T$, but on the iterates $\widehat{\boldsymbol{\theta}}_t$; and, the approximation factor suffers from an additional offset $2L^2 h_{\max}/\sigma$, even as $\epsilon \to 0$. In Appendix E.2, we present our results when $F$ is convex, where this bound becomes stronger.

## 4 EXPERIMENTS

In this section, we conduct experiments with real world datasets which show that ROGET achieves a better overall accuracy than the state-of-the-art methods for both white-box and black-box attacks. Appendix H contains additional results.

### 4.1 EXPERIMENTAL SETUP

**Datasets and state-of-the art competitors.** We experiment with CIFAR10 (Krizhevsky et al., 2014) and Fashion MNIST (Xiao et al., 2017) (FMNIST) in this section. In Appendix H, we also report results on CIFAR100. We consider seven state-of-the-art methods to compare our method– they are: GAT (Sriramanan et al., 2020), FBF (Wong et al., 2019), TRADES (Zhang et al., 2019b), Nu-AT (Sriramanan et al., 2021), MART (Wang et al., 2019), PGD-AT (Madry et al., 2017) and RFGSM-AT (Tramèr et al., 2018). Appendix G contains more details about the baselines.

**Evaluation protocol.** We split the datasets into training ($D_{\text{Tr}}$), validation ($D_{\text{Val}}$), and test set ($D_{\text{Test}}$) in the ratio of 4:1:1 and 5:1:1 for CIFAR10 and FMNIST respectively. We use the validation set for hyperparameter selection (mentioned later), early stopping (See Appendix G for details), etc. We report three types of accuracies, *viz.*, (i) accuracy on clean examples $\mathcal{A}_{\text{clean}} = \mathbb{P}(y = \hat{y} \,|\, \boldsymbol{x}$ is *not* chosen for attack), (ii) robustness to adversarial perturbations measured using the accuracy

on the perturbed examples $\mathcal{A}_{\text{robust}} = \mathbb{P}(y = \hat{y} \mid \boldsymbol{x}$ is chosen for attack) and (iii) overall accuracy $\mathcal{A} = \mathbb{P}(y = \hat{y})$.

**Models for the learner.** We consider two candidates for $a_\phi$, which model the learner's belief about the adversary during training. Specifically, we set either $a_\phi = \text{PGD}$ (Madry et al., 2017) or $a_\phi = \text{AdvGAN}$ (Xiao et al., 2018). This gives rise to two variants of our model, *viz.*, ROGET $(a_\phi = \text{PGD})$ and ROGET $(a_\phi = \text{AdvGAN})$. We use ResNet-18 (He et al., 2016) and LeNet-5 (LeCun et al., 2015) architectures for CIFAR10 and FMNIST datasets respectively.

**Model for the adversarial perturbation.** We consider two white-box attacks, *viz.*, PGD (Madry et al., 2017) and Auto Attack (AA) (Croce & Hein, 2020) as well as three black-box attacks, *viz.*, Square (Andriushchenko et al., 2020) and black-box MI-FGSM (Dong et al., 2018) and AdvGAN (Xiao et al., 2018) to perturb test samples. The exact details of the attacks are given in Appendix G.

**The subset selection strategy of the adversary.** In addition to the adversarial perturbation mechanism, the adversary also has a strategy to select a subset $S^{\text{latent}}$ of test instances to attack on. We experiment with two latent subset selection mechanisms. (1) *Uncertainty based subset selection:* Here, the adversary selects the top-$10\%$ instances in terms of the prediction uncertainty of a classifier trained on clean examples from $D_{\text{Tr}}$. Here, the prediction uncertainty for an instance $\boldsymbol{x}$ is computed as $u(\boldsymbol{x}) = 1 - \max_y h(\boldsymbol{x})[y]$. (2) *Label based subset selection:* Here, the adversary selects instances who have a specific label $y \in \mathcal{Y}$ to perform attack, *e.g.*, $S^{\text{latent}} = \{i \mid y_i = \texttt{aeroplane}\}$. Note that, the underlying subset selection strategy is realized only during test— it is not revealed to the learner during training and validation, in practice. Appendix H also presents other strategies.

**Hyperparameter selection.** Suppose, for instance, that the adversary's latent strategy of selecting a subset to perform attack is known during validation stage. Then one could easily simulate such strategy to create the perturbed instances in the validation set and use the resulting validation set to cross validate the underlying hyperparameters. However, the subset selection strategy is never revealed to the learner during both training and *validation* stage. Thus, the selection of hyperparameters becomes challenging and it completely depends on the underlying assumption about the adversary. In such a situation, we experiment with two methods for hyperparameter selection. (1) *Default selection:* Here, we use the hyperparameters of the baselines directly used in their original papers and codes (details in Appendix G). (2) *Worst case selection:* The key goal of ROGET is to learn a model which minimizes the worst case loss across all data subsets. In a similar spirit, here we aim to select the hyperparameters which would maximize the minimum accuracy across a large number of subsets that underwent attacks. Formally, if we denote $\mathcal{A}(\boldsymbol{\beta}, S)$ to be the overall accuracy of the trained model on the subset $S$, with hyperparameters $\boldsymbol{\beta}$, we aim to estimate $\boldsymbol{\beta}^* = \text{argmax}_{\boldsymbol{\beta}} \min_S \mathcal{A}(\boldsymbol{\beta}, S)$. To this aim, we draw $R = 10000$ subsets $\{S_j\}_{j=1}^R$ uniformly at random from $D_{\text{Val}}$ having size $|S_j| = 0.05 |D_{\text{Val}}|$ and search over the hyperparameters $\boldsymbol{\beta}$ that maximizes the minimum accuracy across these $R$ subets. In our case as well, we tune $\rho$ in similar manner to obtain $\rho^*$. Hence, the goal of this type of hyperparameter selection is same as the key goal of ROGET. Thus, it will provide a fair comparison between all the methods.

## 4.2 RESULTS

**Uncertainty based subset selection with default hyperparameters.** Here we compare our method against all the state-of-the-art defense methods under the default hyperparameter selection on CIFAR10 and FMNIST, where all the hyperparameters of the baselines are set as what is reported in their respective work and for our method, we experiment with different values of $\rho$. Moreover, the adversary adopts uncertainty based subset selection, where it selects top $10\%$ of the test set based on the uncertainty of a classifier trained on all clean examples. We report the results in Table 1 and make the following observations. **(1)** ROGET $(a_\phi = \text{PGD})$ and ROGET $(a_\phi = \text{AdvGAN})$ achieve better overall accuracy $\mathcal{A}$ than the existing methods for all attacks on both the datasets (except for Square attack on CIFAR10). Among the two variants of ROGET, ROGET $(a_\phi = \text{AdvGAN})$ is the predominant winner. **(2)** On CIFAR10, ROGET outperforms the baselines (except RFGSM-AT) in terms of the clean accuracy $\mathcal{A}_{\text{clean}}$. This is because, ROGET is trained to defend much better when the adversary plans to attack a subset of instances rather than every instance. In contrast, the baselines are often trained in a pessimistic manner— they assume attack on every possible instance and consequently show sub-optimal accuracy on the clean examples. **(3)** There is no consistent winner among the baselines in terms of the robustness $\mathcal{A}_{\text{robust}}$. Robust accuracy of ROGET is often

| CIFAR10 | $\mathcal{A}_{\text{clean}}$ | PGD | | AA | | Square | | MIFGSM | | AdvGAN | |
|---|---|---|---|---|---|---|---|---|---|---|---|
| | | $\mathcal{A}_{\text{robust}}$ | $\mathcal{A}$ | $\mathcal{A}_{\text{robust}}$ | $\mathcal{A}$ | $\mathcal{A}_{\text{robust}}$ | $\mathcal{A}$ | $\mathcal{A}_{\text{robust}}$ | $\mathcal{A}$ | $\mathcal{A}_{\text{robust}}$ | $\mathcal{A}$ |
| GAT | 78.76 | 59.65 | 76.85 | 46.80 | 75.56 | 59.47 | 76.83 | 49.82 | 75.86 | 82.04 | 79.08 |
| FBF | 74.92 | 72.04 | 74.64 | 31.10 | 70.54 | 31.59 | 70.59 | 36.09 | 71.15 | 37.22 | 71.15 |
| TRADES | 80.25 | 64.68 | 78.69 | 53.37 | 77.56 | 63.06 | 78.53 | 54.82 | 77.70 | 83.82 | 80.60 |
| Nu-AT | 83.12 | 57.15 | 80.53 | 47.44 | 79.56 | 61.78 | 80.99 | 50.14 | 79.83 | 86.82 | 83.49 |
| MART | 81.30 | 65.04 | 79.68 | 52.21 | 78.39 | 63.78 | 79.55 | 54.15 | 78.59 | 84.75 | 81.65 |
| PGD-AT | 83.36 | 57.33 | 80.76 | 48.50 | 79.88 | 62.71 | 81.30 | 50.00 | 80.03 | 87.13 | 83.74 |
| RFGSM-AT | 89.10 | 33.25 | 83.51 | 26.37 | 82.83 | 57.68 | 85.96 | 23.21 | 82.51 | 89.42 | 86.20 |
| ROGET (PGD, $\rho = 1.0$) | 85.27 | 55.79 | 82.32 | 47.72 | 81.49 | 63.18 | 83.06 | 47.90 | 81.53 | 88.96 | 85.64 |
| ROGET (AdvGAN, $\rho = 1.0$) | 87.86 | 47.97 | 83.87 | 39.97 | 83.07 | 62.21 | 85.29 | 40.27 | 83.10 | 91.53 | 88.22 |
| ROGET (AdvGAN, $\rho = 1.5$) | 88.23 | 44.24 | 83.83 | 36.48 | 83.05 | 61.01 | 85.51 | 37.42 | 83.15 | 91.96 | 88.60 |

| FMNIST | $\mathcal{A}_{\text{clean}}$ | PGD | | AA | | Square | | MIFGSM | | AdvGAN | |
|---|---|---|---|---|---|---|---|---|---|---|---|
| | | $\mathcal{A}_{\text{robust}}$ | $\mathcal{A}$ | $\mathcal{A}_{\text{robust}}$ | $\mathcal{A}$ | $\mathcal{A}_{\text{robust}}$ | $\mathcal{A}$ | $\mathcal{A}_{\text{robust}}$ | $\mathcal{A}$ | $\mathcal{A}_{\text{robust}}$ | $\mathcal{A}$ |
| GAT | 91.47 | 3.18 | 82.64 | 0.00 | 82.32 | 0.01 | 82.32 | 8.32 | 83.15 | 29.09 | 85.24 |
| FBF | 80.47 | 14.29 | 73.85 | 66.74 | 79.10 | 68.43 | 79.27 | 84.81 | 80.90 | 85.94 | 81.02 |
| TRADES | 85.97 | 65.95 | 83.97 | 54.58 | 82.83 | 57.88 | 83.16 | 81.47 | 85.52 | 70.15 | 84.39 |
| Nu-AT | 90.41 | 5.13 | 81.88 | 0.00 | 81.37 | 0.06 | 81.38 | 18.24 | 83.19 | 15.72 | 82.94 |
| MART | 78.94 | 77.72 | 78.82 | 59.73 | 77.02 | 61.46 | 77.19 | 79.81 | 79.03 | 70.10 | 78.06 |
| PGD-AT | 83.77 | 79.62 | 83.36 | 53.62 | 80.76 | 57.05 | 81.10 | 83.73 | 83.77 | 72.88 | 82.69 |
| RFGSM-AT | 89.11 | 14.51 | 81.65 | 0.00 | 80.20 | 0.04 | 80.20 | 33.26 | 83.52 | 21.13 | 82.31 |
| ROGET (PGD, $\rho = 1.0$) | 86.98 | 63.45 | 84.63 | 49.67 | 83.25 | 54.24 | 83.71 | 80.99 | 86.38 | 69.17 | 85.20 |
| ROGET (AdvGAN, $\rho = 1.0$) | 87.85 | 35.95 | 82.66 | 37.08 | 82.78 | 43.25 | 83.39 | 81.17 | 87.19 | 71.46 | 86.22 |
| ROGET (AdvGAN, $\rho = 0.01$) | 86.23 | 72.61 | 84.87 | 54.86 | 83.10 | 58.19 | 83.43 | 83.97 | 86.01 | 74.30 | 85.04 |

Table 1: Performance comparison under *default hyperparameter setting* for two white box attacks (PGD and AA) and three black box attacks (Square, MIFGSM, and AdvGAN). Here, the hyperparameters are set as what the baselines mentioned in their papers. We report (percentage) (i) accuracy on the clean examples $\mathcal{A}_{\text{clean}}$, (ii) robustness to the adversarial perturbations $\mathcal{A}_{\text{robust}}$ and (iii) overall accuracy $\mathcal{A}$. Here, the adversary adopts uncertainty based subset selection to perform attack— the true subset chosen for attack $S^{\text{latent}}$ consists of top $10\%$ test instances in terms of the uncertainty of a classifier trained on all the clean examples. (Yellow) Green indicate the (second) best performers.

| CIFAR10 | $\mathcal{A}_{\text{clean}}$ | PGD | | AA | | Square | | MIFGSM | | AdvGAN | |
|---|---|---|---|---|---|---|---|---|---|---|---|
| | | $\mathcal{A}_{\text{robust}}$ | $\mathcal{A}$ | $\mathcal{A}_{\text{robust}}$ | $\mathcal{A}$ | $\mathcal{A}_{\text{robust}}$ | $\mathcal{A}$ | $\mathcal{A}_{\text{robust}}$ | $\mathcal{A}$ | $\mathcal{A}_{\text{robust}}$ | $\mathcal{A}$ |
| GAT | 77.55 | 63.26 | 76.12 | 49.73 | 74.77 | 60.07 | 75.80 | 52.91 | 75.09 | 81.07 | 77.91 |
| FBF | 74.92 | 72.04 | 74.64 | 31.10 | 70.54 | 31.59 | 70.59 | 36.09 | 71.04 | 37.22 | 71.15 |
| TRADES | 83.63 | 51.43 | 80.41 | 42.55 | 79.52 | 58.60 | 81.13 | 48.00 | 80.07 | 87.37 | 84.02 |
| Nu-AT | 84.66 | 54.64 | 81.66 | 45.79 | 80.77 | 61.34 | 82.33 | 47.41 | 80.93 | 88.18 | 85.01 |
| MART | 80.49 | 61.30 | 78.57 | 49.32 | 77.37 | 61.26 | 78.57 | 51.13 | 77.56 | 84.25 | 80.87 |
| PGD-AT | 83.36 | 57.33 | 80.76 | 48.50 | 79.88 | 62.71 | 81.30 | 50.00 | 80.03 | 87.13 | 83.74 |
| RFGSM-AT | 85.84 | 45.94 | 81.85 | 37.49 | 81.00 | 60.25 | 83.28 | 37.91 | 81.04 | 89.42 | 86.20 |
| ROGET (PGD) | 86.63 | 51.16 | 83.09 | 43.06 | 82.28 | 62.33 | 84.20 | 42.20 | 82.19 | 90.15 | 86.99 |
| ROGET (AdvGAN) | 87.86 | 47.97 | 83.87 | 39.97 | 83.07 | 62.21 | 85.29 | 40.27 | 83.10 | 91.53 | 88.22 |

Table 2: Performance comparison under *worst case hyperparameter setting* on CIFAR10. Numbers in green (yellow) indicate the best (second best) performers.

competitive, *e.g.*, in CIFAR10, ROGET ($a_\phi = \text{AdvGAN}$) is the best performer for AdvGAN attack and ROGET ($a_\phi = \text{PGD}$) is the second best performer for Square attack. In FMNIST, ROGET ($a_\phi = \text{AdvGAN}$) is the second best performer for MIFGSM and AdvGAN attack.

**Uncertainty based subset selection with worst case hyperparameter setup.** Next, we tune the hyperparameters of all the methods using the worst case hyperparameter tuning, where we select the hyperparameters that would maximize the minimum accuracy across a large number of subsets of the validation set. For ROGET, we tune $\rho$ in the same manner to obtain $\rho^*$. We present the results in Table 2 for CIFAR100 and make the following observations. **(1)** ROGET ($a_\phi = \text{PGD}, \rho = \rho^*$) and ROGET ($a_\phi = \text{AdvGAN}, \rho = \rho^*$) outperform all the baselines in terms of the overall accuracy $\mathcal{A}$. **(2)** In most of the cases, the overall accuracy $\mathcal{A}$ of ROGET with the worst case hyperparameter $\rho = \rho^*$ is better than $\mathcal{A}$ with the default hyperparameter $\rho = 1$. In contrast, the worst case hyperparameter selection improves $\mathcal{A}$ only for Nu-AT and TRADES among the baselines, from its performance with the default hyperparameter setup (Table 1 vs. Table 2).

**Evaluation on label based subset selection strategy.** We now consider label based subset selection strategy of the adversary. We report the results for AA attack in Table 3, where we also consider class focused online learning (CFOL) (Anonymous, 2022) as an additional baseline which provides guarantee on worst class loss in presence of adversarial attack. We observe that ROGET ($a_\phi = \text{AdvGAN}$) performs the best followed by ROGET ($a_\phi = \text{PGD}$) for all classes. Additional results can be found in Appendix H.

**Impact of revealing the true subset selection strategy during validation.** In practice, the learner does not know the adversary's true (uncertainty based) subset selection strategy during training and

validation. Here, we leak this information to the learner during validation. Then, we mimic the adversary's true strategy to select the subsets from the validation set and perform attack on them.

|  | Airplane | Dog | Truck |
|---|---|---|---|
| GAT | 75.19 | 75.06 | 75.28 |
| FBF | 72.39 | 71.77 | 67.11 |
| TRADES | 80.17 | 79.52 | 80.44 |
| Nu-AT | 81.32 | 80.54 | 81.68 |
| MART | 77.69 | 77.34 | 78.02 |
| PGD-AT | 80.38 | 79.60 | 80.45 |
| RFGSM-AT | 81.80 | 80.84 | 81.46 |
| CFOL | 70.10 | 69.61 | 70.63 |
| ROGET (PGD) | 82.97 | 84.19 | 83.68 |
| ROGET (AdvGAN) | 84.24 | 85.63 | 84.03 |

Table 3: $\mathcal{A}$ for label based subset selection with AA attack on CIFAR10. Green (Yellow) indicates the (second) best method.

|  | Default | Worst | Oracle |
|---|---|---|---|
| GAT | 76.85 | 76.12 | 81.13 |
| FBF | 74.64 | 74.64 | 74.64 |
| TRADES | 78.69 | 80.41 | 81.33 |
| Nu-AT | 80.53 | 81.66 | 83.85 |
| MART | 79.68 | 78.57 | 79.94 |
| PGD-AT | 80.76 | 80.76 | 80.76 |
| RFGSM-AT | 83.51 | 81.85 | 81.85 |
| ROGET (PGD) | 82.32 | 83.09 | 83.65 |
| ROGET (AdvGAN) | 83.87 | 83.87 | 83.83 |

Table 4: Revealing the oracle subset selection strategy (uncertainty) on $\mathcal{A}$ for PGD attack on CIFAR10.

Next, we select the hyperparameters resulting in highest overall validation accuracy. Table 4 compares this strategy ("oracle") with previous hyperparameter selection strategies (default and worse). We make the following observations. (1) ROGET shows very stable performance across different hyperparameter selection methods. (2) GAT, TRADES, Nu-AT and MART significantly improve the performance and Nu-AT outperforms ROGET by a small margin. (3) ROGET's focus is to maintain good accuracy across all subsets. Hence, the performance of ROGET, in absence of any knowledge about the adversary's selected subset, becomes very close to or even better than the baselines having full knowledge of oracle selection.

**Variation of $\mathcal{A}$ vs $|S^{\text{latent}}|$.** The learner's estimate about $|S^{\text{latent}}|$, the number of instances chosen for perturbation, can indeed be heavily inaccurate. In this context, we train both the variants of ROGET using $b = 0.1|D_{\text{Tr}}|$ and evaluate using different number of instances $|S^{\text{latent}}|$ perturbed during test. In Figure 5, we observe that the performances of all the models deteriorate; and our model performs better than the baselines for $|S^{\text{latent}}|/|D^{\text{test}}| \leq 30\%$.

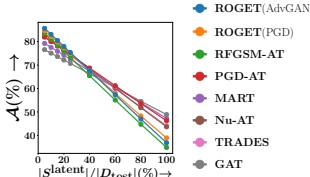

Figure 5: $\mathcal{A}$ vs. $|S^{\text{latent}}|$

**Comparing robustness subject to a minimum overall accuracy.** In Tables 1- 3, we reported robust and overall accuracy for fixed set of hyperparameters. By tuning these hyperparameters, one can improve robust accuracy by sacrificing overall accuracy. Therefore, here we aim to compare the robust accuracy, subject to the condition that the overall accuracy for all methods crosses some threshold. Specifically, we first tune the hyperparameters of all the methods to ensure that the overall accuracy of all methods reaches a given threshold and then compare their robustness. If $P$ indicates the hyperparameters, then we find $\max_P \mathcal{A}_{\text{robust}}(P)$ such that $\mathcal{A}(P) \geq a$ for some given $a$. Results on CIFAR10 for $a = 0.81$ are shown in Table 6. We observe that ROGET (PGD) is the best performer in terms of robust accuracy and ROGET (AdvGAN) is the best performer in terms of overall accuracy. ROGET (AdvGAN) is the second-best performer in terms of robust accuracy. More related results are in Appendix H.

| CIFAR10 | PGD | | Square | |
|---|---|---|---|---|
| | $\mathcal{A}_{\text{robust}}$ | $\mathcal{A}$ | $\mathcal{A}_{\text{robust}}$ | $\mathcal{A}$ |
| GAT | 35.68 | 81.13 | 62.05 | 81.56 |
| TRADES | 53.78 | 81.01 | 61.13 | 81.74 |
| Nu-AT | 54.64 | 81.66 | 61.34 | 82.33 |
| PGD-AT | 35.35 | 81.64 | 55.84 | 81.88 |
| RFGSM-AT | 45.94 | 81.85 | 60.25 | 83.28 |
| ROGET (PGD) | 59.42 | 81.40 | 64.73 | 82.60 |
| ROGET (AdvGAN) | 55.73 | 82.85 | 63.93 | 83.67 |

Table 6: Comparison of $\mathcal{A}_{\text{robust}}$, subject to $\mathcal{A}_{\text{clean}} > 0.81$.

## 5 CONCLUSION

In this paper we motivated a novel setting in adversarial attacks—where an attacker aims to perturb only a subset of the dataset instead of the entire dataset. We presented a defense strategy, ROGET which trains a robust model as a min-max game involving worst-case subset selection along with optimization of model parameters. To solve the optimization problem we designed a framework of efficient algorithms, which admits approximation guarantees for convex and Polyak-Lojasiewicz loss functions. Finally, our experiments showed that ROGET achieves better overall accuracy as compared to several state-of-the-art defense methods across several subset selection strategies. Our work opens several avenues of future research. We can extend our work to a slightly different setting where each instance has some importance score assigned to it. Another extension is to design a differentiable method for computing the worst-case attacked set, instead of using a greedy selection algorithm.

## 6 ETHICS STATEMENT

Our work tries to help ML models in achieving a better trade-off between robustness against adversarial attacks and performance on unperturbed/clean instances. Due to the vulnerability of ML models against adversarial perturbations, they have not been widely used in high-stake real world scenarios. Furthermore, the defense methods proposed so far to achieve robustness against attacks, face a considerable drop in accuracy on clean (unperturbed) instances. Here, our framework takes a step towards improving the performance on clean instances, while being robust against attacks on any subset of the dataset.

On the flip side, our method discusses a different type of adversarial attack, where the attacks are made on a subset of instances. If one uses such attacks in practice, the attacked ML systems can become vulnerable. However, these systems can use the defense method proposed in this paper, which can provide notable defense irrespective of the subset selection strategy of the adversary. The capability of our method to achieve good performance without being aware of adversary's strategy makes it suitable to be applied in a wide range of applications.

## 7 REPRODUCIBILITY STATEMENT

We have provided our code in supplementary material and provide the implementation details in Appendix G.

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

# Robust Training through Adversarially Selected Data Subsets (Appendix)

## A  FUTURE WORK

Our method selects a worst-case subset and trains the model parameters in an end-to-end manner to output a robust defense model. However, it might be more interesting to take into account an importance score for each instance to decide which instances need more protection against attacks. Additionally, it will be interesting to design a completely differentiable training method for computing the worst-case subset of attacked instances, instead of using a greedy selection algorithm along with gradient descent.

## B  RELATED WORK

**Adversarial attacks.** The attack methods (Szegedy et al., 2014; Kurakin et al., 2018; Madry et al., 2017; Goodfellow et al., 2015; Kurakin et al., 2017; Carlini & Wagner, 2017) discussed in the main paper can be broadly classified in three different settings i) white-box attacks, ii) black-box attacks, and iii) transfer-based attacks. In white-box attacks, the attacker assumes full knowledge about the defense model, and hence can access the gradients of the defense model. This led to the design of several gradient-based attacks (Szegedy et al., 2014; Madry et al., 2017; Goodfellow et al., 2015). Specifically, Szegedy et al. (2014) use constrained optimization, Carlini & Wagner (2017) use special regularization in the optimization. However, in black-box setting the attacker can only *query* the defense model being attacked (Ilyas et al., 2018). This led to the design of score-based attacks (Chen et al., 2017) which use the scores output by the defense model to approximate gradients, and craft adversarial perturbations. Additionally, to tackle this setting, researchers came up with transfer-based attacks (Papernot et al., 2017; 2016; Liu et al., 2017) where the attacker computes adversarial examples on a completely different model, in order to attack the defense model.

**Defense methods.** Apart from Adversarial training methods (Wang et al., 2019; Zhang et al., 2019b; Robey et al., 2021; Shafahi et al., 2019; Zhang et al., 2019a; Kim et al., 2021; Andriushchenko & Flammarion, 2020) already discussed in the main paper, there are several other approaches to designing robust defense models. One of them leverages randomness to defend against adversarial examples (Xie et al., 2018; Cohen et al., 2019; Liu et al., 2018). These works introduce randomness either in the inputs or in the model parameters which makes the model robust against crafted-instances. Yet another line of work (Ilyas et al., 2017; Meng & Chen, 2017; Samangouei et al., 2018) tries to use generative models to first project the input using an encoder, and then use the output of the encoder for classification. The projection step is claimed to reduce the impact of adversarial perturbations. Another popular approach (Gong et al., 2017; Metzen et al., 2017; Li & Li, 2017; Zheng & Hong, 2018) is to detect the presence of adversarial perturbations, before attempting to classify the input. The idea is motivated by the assumption that adversarial perturbed instances follow a different distribution as compared to natural (unperturbed) instances. Recent works have considered assigning different weights to different classes based on the underlying loss to ensure robustness against adversarial perturbation (Anonymous, 2022; Tian et al., 2021; Wang et al., 2021; Leavitt & Morcos, 2020). Tian et al. (2021) pointed out that accuracy across different classes significantly vary during adversarial training. Leavitt & Morcos (2020) showed that increasing class selectivity improves worst case perturbation while decreasing class selectivity improves average case perturbation. Wang et al. (2021) proposed Separable Reweighted Adversarial Training (SRAT) assign weights to different instances to learn separable features for imbalanced dataset. Recent work (Anonymous, 2022) develop adversarial training approaches that attempt to control the worst possible loss across different classes.

**Data subset selection.** There is a rich literature of work proposed for data subset selection in conjunction with model training (Campbell & Broderick, 2018; Lucic et al., 2017; Durga et al., 2021; Killamsetty et al., 2021; De et al., 2020; 2021). As mentioned in the main paper, these existing works focus more towards efficient learning (Durga et al., 2021; Mirzasoleiman et al., 2020), human-centric learning (De et al., 2020; 2021) and active learning (Wei et al., 2015; Kaushal et al., 2019). As a results these works operate in a completely different setting, and consequently require different solution techniques. For example, De et al. (2020) employ data subset selection in order to distribute instances between human and machine to generate semi-automated ML models. Also, the works

in (Durga et al., 2021; Mirzasoleiman et al., 2020) uses subset selection to reduce training time of ML models by reducing the size of the effective training dataset.

## C ILLUSTRATION OF OUR ADVERSARIAL TRAINING SETUP AND THE HARDNESS ANALYSIS

**Probabilistic generation of $S$.** Let us assume that the adversary follows a probabilistic strategy, *i.e.*, $\pi(\boldsymbol{x}_i, y_i) = P(i \in S)$. Thus $\pi(\boldsymbol{x}_i, y_i)$ indicates the probability that the instance $\boldsymbol{x}_i$ is chosen for attack. Here, one may wish to minimize the following loss:

$$\min_{\boldsymbol{\theta}} \max_{\{\pi(\boldsymbol{x}_i, y_i) \,|\, i \in D\}} \sum_{i \in D} \left[ \pi(\boldsymbol{x}_i, y_i) \ell(h_{\boldsymbol{\theta}}(a_{\boldsymbol{\phi}}(\boldsymbol{x}_i)), y_i) + (1 - \pi(\boldsymbol{x}_i, y_i)) \rho \, \ell(h_{\boldsymbol{\theta}}(\boldsymbol{x}_i), y_i) \right]. \quad (9)$$

$$\text{such that, } \sum_{i \in D} \pi(\boldsymbol{x}_i, y_i) \le b, \pi(\boldsymbol{x}_i, y_i) \in [0, 1]. \quad (10)$$

The inner optimization problem is a linear optimization problem in each $\pi(\boldsymbol{x}_i, y_i)$, so $\pi(\boldsymbol{x}_i, y_i) \in \{0, 1\}$. Then, if we define $S = \{i \,|\, \pi(\boldsymbol{x}_i, y_i) = 1\}$,

$$\min_{\boldsymbol{\theta}} \max_{S:|S| \le b} \left[ \sum_{i \in S} \ell(h_{\boldsymbol{\theta}}(a_{\boldsymbol{\phi}}(\boldsymbol{x}_i)), y_i) + \sum_{j \in D \setminus S} \rho \, \ell(h_{\boldsymbol{\theta}}(\boldsymbol{x}_j), y_j) \right] \quad (11)$$

**Hardness analysis.** Let us consider a specific instance of the problem where $y \in \{0, +1, -1\}$ and $h_{\boldsymbol{\theta}}$ is a fixed function independent of $\boldsymbol{\theta}$ given as

$$h_{\boldsymbol{\theta}}(\boldsymbol{x}) = \begin{cases} \Lambda e^{(1 - y \cdot \boldsymbol{\theta}_0^T \boldsymbol{x})^2} & \text{if, } y \in \{+1, -1\} \\ 1 - \sum_{y \in \{-1, +1\}} \Lambda e^{(1 - y \cdot \boldsymbol{\theta}_0^T \boldsymbol{x})^2} & \text{if, } y = 0 \end{cases} \quad (12)$$

where, $\Lambda << 1$, $\boldsymbol{\theta}_0$ is a constant vector, $\boldsymbol{x}$ is bounded such that $1 - \sum_{y \in \{-1, +1\}} \Lambda e^{(1 - y \cdot \boldsymbol{\theta}_0^T \boldsymbol{x})^2} > 0$. Similar (not exact) distributions were also used for instantiating anecdotal examples in (Zhang et al., 2021a). Let us further assume that the training set $D$ consists of only instances with $y \in \{+1, -1\}$ and no instance with $y = 0$. Additionally, let $a_{\boldsymbol{\phi}}(\boldsymbol{x}) = \boldsymbol{\phi} \odot \boldsymbol{x}$, where $\odot$ denotes element-wise multiplication operation. Here, $\boldsymbol{\phi}$ is restricted to a set, such that $1 - \sum_{y \in \{-1, +1\}} \Lambda e^{(1 - y \cdot \boldsymbol{\theta}_0^T a_{\boldsymbol{\phi}}(\boldsymbol{x}))^2} > 0$. Also, consider that $\rho = 0$ and $C = 0$ in this specific setting. Since $h_{\boldsymbol{\theta}}$ is independent of $\boldsymbol{\theta}$, the optimization problem (2) in this setting reduces to

$$\max_{\boldsymbol{\phi}, S:|S| \le b} \sum_{i \in S} \ell(h_{\boldsymbol{\theta}}(a_{\boldsymbol{\phi}}(\boldsymbol{x}_i)), y_i) \quad (13)$$

$$= \max_{\boldsymbol{\phi}, S:|S| \le b} \sum_{i \in S} -\log(\Lambda e^{(1 - y_i \cdot \boldsymbol{\theta}_0^T a_{\boldsymbol{\phi}}(\boldsymbol{x}_i))^2}) \quad (14)$$

$$= -\min_{\boldsymbol{\phi}, S:|S| \le b} \sum_{i \in S} \underbrace{\log(\Lambda) + (1 - y_i \cdot \boldsymbol{\theta}_0^T a_{\boldsymbol{\phi}}(\boldsymbol{x}_i))^2}_{\le 0} \quad (15)$$

$$= -\min_{\boldsymbol{\phi}, S:|S|=b} \sum_{i \in S} \log(\Lambda) + (1 - y_i \cdot \boldsymbol{\theta}_0^T a_{\boldsymbol{\phi}}(\boldsymbol{x}_i))^2 \quad (16)$$

$$= -\min_{\boldsymbol{\phi}, S:|S|=b} b \log(\Lambda) + \sum_{i \in S} (1 - y_i \cdot \boldsymbol{\theta}_0^T a_{\boldsymbol{\phi}}(\boldsymbol{x}_i))^2 \quad (17)$$

Thus, the solution of above optimization (17) is equal to the following optimization problem:

$$\min_{\boldsymbol{\phi}, S:|S|=b} \sum_{i \in S} (1 - y_i \cdot \boldsymbol{\theta}_0^T a_{\boldsymbol{\phi}}(\boldsymbol{x}_i))^2 \quad (18)$$

which is equivalent to the following optimization problem:

$$\min_{\boldsymbol{\phi}, S:|S|=b} \sum_{i \in S} (1 - y_i \cdot \boldsymbol{\phi}^T (\boldsymbol{\theta}_0 \odot \boldsymbol{x}_i))^2 \quad (19)$$

The next steps follow directly from ((De et al., 2021), Proof of Theorem 1). We describe the steps below to make the proof self-contained. Assume that $\boldsymbol{X} = [(\boldsymbol{\theta}_0 \odot \boldsymbol{x}_1)^T; \dots; (\boldsymbol{\theta}_0 \odot \boldsymbol{x}_{|D|})^T]$ has full row rank $|D|$. Now, let $\boldsymbol{y} = [y_1, \dots, y_i, \dots, y_{|D|}]$, $\boldsymbol{r} \in \mathbb{R}^{|D|}$ be an arbitrary vector of real

numbers and $\boldsymbol{X_R}^{-1}$ be the right inverse of $\boldsymbol{X}$ (exists because $\boldsymbol{X}$ has full row rank). By definition, $(\boldsymbol{X_R}^{-1})^T(\boldsymbol{\theta}_0 \odot \boldsymbol{x}_i) = \boldsymbol{e}_i^T$, where $\boldsymbol{e}_i$ is a column vector with entry 1 at position $i$, and 0 elsewhere. Further, let $\boldsymbol{\phi}' = \boldsymbol{\phi} - \boldsymbol{X_R}^{-1}(\boldsymbol{y} - \boldsymbol{r})$. We rewrite the objective of the optimization (19) in terms of $\boldsymbol{\phi}'$ to obtain

$$\sum_{i \in S}(1 - y_i \cdot (\boldsymbol{\phi}' + \boldsymbol{X_R}^{-1}(\boldsymbol{y} - \boldsymbol{r}))^T(\boldsymbol{\theta}_0 \odot \boldsymbol{x}_i))^2 \tag{20}$$

$$= \sum_{i \in S}(1 - y_i \cdot \boldsymbol{\phi}'^T(\boldsymbol{\theta}_0 \odot \boldsymbol{x}_i) - y_i \cdot (\boldsymbol{y} - \boldsymbol{r})^T \cdot \boldsymbol{e}_i)^2 \tag{21}$$

$$= \sum_{i \in S}(1 - y_i \cdot \boldsymbol{\phi}'^T(\boldsymbol{\theta}_0 \odot \boldsymbol{x}_i) - \underbrace{y_i^2}_{=1} + y_i \cdot r_i)^2 \tag{22}$$

$$= \sum_{i \in S}(r_i - \boldsymbol{\phi}'^T(\boldsymbol{\theta}_0 \odot \boldsymbol{x}_i))^2 \tag{23}$$

Hence, our optimization problem reduces to the following optimization problem:

$$\min_{\boldsymbol{\phi}', S:|S|=b} \sum_{i \in S}(r_i - \boldsymbol{\phi}'^T(\boldsymbol{\theta}_0 \odot \boldsymbol{x}_i))^2 \tag{24}$$

Since the above optimization (24) is known to be NP-hard (Bhatia et al., 2017), we have successfully proven that our optimization problem 2 is also NP-hard.

# D EXPLANATION OF THE ASSUMPTIONS

In this section, we justify the assumptions mentioned in Sections 3.1 and 3.2.

## ASSUMPTIONS 1

**(1, 5) Lipschitz continuity and boundedness of the parameters:** We think that Lipschitz continuity of loss and models are very prevalent. In most practical and well behaved ML algorithms, these assumptions hold. For any differentiable networks that have bounded parameters and no singularity, we usually have these conditions. If we do not have such conditions, gradients may blow up during training.

**(2) Stability:** Algorithmic stability is also a desirable property in ML. It ensures that if we make changes in *one* instance, the parameters do not change. L2 Regularization, drop-out and even SGD algorithm itself encourages stability.

**(3, 4) Metric property and norm boundedness of cost of perturbation:** Most existing works have used $L_\infty$ distance as the cost of perturbation. Our method considers a general set of metrics which is not limited to $L_\bullet$ distances. $L_\bullet$ space norms are bounded by each other by a factor, i.e., $k_1 L_c < L_a < k_2 L_b$. We bounded our metric norm by the L2 norm for standardization of analysis. We do not foresee the deviation of such a condition in a practical scenario.

## ASSUMPTIONS 2

**(1) L-smoothness of F:** This just ensures that gradients of F are Lipschitz too. Indeed this is a bit stronger condition than Lipschitzness of F itself. A wide variety of smooth activation functions like Linear, Sigmoid are L-smooth. Even the discontinuous functions like ReLU are often L-smooth almost everywhere.

**(2,3) Boundedness of F, h and $\ell$:** As long as there is no inherent singularity in the interior of these functions (e.g., unlike 1/(x-a)), this is a redundant condition, given the boundedness of $\theta$. We simply keep it to make our notations simple during analysis. But this condition does not put any additional restriction, if the underlying function does not have a singularity in the interior.

**(4) Size of B:** This is a parameter used in our algorithm— it does not put any restriction of the underlying setup, neither the loss function nor the models.

**(5) Adversarial network always perturbs a feature:** This imposes the notion of a very strong adversary. Even if we deviate, the theoretical bounds would be stronger. We keep this condition for the sake of brevity.

# E  PROOFS OF THE TECHNICAL RESULTS IN SECTION 3

## E.1  MONOTONICITY AND $\gamma$-WEAK SUBMODULARITY OF $G_\lambda$: THEOREM 2

**Theorem 2.** *Given Assumption 1 let there be a value of minimum non-zero constant $\lambda_{\min} > 0$ such that $\ell^* = \min_{\boldsymbol{x} \in \mathcal{X}, y \in \mathcal{Y}} \min_{\boldsymbol{\theta}} [\lambda_{\min} R(\boldsymbol{\theta}) + \ell(h_{\boldsymbol{\theta}}(\boldsymbol{x}), y) - 2q\mu\beta L_\phi] > 0$ where $q$ and $\beta$ are given in Assumption 1. Then, for $\lambda > \lambda_{\min}$, we have the following statements:*

*(1) $G_\lambda(S)$ is monotone in $S$*

*(2) $G_\lambda(S)$ $\gamma$-weakly submodular with $\gamma > \gamma^* = \ell^*[\ell^* + 2L_h L_x L_\phi \phi_{\max} + 3q\mu\beta L_\phi]^{-1}$.*

**Proof sketch.** To prove monotonicity, we first show that $G_\lambda(k \,|\, S) > \ell(h_{\boldsymbol{\theta}}(a_{\boldsymbol{\phi}^*(\boldsymbol{\theta}, S)}(\boldsymbol{x}_k)), y_k) + \sum_{i \in S} [\rho\, C(a_{\boldsymbol{\phi}^*(\boldsymbol{\theta}, S \cup k)}(\boldsymbol{x}_i), \boldsymbol{x}_i) - \rho\, C(a_{\boldsymbol{\phi}^*(\boldsymbol{\theta}, S)}(\boldsymbol{x}_i), \boldsymbol{x}_i)] + \lambda R(\boldsymbol{\theta})$ which is more than $\lambda R(\boldsymbol{\theta}) + \ell(h_{\boldsymbol{\theta}}(a_{\boldsymbol{\phi}^*(\boldsymbol{\theta}, S)}(\boldsymbol{x}_k)), y_k) - \beta\rho|S| \|\boldsymbol{\phi}^*(\boldsymbol{\theta}, S \cup k) - \boldsymbol{\phi}^*(\boldsymbol{\theta}, S)\|$. Next, we use Assumptions 1 to show that this quantity is more than $\ell^*$.

To prove $\gamma$-weak submodularity, we first show that $G_\lambda(k \,|\, T) \leq \ell^* + 2\phi_{\max} + 3q\mu\beta L_\phi$. To that aim, we show that $G_\lambda(k \,|\, T) \leq \lambda R(\boldsymbol{\theta}) + \ell(h_{\boldsymbol{\theta}}(a_{\boldsymbol{\phi}^*(\boldsymbol{\theta}, T \cup k)}(\boldsymbol{x}_k)), y_k) + q\mu\beta \leq \lambda R(\boldsymbol{\theta}) + \ell(h_{\boldsymbol{\theta}}(a_{\boldsymbol{\phi}^*(\boldsymbol{\theta}, S)}(\boldsymbol{x}_k)), y_k) + |\ell(h_{\boldsymbol{\theta}}(a_{\boldsymbol{\phi}^*(\boldsymbol{\theta}, T \cup k)}(\boldsymbol{x}_k)), y_k) - \ell(h_{\boldsymbol{\theta}}(a_{\boldsymbol{\phi}^*(\boldsymbol{\theta}, S)}(\boldsymbol{x}_k)), y_k)| + q\mu\beta$. Next we use the Lipschitzness of different functions to show that it is less than $\ell^* + 2L_h L_x L_\phi \phi_{\max} + 3q\mu\beta L_\phi$. This, together with the fact that $G_\lambda(k \,|\, S) > \ell^*$ derived during the proof of monotonicity, results in the inequality $G_\lambda(k \,|\, S)/G_\lambda(k \,|\, T) \geq \ell^*[\ell^* + 2L_h L_x L_\phi \phi_{\max} + 3q\mu\beta L_\phi]^{-1}$. Finally, we use the result of Proposition 6 (in Appendix F) to complete the proof.

*Proof.* **Monotonicity of $G_\lambda$.** Let $S \subset D$ and let $k \in D \setminus S$. For any $\boldsymbol{\theta} \in \Theta$,
$$G_\lambda(\boldsymbol{\theta}, S \cup k) - G_\lambda(\boldsymbol{\theta}, S)$$

$$= \frac{1}{|D|} \left[ \sum_{i \in S \cup k} \lambda R(\boldsymbol{\theta}) + \ell(h_{\boldsymbol{\theta}}(a_{\boldsymbol{\phi}^*(\boldsymbol{\theta}, S \cup k)}(\boldsymbol{x}_i)), y_i) \right] - \frac{1}{|D|} \left[ \sum_{i \in S} \lambda R(\boldsymbol{\theta}) + \ell(h_{\boldsymbol{\theta}}(a_{\boldsymbol{\phi}^*(\boldsymbol{\theta}, S)}(\boldsymbol{x}_i)), y_i) \right]$$

$$= \frac{1}{|D|} \left[ \lambda R(\boldsymbol{\theta}) + \sum_{i \in S \cup k} \ell(h_{\boldsymbol{\theta}}(a_{\boldsymbol{\phi}^*(\boldsymbol{\theta}, S \cup k)}(\boldsymbol{x}_i)), y_i) - \sum_{i \in S} \ell(h_{\boldsymbol{\theta}}(a_{\boldsymbol{\phi}^*(\boldsymbol{\theta}, S)}(\boldsymbol{x}_i)), y_i) \right] \tag{25}$$

We now derive a bound on $\sum_{i \in S \cup k} \ell(h_{\boldsymbol{\theta}}(a_{\boldsymbol{\phi}^*(\boldsymbol{\theta}, S \cup k)}(\boldsymbol{x}_i)), y_i) - \sum_{i \in S} \ell(h_{\boldsymbol{\theta}}(a_{\boldsymbol{\phi}^*(\boldsymbol{\theta}, S)}(\boldsymbol{x}_i)), y_i)$. Using the definition of $a_{\boldsymbol{\phi}^*(S \cup k)}$ from Eq. (3), we obtain:

$$\sum_{i \in S \cup k} [\ell(h_{\boldsymbol{\theta}}(a_{\boldsymbol{\phi}^*(\boldsymbol{\theta}, S \cup k)}(\boldsymbol{x}_i)), y_i) - \mu C(a_{\boldsymbol{\phi}^*(\boldsymbol{\theta}, S \cup k)}(\boldsymbol{x}_i), \boldsymbol{x}_i)]$$

$$\geq \sum_{i \in S \cup k} [\ell(h_{\boldsymbol{\theta}}(a_{\boldsymbol{\phi}^*(\boldsymbol{\theta}, S)}(\boldsymbol{x}_i)), y_i) - \mu C(a_{\boldsymbol{\phi}^*(\boldsymbol{\theta}, S)}(\boldsymbol{x}_i), \boldsymbol{x}_i)] \tag{26}$$

$$\implies \sum_{i \in S \cup k} [\ell(h_{\boldsymbol{\theta}}(a_{\boldsymbol{\phi}^*(\boldsymbol{\theta}, S \cup k)}(\boldsymbol{x}_i)), y_i) - \ell(h_{\boldsymbol{\theta}}(a_{\boldsymbol{\phi}^*(\boldsymbol{\theta}, S)}(\boldsymbol{x}_i)), y_i)]$$

$$\geq \mu \sum_{i \in S \cup k} [C(a_{\boldsymbol{\phi}^*(\boldsymbol{\theta}, S \cup k)}(\boldsymbol{x}_i), \boldsymbol{x}_i) - C(a_{\boldsymbol{\phi}^*(\boldsymbol{\theta}, S)}(\boldsymbol{x}_i), \boldsymbol{x}_i)] \tag{27}$$

Substituting inequality (27) in Eq. (25), we obtain:

$$G_\lambda(\boldsymbol{\theta}, S \cup k) - G_\lambda(\boldsymbol{\theta}, S) \geq \frac{1}{|D|} \left[ \lambda R(\boldsymbol{\theta}) + \ell(h_{\boldsymbol{\theta}}(a_{\boldsymbol{\phi}^*(\boldsymbol{\theta}, S)}(\boldsymbol{x}_k)), y_k) \right. \tag{28}$$

$$\left. + \sum_{i \in S \cup k} \mu[C(a_{\boldsymbol{\phi}^*(\boldsymbol{\theta}, S \cup k)}(\boldsymbol{x}_i), \boldsymbol{x}_i) - C(a_{\boldsymbol{\phi}^*(\boldsymbol{\theta}, S)}(\boldsymbol{x}_i), \boldsymbol{x}_i)] \right] \tag{29}$$

$$= \frac{1}{|D|} \left[ \lambda R(\boldsymbol{\theta}) + \ell(h_{\boldsymbol{\theta}}(a_{\boldsymbol{\phi}^*(\boldsymbol{\theta}, S)}(\boldsymbol{x}_k)), y_k) \right.$$

$$\left. - \sum_{i \in S \cup k} \mu[C(a_{\boldsymbol{\phi}^*(\boldsymbol{\theta}, S)}(\boldsymbol{x}_i), \boldsymbol{x}_i)] - C(a_{\boldsymbol{\phi}^*(\boldsymbol{\theta}, S \cup k)}(\boldsymbol{x}_i), \boldsymbol{x}_i) \right] \tag{30}$$

$$\geq \frac{1}{|D|}\Bigg[\lambda R(\boldsymbol{\theta}) + \ell(h_{\boldsymbol{\theta}}(a_{\boldsymbol{\phi}^*(\boldsymbol{\theta},S)}(\boldsymbol{x}_k)), y_k)$$

$$- \mu \sum_{i \in S \cup k} C(a_{\boldsymbol{\phi}^*(\boldsymbol{\theta},S \cup k)}(\boldsymbol{x}_i), a_{\boldsymbol{\phi}^*(\boldsymbol{\theta},S)}(\boldsymbol{x}_i))\Bigg] \tag{31}$$

$$\geq \frac{1}{|D|}\Bigg[\lambda R(\boldsymbol{\theta}) + \ell(h_{\boldsymbol{\theta}}(a_{\boldsymbol{\phi}^*(\boldsymbol{\theta},S)}(\boldsymbol{x}_k)), y_k) - q\mu\beta L_{\phi}\frac{|S \cup k|}{|S|}\Bigg] \tag{32}$$

$$\geq \frac{1}{|D|}[\lambda R(\boldsymbol{\theta}) + \ell(h_{\boldsymbol{\theta}}(a_{\boldsymbol{\phi}^*(\boldsymbol{\theta},S)}(\boldsymbol{x}_k)), y_k) - 2q\mu\beta L_{\phi}] \tag{33}$$

Here, inequality (31) follows from using triangle inequality of Assumption 1 and inequality (32) follows from the stability assumption in Assumption 1. We use the assumption on $\ell^*$ to conclude that the right hand side of inequality (33) is non-negative. This shows that $G_{\lambda}$ is monotone in $S$.

$\gamma$-**weak submodularity of** $G_{\lambda}$**.** We first provide an upper bound of $G_{\lambda}(\boldsymbol{\theta}, T \cup k) - G_{\lambda}(\boldsymbol{\theta}, T)$.

$$G_{\lambda}(\boldsymbol{\theta}, T \cup k) - G_{\lambda}(\boldsymbol{\theta}, T)$$

$$= \frac{1}{|D|}\Bigg[\sum_{i \in T \cup k}\lambda R(\boldsymbol{\theta}) + \ell(h_{\boldsymbol{\theta}}(a_{\boldsymbol{\phi}^*(\boldsymbol{\theta},T \cup k)}(\boldsymbol{x}_i)), y_i)\Bigg] - \frac{1}{|D|}\Bigg[\sum_{i \in T}\lambda R(\boldsymbol{\theta}) + \ell(h_{\boldsymbol{\theta}}(a_{\boldsymbol{\phi}^*(\boldsymbol{\theta},T)}(\boldsymbol{x}_i)), y_i)\Bigg]$$

$$= \frac{1}{|D|}\Bigg[\lambda R(\boldsymbol{\theta}) + \sum_{i \in T \cup k}\ell(h_{\boldsymbol{\theta}}(a_{\boldsymbol{\phi}^*(\boldsymbol{\theta},T \cup k)}(\boldsymbol{x}_i)), y_i) - \sum_{i \in T}\ell(h_{\boldsymbol{\theta}}(a_{\boldsymbol{\phi}^*(\boldsymbol{\theta},T)}(\boldsymbol{x}_i)), y_i)\Bigg] \tag{34}$$

$$= \frac{1}{|D|}\Bigg[\lambda R(\boldsymbol{\theta}) + \sum_{i \in T}\big(\ell(h_{\boldsymbol{\theta}}(a_{\boldsymbol{\phi}^*(\boldsymbol{\theta},T \cup k)}(\boldsymbol{x}_i)), y_i) - \mu C(a_{\boldsymbol{\phi}^*(\boldsymbol{\theta},T \cup k)}(\boldsymbol{x}_i), \boldsymbol{x}_i)\big)$$

$$- \sum_{i \in T}\big(\ell(h_{\boldsymbol{\theta}}(a_{\boldsymbol{\phi}^*(\boldsymbol{\theta},T)}(\boldsymbol{x}_i)), y_i) - \mu C(a_{\boldsymbol{\phi}^*(\boldsymbol{\theta},T)}(\boldsymbol{x}_i), \boldsymbol{x}_i)\big)$$

$$+ \ell(h_{\boldsymbol{\theta}}(a_{\boldsymbol{\phi}^*(\boldsymbol{\theta},T \cup k)}(\boldsymbol{x}_k)), y_k) + \mu \sum_{i \in T}\big(C(a_{\boldsymbol{\phi}^*(\boldsymbol{\theta},T \cup k)}(\boldsymbol{x}_i), \boldsymbol{x}_i) - C(a_{\boldsymbol{\phi}^*(\boldsymbol{\theta},T)}(\boldsymbol{x}_i), \boldsymbol{x}_i)\big)\Bigg] \tag{35}$$

$$\leq \frac{1}{|D|}\Bigg[\lambda R(\boldsymbol{\theta}) + \ell(h_{\boldsymbol{\theta}}(a_{\boldsymbol{\phi}^*(\boldsymbol{\theta},T \cup k)}(\boldsymbol{x}_k)), y_k) + q\mu\beta L_{\phi}\Bigg] \tag{36}$$

Here, the last inequality is due to the fact that:

$$\sum_{i \in T}\big(\ell(h_{\boldsymbol{\theta}}(a_{\boldsymbol{\phi}^*(\boldsymbol{\theta},T \cup k)}(\boldsymbol{x}_i)), y_i) - \mu C(a_{\boldsymbol{\phi}^*(\boldsymbol{\theta},T \cup k)}(\boldsymbol{x}_i), \boldsymbol{x}_i)\big) \tag{37}$$

$$- \sum_{i \in T}\big(\ell(h_{\boldsymbol{\theta}}(a_{\boldsymbol{\phi}^*(\boldsymbol{\theta},T)}(\boldsymbol{x}_i)), y_i) - \mu C(a_{\boldsymbol{\phi}^*(\boldsymbol{\theta},T)}(\boldsymbol{x}_i), \boldsymbol{x}_i)\big) \leq 0 \tag{38}$$

since $\boldsymbol{\phi}^*(\boldsymbol{\theta}, T)$ provides the maximum of the second term and the fact that:

$$\mu \sum_{i \in T}\big(C(a_{\boldsymbol{\phi}^*(\boldsymbol{\theta},T \cup k)}(\boldsymbol{x}_i), \boldsymbol{x}_i) - C(a_{\boldsymbol{\phi}^*(\boldsymbol{\theta},T)}(\boldsymbol{x}_i), \boldsymbol{x}_i)\big) \leq q\mu \sum_{i \in T}||a_{\boldsymbol{\phi}^*(\boldsymbol{\theta},T \cup k)}(\boldsymbol{x}_i) - a_{\boldsymbol{\phi}^*(\boldsymbol{\theta},T)}(\boldsymbol{x}_i)|| \tag{39}$$

$$\leq |T|q\mu L_{\phi}\frac{\beta}{|T|} \text{ (Stability of } \boldsymbol{\phi}) \tag{40}$$

**Lower bounding the ratio.** Using inequality (33) and inequality (36), we get the following

$$\frac{G_{\lambda}(\boldsymbol{\theta}, S \cup k) - G_{\lambda}(\boldsymbol{\theta}, S)}{G_{\lambda}(\boldsymbol{\theta}, T \cup k) - G_{\lambda}(\boldsymbol{\theta}, T)}$$

$$\geq \frac{\lambda R(\boldsymbol{\theta}) + \ell(h_{\boldsymbol{\theta}}(a_{\boldsymbol{\phi}^*(\boldsymbol{\theta},S)}(\boldsymbol{x}_k)), y_k) - 2q\mu\beta L_{\phi}}{\lambda R(\boldsymbol{\theta}) + \ell(h_{\boldsymbol{\theta}}(a_{\boldsymbol{\phi}^*(\boldsymbol{\theta},T \cup k)}(\boldsymbol{x}_k)), y_k) + q\mu\beta L_{\phi}}$$

$$\geq \frac{\lambda R(\boldsymbol{\theta}) + \ell(h_{\boldsymbol{\theta}}(a_{\boldsymbol{\phi}^*(\boldsymbol{\theta},S)}(\boldsymbol{x}_k)), y_k) - 2q\mu\beta L_{\phi}}{\lambda R(\boldsymbol{\theta}) + \ell(h_{\boldsymbol{\theta}}(a_{\boldsymbol{\phi}^*(\boldsymbol{\theta},S)}(\boldsymbol{x}_k)), y_k) + \ell(h_{\boldsymbol{\theta}}(a_{\boldsymbol{\phi}^*(\boldsymbol{\theta},T \cup k)}(\boldsymbol{x}_k)), y_k) - \ell(h_{\boldsymbol{\theta}}(a_{\boldsymbol{\phi}^*(\boldsymbol{\theta},S)}(\boldsymbol{x}_k)), y_k) + q\mu\beta L_{\phi}}$$

$$\geq \frac{\ell^*}{\ell^* + 2L_h L_x L_{\phi}\phi_{\max} + 3q\mu\beta L_{\phi}} = \gamma^* \tag{41}$$

Above inequality (41) follows from Lipschitz continuity assumption, and upper boundedness of $\Phi$ mentioned in Assumptions 1. This together with Proposition 6 gives us the result. $\qquad\square$

### E.2 APPROXIMATION GUARANTEES

**Theorem 4.** *Given the conditions of Theorem 2 and Assumptions 2, let $F(h_{\boldsymbol{\theta}}, S \,|\, a_{\boldsymbol{\phi}^*(\boldsymbol{\theta},S)})$ be a PL function in $\boldsymbol{\theta}$ for all $S$, i.e., $\|\nabla_{\boldsymbol{\theta}} F(h_{\boldsymbol{\theta}}, S \,|\, a_{\boldsymbol{\phi}^*(\boldsymbol{\theta},S)})\| \geq \sigma[F(h_{\boldsymbol{\theta}}, S \,|\, a_{\boldsymbol{\phi}^*(\boldsymbol{\theta},S)}) - \min_{\boldsymbol{\theta}'} F(h_{\boldsymbol{\theta}'}, S \,|\, a_{\boldsymbol{\phi}^*(\boldsymbol{\theta}',S)})]$ for some constant $\sigma$. If we set the learning rate $\eta = 1/kT$ in Algorithm 1, then for $T = O(1/k\epsilon)$ iterations, $\delta < (1 - e^{-\gamma^*})$ and $\rho < [C_{\min}((e^{-\gamma^*} + \delta)^{-1} - 1) - \lambda\theta_{\max}^2]/\ell_{\max}$, we have the following approximation guarantee for Algorithm 1:*

$$\min_t \mathbb{E}[F(h_{\widehat{\boldsymbol{\theta}}_t}, \widehat{S}_T \,|\, a_{\boldsymbol{\phi}^*(\widehat{\boldsymbol{\theta}}_t, \widehat{S}_T)})] \leq \max_S \min_t \mathbb{E}[F(h_{\widehat{\boldsymbol{\theta}}_t}, S \,|\, a_{\boldsymbol{\phi}^*(\widehat{\boldsymbol{\theta}}_t, S)})]$$

$$\leq \left[1 - (e^{-\gamma^*} + \delta)/\kappa\right]^{-1}\left(OPT + 2L^2 h_{\max}/\sigma + \epsilon\right) \quad (42)$$

*where $\boldsymbol{\theta}_t$ is the iterate in Line 3 in Algorithm 1, $OPT$ is the value at the optimum solution of our adversarial training problem (2), $\kappa = C_{\min}/(\lambda\theta_{\max}^2 + \rho\ell_{\max} + C_{\min})$. Here $\theta_{\max}$ is defined in Assumption 1 and $\ell_{\max}, C_{\min}$ are defined in Assumptions 2.*

*Proof.* Under the assumptions, Lemmas 13 and 14 hold. Using Lemma 13, we have that for all $S$,

$$\left(1 - \frac{e^{-\gamma^*} + \delta}{\kappa}\right) F(h_{\widehat{\boldsymbol{\theta}}_t}, S \,|\, a_{\boldsymbol{\phi}^*(\widehat{\boldsymbol{\theta}}_t, S)}) - \mathbb{E}[F(h_{\widehat{\boldsymbol{\theta}}_t}, \widehat{S}_t \,|\, a_{\boldsymbol{\phi}^*(\widehat{\boldsymbol{\theta}}_t, \widehat{S}_t)})] \leq 4k\eta\nabla_{\max}^2 \quad (43)$$

where the expectation above is taken over the randomness in stochastic distorted greedy algorithm. Now, taking expectation over the randomness in $k$-SGD gives us,

$$\left(1 - \frac{e^{-\gamma^*} + \delta}{\kappa}\right) \mathbb{E}[F(h_{\widehat{\boldsymbol{\theta}}_t}, S \,|\, a_{\boldsymbol{\phi}^*(\widehat{\boldsymbol{\theta}}_t, S)})] - \mathbb{E}[F(h_{\widehat{\boldsymbol{\theta}}_t}, \widehat{S}_t \,|\, a_{\boldsymbol{\phi}^*(\widehat{\boldsymbol{\theta}}_t, \widehat{S}_t)})] \leq 4k\eta\nabla_{\max}^2 \quad (44)$$

where the expectation is computed over all the total randomness of the algorithm. We now sum the above equation over all $T$ to get,

$$\sum_{t=1}^{T}\left[\left(1 - \frac{e^{-\gamma^*} + \delta}{\kappa}\right) \mathbb{E}[F(h_{\widehat{\boldsymbol{\theta}}_t}, S \,|\, a_{\boldsymbol{\phi}^*(\widehat{\boldsymbol{\theta}}_t, S)})] - \mathbb{E}[F(h_{\widehat{\boldsymbol{\theta}}_t}, \widehat{S}_t \,|\, a_{\boldsymbol{\phi}^*(\widehat{\boldsymbol{\theta}}_t, \widehat{S}_t)})]\right] \leq 4Tk\eta\nabla_{\max}^2 \quad (45)$$

Using Lemma 14, we have that for all $\boldsymbol{\theta}$,

$$\mathbb{E}[F(h_{\widehat{\boldsymbol{\theta}}_{t-1}}, \widehat{S}_{t-1} \,|\, a_{\boldsymbol{\phi}^*(\widehat{\boldsymbol{\theta}}_{t-1}, \widehat{S}_{t-1})})] - \mathbb{E}[F(h_{\boldsymbol{\theta}}, \widehat{S}_{t-1} \,|\, a_{\boldsymbol{\phi}^*(\boldsymbol{\theta}, \widehat{S}_{t-1})})]$$

$$\leq 2k\eta\nabla_{\max}^2 + \frac{4L\theta_{\max}^2(1 - \eta\sigma)^k}{\sigma} + \frac{L\eta\nabla_{\max}^2}{2\sigma} \quad (46)$$

where the expectation is w.r.t. the total randomness of the algorithm. Summing over all $T$, we obtain:

$$\sum_{t=1}^{T}\left[\mathbb{E}[F(h_{\widehat{\boldsymbol{\theta}}_t}, \widehat{S}_t \,|\, a_{\boldsymbol{\phi}^*(\widehat{\boldsymbol{\theta}}_t, \widehat{S}_t)})] - \mathbb{E}[F(h_{\boldsymbol{\theta}}, \widehat{S}_t \,|\, a_{\boldsymbol{\phi}^*(\boldsymbol{\theta}, \widehat{S}_t)})]\right]$$

$$\leq 2Tk\eta\nabla_{\max}^2 + \frac{4TL\theta_{\max}^2(1 - \eta\sigma)^k}{\sigma} + \frac{LT\eta\nabla_{\max}^2}{2\sigma} \quad (47)$$

Combining inequality (45) and inequality (47) and then dividing by $T$ throughout we obtain:

$$\left(1 - \frac{e^{-\gamma^*} + \delta}{\kappa}\right) \sum_{t=1}^{T}\left[\frac{\mathbb{E}[F(h_{\widehat{\boldsymbol{\theta}}_t}, S \,|\, a_{\boldsymbol{\phi}^*(\widehat{\boldsymbol{\theta}}_t, S)})]}{T}\right]$$

$$\leq \max_S \mathbb{E}[F(h_{\boldsymbol{\theta}}, S \,|\, a_{\boldsymbol{\phi}^*(\boldsymbol{\theta}, S)})] + 6k\eta\nabla_{\max}^2 + \frac{4L\theta_{\max}^2(1 - \eta\sigma)^k}{\sigma} + \frac{L\eta\nabla_{\max}^2}{2\sigma}$$

Taking min over $\boldsymbol{\theta}$ on both sides and noting that all terms except one are independent of $\boldsymbol{\theta}$, we get

$$\left(1 - \frac{e^{-\gamma^*} + \delta}{\kappa}\right) \sum_{t=1}^{T} \left[\frac{\mathbb{E}[F(h_{\widehat{\boldsymbol{\theta}}_t}, S \mid a_{\boldsymbol{\phi}^*(\widehat{\boldsymbol{\theta}}_t, S)})]}{T}\right]$$

$$\leq \min_{\boldsymbol{\theta}} \max_{S} \mathbb{E}[F(h_{\boldsymbol{\theta}}, S \mid a_{\boldsymbol{\phi}^*(\boldsymbol{\theta}, S)})] + 6k\eta\nabla_{\max}^2 + e^{-k\eta\sigma}\left[\frac{4L\theta_{\max}^2}{\sigma}\right] + \frac{L\eta\nabla_{\max}^2}{2\sigma} \quad (48)$$

Putting $\eta = \frac{1}{kT}$, we obtain:

$$\left(1 - \frac{e^{-\gamma^*} + \delta}{\kappa}\right) \sum_{t=1}^{T} \left[\frac{\mathbb{E}[F(h_{\widehat{\boldsymbol{\theta}}_t}, S \mid a_{\boldsymbol{\phi}^*(\widehat{\boldsymbol{\theta}}_t, S)})]}{T}\right] \leq OPT + e^{-\sigma/T}\left[\frac{4L\theta_{\max}^2}{\sigma}\right] + \frac{6\nabla_{\max}^2}{T} + \frac{L\nabla_{\max}^2}{2kT\sigma}$$

$$\sum_{t=1}^{T} \left[\frac{\mathbb{E}[F(h_{\widehat{\boldsymbol{\theta}}_t}, S \mid a_{\boldsymbol{\phi}^*(\widehat{\boldsymbol{\theta}}_t, S)})]}{T}\right] \leq \frac{OPT}{\left(1 - \frac{e^{-\gamma^*} + \delta}{\kappa}\right)} + \frac{e^{-\sigma/T}\left[\frac{4L\theta_{\max}^2}{\sigma}\right]}{\left(1 - \frac{e^{-\gamma^*} + \delta}{\kappa}\right)} + \frac{\epsilon}{\left(1 - \frac{e^{-\gamma^*} + \delta}{\kappa}\right)} \quad (49)$$

This gives us the statement in the proof of the theorem. $\qquad\square$

---

**Algorithm 2** ROGET Algorithm (with additional variants of gradient descent for convex $F$)

---

**Require:** Training instances $D$, regularization parameter $\lambda$, budget $b$, # of iterations $T$, learning rate $\eta$, METHOD $\in \{\text{GD}, \text{SGD}, \text{k}-\text{SGD}\}$

1: INIT($h_{\boldsymbol{\theta}}$), $\widehat{S}_0 \leftarrow \emptyset$
2: **for** $t = 0$ to $T-1$ **do**
3:    $\widehat{\boldsymbol{\theta}}_{t+1} \leftarrow \text{TRAIN}(\widehat{\boldsymbol{\theta}}_t, \widehat{S}_t, \eta_t, \text{METHOD})$
4:    $\widehat{S}_{t+1} \leftarrow \text{SDG}(G_\lambda, m_\lambda, \widehat{\boldsymbol{\theta}}_{t+1}, b)$.
5: $\widehat{\boldsymbol{\theta}} = \sum_{t=1}^{T} \eta_t \widehat{\boldsymbol{\theta}}_t / \sum_{t=1}^{T} \eta_t$
6: **return** $\widehat{\boldsymbol{\theta}}, \widehat{S}_T$

---

1: **procedure** SDG $(G_\lambda, m_\lambda, \widehat{\boldsymbol{\theta}}, b)$
2:    $S \leftarrow \emptyset$
3:    **for** $s \in [b]$ **do**
4:       $\gamma_{b,s} \leftarrow (1 - \gamma/b)^{b-s-1}$
5:       Randomly draw a subset $B$ from $D$
6:       $e^* = \underset{e \in B}{\text{argmax}}\ \gamma_{b,s} G_\lambda(e \mid S) - m_\lambda(\{e\})$
7:       **if** $\gamma_{b,s} G_\lambda(e^* \mid S) - m_\lambda(\{e^*\}) \geq 0$ **then**
8:          $S \leftarrow S \cup \{e\}$
9:    **return** $S$

---

1: **procedure** TRAIN($\widehat{\boldsymbol{\theta}}, S, \eta, \text{METHOD}$)

2:    //deterministic gradient descent
3:    **if** METHOD is GD **then**
4:       **return** $\widehat{\boldsymbol{\theta}} - \eta\, \partial F(h_{\boldsymbol{\theta}}, S \mid a_{\boldsymbol{\phi}^*(\boldsymbol{\theta},S)}) / \partial\boldsymbol{\theta}\big|_{\boldsymbol{\theta}=\widehat{\boldsymbol{\theta}}}$

5:    //stochastic gradient descent
6:    **if** METHOD is SGD **then**
7:       Draw $i \sim D$ uniformly at random
8:       **if** $i \in S$ **then**
9:          $F \leftarrow \ell(h_{\boldsymbol{\theta}}(a_{\boldsymbol{\phi}}(\boldsymbol{x}_i)), y_i)$
10:      **else**
11:          $F \leftarrow \rho\, \ell(h_{\boldsymbol{\theta}}(\boldsymbol{x}_i), y_i)$
12:      **return** $\widehat{\boldsymbol{\theta}} - \eta\, \frac{\partial F}{\partial\boldsymbol{\theta}}\big|_{\boldsymbol{\theta}=\widehat{\boldsymbol{\theta}}}$

---

**Approximation guarantee for convex $F$.** Next we consider the unlikely case when $F$ is convex. Here, we present the approximation guarantees of our algorithm, again copied here in Algorithm 2, specifically with two different variants of stochastic gradient descent, *viz.*, simple gradient descent (GD) and one-step stochastic gradient descent (SGD) (instead of $k$-SGD) that allows us to derive approximation guarantees for convexity. These guarantees generalize the results from Adibi et al. (2021).

**Theorem 5.** *Given the conditions of Theorem 2 and Assumption 2, let $F(h_{\boldsymbol{\theta}}, S \mid a_{\boldsymbol{\phi}^*(\boldsymbol{\theta},S)})$ be convex in $\boldsymbol{\theta}$ for all $S$, the learning rate $\eta = 1/\sqrt{T}$. Now, suppose we set either* method = GD *or* method = SGD *in line 3 in Algorithm 2, i.e., we use either one step gradient descent or stochastic gradient descent during training $\boldsymbol{\theta}$ for fixed $S$, then for $T = O(1/\epsilon^2)$ iterations, $\delta < (1 - e^{-\gamma^*})$ and $\rho > \lambda\theta_{\max}^2(1 + e^{\gamma^*})/(e^{\gamma^*}(1-\delta)C_\delta)\ \rho < [C_{\min}((e^{-\gamma^*} + \delta)^{-1} - 1) - \lambda\theta_{\max}^2]/\ell_{\max}$, we have the following approximation guarantee.*

$$\mathbb{E}[F(h_{\widehat{\boldsymbol{\theta}}}, \widehat{S}_T \mid a_{\boldsymbol{\phi}^*(\widehat{\boldsymbol{\theta}},\widehat{S}_T)})] \leq \max_S \mathbb{E}[F(h_{\widehat{\boldsymbol{\theta}}}, S \mid a_{\boldsymbol{\phi}^*(\widehat{\boldsymbol{\theta}},S)})] \leq \left[1 - (e^{-\gamma^*} + \delta)/\kappa\right]^{-1}(OPT + \epsilon) \tag{50}$$

*where $OPT$ is the value at the optimum solution of our adversarial training problem (2), $\kappa = C_{\min}/(\lambda\theta_{\max}^2 + \rho\ell_{\max} + C_{\min})$. Here, $\theta_{\max}$ is defined in Assumption 1 and $\ell_{\max}, C_{\min}$ are defined in Assumption 2.*

*Proof.* Under the Assumptions 2 and convexity of $F$, Lemma 11 holds. Hence, we have that $\boldsymbol{\theta} \in \Theta$,

$$\sum_{t=1}^{T} \left[\mathbb{E}[F(h_{\widehat{\boldsymbol{\theta}}_t}, \widehat{S}_t \mid a_{\boldsymbol{\phi}^*(\widehat{\boldsymbol{\theta}}_t,\widehat{S}_t)})] - \mathbb{E}[F(h_{\boldsymbol{\theta}}, \widehat{S}_t \mid a_{\boldsymbol{\phi}^*(\boldsymbol{\theta},\widehat{S}_t)})]\right] \leq 2T\eta\nabla_{\max}^2 + \frac{2\theta_{\max}^2}{\eta} \tag{51}$$

Under Assumptions 1 and 2, Lemma 9 also holds, which gives us, for all $S, t$

$$\left(1 - \frac{e^{-\gamma^*} + \delta}{\kappa}\right)F(h_{\widehat{\boldsymbol{\theta}}_t}, S \mid a_{\boldsymbol{\phi}^*(\widehat{\boldsymbol{\theta}}_t,S)}) - \mathbb{E}[F(h_{\widehat{\boldsymbol{\theta}}_t}, \widehat{S}_t \mid a_{\boldsymbol{\phi}^*(\widehat{\boldsymbol{\theta}}_t,\widehat{S}_t)})] \leq 2\eta\nabla_{\max}^2 \tag{52}$$

We note that in Lemma 9, the expectation was taken over the randomness in stochastic distorted greedy algorithm. We now include the randomness due to stochastic gradient descent and take the total expectation to obtain,

$$\left(1 - \frac{e^{-\gamma^*} + \delta}{\kappa}\right)\mathbb{E}[F(h_{\widehat{\boldsymbol{\theta}}_t}, S \mid a_{\boldsymbol{\phi}^*(\widehat{\boldsymbol{\theta}}_t,S)})] - \mathbb{E}[F(h_{\widehat{\boldsymbol{\theta}}_t}, \widehat{S}_t \mid a_{\boldsymbol{\phi}^*(\widehat{\boldsymbol{\theta}}_t,\widehat{S}_t)})] \leq 2\eta\nabla_{\max}^2 \tag{53}$$

Summing the above equation over $t = 1$ to $T$ gives us, for all $\boldsymbol{\theta} \in \Theta, S$,

$$\sum_{t=1}^{T}\left[\left(1 - \frac{e^{-\gamma^*} + \delta}{\kappa}\right)\mathbb{E}[F(h_{\widehat{\boldsymbol{\theta}}_t}, S \,|\, a_{\boldsymbol{\phi}^*(\widehat{\boldsymbol{\theta}}_t, S)})] - \mathbb{E}[F(h_{\widehat{\boldsymbol{\theta}}_t}, \widehat{S}_t \,|\, a_{\boldsymbol{\phi}^*(\widehat{\boldsymbol{\theta}}_t, \widehat{S}_t)})]\right] \leq 2T\eta\nabla_{\max}^2 \quad (54)$$

Summing inequality (51) and inequality (54) and dividing by $T$, we obtain:

$$\left(1 - \frac{e^{-\gamma^*} + \delta}{\kappa}\right)\sum_{t=1}^{T}\frac{\mathbb{E}[F(h_{\widehat{\boldsymbol{\theta}}_t}, S \,|\, a_{\boldsymbol{\phi}^*(\widehat{\boldsymbol{\theta}}_t, S)})]}{T} \leq \left[\sum_{t=1}^{T}\frac{\mathbb{E}[F(h_{\boldsymbol{\theta}}, \widehat{S}_t \,|\, a_{\boldsymbol{\phi}^*(\boldsymbol{\theta}, \widehat{S}_t)})]}{T}\right] + 4\eta\nabla_{\max}^2 + \frac{2\theta_{\max}^2}{T\eta}$$

Since the above holds for all $S$, we take the maximum over all possible $S$, on both sides to obtain,

$$\left(1 - \frac{e^{-\gamma^*} + \delta}{\kappa}\right)\max_{S}\sum_{t=1}^{T}\frac{\mathbb{E}[F(h_{\widehat{\boldsymbol{\theta}}_t}, S \,|\, a_{\boldsymbol{\phi}^*(\widehat{\boldsymbol{\theta}}_t, S)})]}{T} \leq \left[\sum_{t=1}^{T}\frac{\max_{S}\mathbb{E}[F(h_{\boldsymbol{\theta}}, S \,|\, a_{\boldsymbol{\phi}^*(\boldsymbol{\theta}, S)})]}{T}\right] + 4\eta\nabla_{\max}^2 + \frac{2\theta_{\max}^2}{T\eta}$$

$$\leq \max_{S} F(h_{\boldsymbol{\theta}}, S \,|\, a_{\boldsymbol{\phi}^*(\boldsymbol{\theta}, S)}) + 4\eta\nabla_{\max}^2 + \frac{2\theta_{\max}^2}{T\eta}$$

Since $F$ is convex in $\boldsymbol{\theta}$, we get that for all $S$ and for $\widehat{\boldsymbol{\theta}} = \sum_{t=1}^{T}\frac{\widehat{\boldsymbol{\theta}}_t}{T}$,

$$F(h_{\widehat{\boldsymbol{\theta}}}, S \,|\, a_{\boldsymbol{\phi}^*(\widehat{\boldsymbol{\theta}}, S)}) \leq \sum_{t=1}^{T}\frac{F(h_{\widehat{\boldsymbol{\theta}}_t}, S \,|\, a_{\boldsymbol{\phi}^*(\widehat{\boldsymbol{\theta}}_t, S)})}{T} \quad (55)$$

Along with linearity of expectation, this gives us that for all $\boldsymbol{\theta} \in \Theta$,

$$\max_{S}\left(1 - \frac{e^{-\gamma^*} + \delta}{\kappa}\right)\mathbb{E}[F(h_{\widehat{\boldsymbol{\theta}}}, S \,|\, a_{\boldsymbol{\phi}^*(\widehat{\boldsymbol{\theta}}, S)})]$$

$$\leq \max_{S} F(h_{\boldsymbol{\theta}}, S \,|\, a_{\boldsymbol{\phi}^*(\boldsymbol{\theta}, S)}) + 4\eta\nabla_{\max}^2 + \frac{2\theta_{\max}^2}{T\eta} \quad (56)$$

Finally, we take the minimum over $\boldsymbol{\theta}$ on both sides. Noting that the LHS is independent of $\boldsymbol{\theta}$, we obtain:

$$\left(1 - \frac{e^{-\gamma^*} + \delta}{\kappa}\right)\max_{S}\mathbb{E}[F(h_{\widehat{\boldsymbol{\theta}}}, S \,|\, a_{\boldsymbol{\phi}^*(\widehat{\boldsymbol{\theta}}, S)})]$$

$$\leq \min_{\boldsymbol{\theta}}\max_{S} F(h_{\boldsymbol{\theta}}, S \,|\, a_{\boldsymbol{\phi}^*(\boldsymbol{\theta}, S)}) + 4\eta\nabla_{\max}^2 + \frac{2\theta_{\max}^2}{T\eta}$$

$$= OPT + 4\eta\nabla_{\max}^2 + \frac{2\theta_{\max}^2}{T\eta} \quad (57)$$

Setting $\eta = \frac{1}{\sqrt{T}}$, we obtain:

$$\left(1 - \frac{e^{-\gamma^*} + \delta}{\kappa}\right)\max_{S}\mathbb{E}[F(h_{\widehat{\boldsymbol{\theta}}}, S \,|\, a_{\boldsymbol{\phi}^*(\widehat{\boldsymbol{\theta}}, S)})] \leq OPT + \frac{1}{\sqrt{T}}\left[4\nabla_{\max}^2 + 2\theta_{\max}^2\right] \quad (58)$$

$$= OPT + \epsilon \quad (59)$$

Rearranging, we get the statement in the theorem. This completes the proof. $\square$

## F    AUXILIARY LEMMAS

**Proposition 6.** *If a function $Q$ satisfies $\alpha$-submodularity, i.e., $Q(k\,|\,S) > \alpha Q(k\,|\,T)$ for all $k \in D\backslash T$ with $S \subset T$, then $Q$ satisfies $\gamma$-weak submodularity (El Halabi et al., 2018, Proposition 8, Appendix)*

**Lemma 7.** *(Guarantees from Stochastic Distorted Greedy) Let , $g, c : 2^D \to \mathbb{R}^+$ be two non-negative, monotone set functions, such that $g$ is $\gamma$-weakly submodular for some $\gamma \in (0, 1]$ and $c$ is modular (see Definition 1). Furthermore, suppose that for all $S, c(S)/g(S) \leq 1 - \kappa$ for some $\kappa \in [0, 1)$.*

*Suppose we wish to solve the following problem:*

$$\max_{|S| \leq b} [g(S) - c(S)] \tag{60}$$

*for some fixed $K$. Let $S^*$ denote the value of set for which the maximum in optimization* (60) *is attained. Then, for a given value of $\delta > 0$ such that $\delta + e^{-\gamma} < 1$, Stochastic Distorted Greedy Algorithm makes $O(Dlog(1/\delta))$ evaluations of $g$ and returns a set $S'$ of size $|S'| \leq b$ which satisfies,*

$$g(S^*) - c(S^*) \leq \left(\frac{\kappa}{\kappa - e^{-\gamma} - \delta}\right) \mathbb{E}[g(S') - c(S')] \tag{61}$$

*As a corollary, we observe that for all $S$ such that $|S| \leq K$, it holds that,*

$$g(S) - c(S) \leq g(S^*) - c(S^*) \leq \left(\frac{\kappa}{\kappa - e^{-\gamma} - \delta}\right) \mathbb{E}[g(S') - c(S')] \tag{62}$$

*Proof.* We begin with the approximation guarantee on Stochastic Distorted Greedy Algorithm as stated in (Harshaw et al., 2019), Theorem 3. (Harshaw et al., 2019) show that on running Distorted Greedy with $K$ iterations for optimization problem (60), the output is a set $S'$ with size $|S'| \leq K$ and,

$$(1 - e^{-\gamma} - \delta)g(S^*) - c(S^*) \leq \mathbb{E}[g(S') - c(S')] \tag{63}$$

$$g(S^*) \leq \frac{\mathbb{E}[g(S') - c(S')]}{(1 - e^{-\gamma} - \delta)} + \frac{c(S^*)}{(1 - e^{-\gamma} - \delta)} \tag{64}$$

$$g(S^*) - c(S^*) \leq \frac{\mathbb{E}[g(S') - c(S')]}{(1 - e^{-\gamma} - \delta)} + \frac{e^{-\gamma} + \delta}{(1 - e^{-\gamma} - \delta)}c(S^*) \tag{65}$$

$$g(S^*) - c(S^*) \leq \frac{\mathbb{E}[g(S') - c(S')]}{(1 - e^{-\gamma} - \delta)} + \frac{(e^{-\gamma} + \delta)(1 - \kappa)}{(1 - e^{-\gamma} - \delta)\kappa}(g(S^*) - c(S^*)) \tag{66}$$

where in the last inequality we use definition of $\kappa$. Collecting the terms involving $g(S^*) - c(S^*)$ on the left, $g(S') - c(S')$ on the right and then simplifying gives us the result of the lemma. $\qquad\square$

**Lemma 8.** *(Derivation of $\kappa$) Let $G_\lambda(\boldsymbol{\theta}, S)$ and $m_\lambda(\boldsymbol{\theta}, S)$ be as defined in definition 4 and 5 respectively. Let Assumption 1 hold and suppose $\max_{\boldsymbol{\theta} \in \Theta} R(\boldsymbol{\theta}) = \theta_{\max}^2$ and for all $\boldsymbol{\theta} \in \Theta$, and let $\ell_{\max}, C_{\min}, \theta_{\max}$ be defined as in Assumption 1. Then, for all $\boldsymbol{\theta}, \forall S$, it holds that*

$$\frac{m_\lambda(\boldsymbol{\theta}, S)}{G_\lambda(\boldsymbol{\theta}, S)} \leq \frac{\lambda\theta_{\max}^2 + \rho\ell_{\max}}{\lambda\theta_{\max}^2 + \rho\ell_{\max} + \mu C_{\min}} \tag{67}$$

*Proof.* We first note that:

$$\frac{m_\lambda(\boldsymbol{\theta}, S)}{G_\lambda(\boldsymbol{\theta}, S)} = \frac{\lambda R(\boldsymbol{\theta})|S| + \rho \sum_{i \in S} \ell(h_{\boldsymbol{\theta}}(\boldsymbol{x}_i), y_i)}{\lambda R(\boldsymbol{\theta})|S| + \sum_{i \in S} \ell(h_{\boldsymbol{\theta}}(a_{\boldsymbol{\phi}^*(\boldsymbol{\theta},S)}(\boldsymbol{x}_i)), y_i) + \rho \sum_{j \in \mathcal{D}} \ell(h_{\boldsymbol{\theta}}(\boldsymbol{x}_j), y_j)} \tag{68}$$

Using the definition of $a_{\boldsymbol{\phi}^*(\boldsymbol{\theta},S)}$ from Eq. (3), we have, for all $\boldsymbol{\phi} \in \Phi$

$$\sum_{i \in S} \left[\ell(h_{\boldsymbol{\theta}}(a_{\boldsymbol{\phi}^*(\boldsymbol{\theta},S)}(\boldsymbol{x}_i)), y_i) - \mu C(a_{\boldsymbol{\phi}^*(\boldsymbol{\theta},S)}(\boldsymbol{x}_i), \boldsymbol{x}_i)\right] \geq \sum_{i \in S} \left[\ell(h_{\boldsymbol{\theta}}(a_{\boldsymbol{\phi}}(\boldsymbol{x}_i)), y_i) - \mu C(a_{\boldsymbol{\phi}}(\boldsymbol{x}_i), \boldsymbol{x}_i)\right]$$

$$\tag{69}$$

In particular, this also holds for $\phi$ such that for all $\boldsymbol{x}, a_\phi(\boldsymbol{x}) = \boldsymbol{x}$. This gives,

$$\sum_{i \in S} \left[ \ell(h_{\boldsymbol{\theta}}(a_{\phi^*(\boldsymbol{\theta}, S)}(\boldsymbol{x}_i)), y_i) - \mu C(a_{\phi^*(\boldsymbol{\theta}, S)}(\boldsymbol{x}_i), \boldsymbol{x}_i) \right] \geq \sum_{i \in S} \left[ \ell(h_{\boldsymbol{\theta}}(\boldsymbol{x}_i), y_i) - \mu C(\boldsymbol{x}_i, \boldsymbol{x}_i) \right] \quad (70)$$

$$\implies \sum_{i \in S} \left[ \ell(h_{\boldsymbol{\theta}}(a_{\phi^*(\boldsymbol{\theta}, S)}(\boldsymbol{x}_i)), y_i) \right] \geq \sum_{i \in S} \left[ \ell(h_{\boldsymbol{\theta}}(\boldsymbol{x}_i), y_i) + \mu C_{\min} \right] \quad (71)$$

The last line uses the definition of $C_{\min}$ from Assumption 1, and the fact that for all $\boldsymbol{x}, C(\boldsymbol{x}, \boldsymbol{x}) = 0$. Plugging inequality (71) back into inequality. (68), we obtain:

$$\frac{m_\lambda(\boldsymbol{\theta}, S)}{G_\lambda(\boldsymbol{\theta}, S)} \leq \frac{\lambda R(\boldsymbol{\theta})|S| + \rho \sum_{i \in S} \ell(h_{\boldsymbol{\theta}}(\boldsymbol{x}_i), y_i)}{\lambda R(\boldsymbol{\theta})|S| + \sum_{i \in S} \left[ \ell(h_{\boldsymbol{\theta}}(\boldsymbol{x}_i), y_i) + \mu C_{\min} \right] + \rho \sum_{j \in \mathcal{D}} \ell(h_{\boldsymbol{\theta}}(\boldsymbol{x}_j), y_j)} \quad (72)$$

$$\leq \frac{\lambda R(\boldsymbol{\theta})|S| + \rho \sum_{i \in S} \ell(h_{\boldsymbol{\theta}}(\boldsymbol{x}_i), y_i)}{\lambda R(\boldsymbol{\theta})|S| + \sum_{i \in S} \left[ \ell(h_{\boldsymbol{\theta}}(\boldsymbol{x}_i), y_i) + \mu C_{\min} \right] + \rho \sum_{j \in S} \ell(h_{\boldsymbol{\theta}}(\boldsymbol{x}_j), y_i)} \quad (73)$$

$$\leq \frac{\lambda R(\boldsymbol{\theta})|S| + \rho \ell_{\max}|S|}{\lambda R(\boldsymbol{\theta})|S| + \mu C_{\min}|S| + \rho \sum_{j \in \mathcal{S}} \ell(h_{\boldsymbol{\theta}}(\boldsymbol{x}_i), y_i)} \quad (74)$$

$$\leq \frac{\lambda R(\boldsymbol{\theta})|S| + \rho \ell_{\max}|S|}{\lambda R(\boldsymbol{\theta})|S| + \mu C_{\min}|S| + \rho \ell_{\max}|S|} \quad (75)$$

$$= \frac{\lambda R(\boldsymbol{\theta}) + \rho \ell_{\max}}{\lambda R(\boldsymbol{\theta}) + \rho \ell_{\max} + \mu C_{\min}} \quad (76)$$

Here inequalities (73) and (74) follow from non-negativity of $\ell$. Finally, substituting $R(\boldsymbol{\theta}) \leq \theta_{\max}^2$, we get the statement in the lemma. $\qquad \square$

**Lemma 9.** *Suppose the assumptions of Lemma 8 and Theorem 2 hold. Let $\kappa = \mu C_{\min}/(\lambda \theta_{\max}^2 + \rho \ell_{\max} + \mu C_{\min})$, where the symbols are the same as defined in the statement of Lemma 8 and $\gamma^*$ is defined in Theorem 2. Furthermore, let $\delta < (1 - e^{-\gamma^*})$ and let $\rho < [C_{\min}((e^{-\gamma^*} + \delta)^{-1} - 1) - \lambda \theta_{\max}^2]/\ell_{\max}$. Then for any $S$ such that $|S| \leq b$,*

$$\left( 1 - (e^{-\gamma^*} + \delta)/\kappa \right) F(h_{\widehat{\boldsymbol{\theta}}_t}, S \mid a_{\phi^*(\widehat{\boldsymbol{\theta}}_t, S)}) - \mathbb{E}[F(h_{\widehat{\boldsymbol{\theta}}_t}, \widehat{S}_t \mid a_{\phi^*(\widehat{\boldsymbol{\theta}}_t, \widehat{S}_t)})] \leq 2\eta \nabla_{\max}^2 \quad (77)$$

*where the expectation is w.r.t. the randomness in Stochastic Distorted Greedy.*

*Proof.* To obtain our guarantees, we require $\left( 1 - \frac{e^{-\gamma^*} + \delta}{\kappa} \right) \geq 0$. Substituting $\kappa = \mu C_{\min}/(\lambda \theta_{\max}^2 + \rho \ell_{\max} + \mu C_{\min})$ and simplifying gives us $\rho < [\mu C_{\min}((e^{-\gamma^*} + \delta)^{-1} - 1) - \lambda \theta_{\max}^2]/\ell_{\max}$. To ensure that the this bound is positive, we require $\delta < (1 - e^{-\gamma^*})$.

In our algorithm, we obtain $\widehat{S}_t$ by applying Stochastic Distorted Greedy using $\widehat{\boldsymbol{\theta}}_{t-1}$. Therefore, by our characterization of $F$ as stated in Eq. (6) and using Lemma 7, we have that for any $S$,

$$F(h_{\widehat{\boldsymbol{\theta}}_{t-1}}, S \mid a_{\phi^*(\widehat{\boldsymbol{\theta}}_{t-1}, S)}) \leq \left( \frac{\kappa}{\kappa - e^{-\gamma^*} - \delta} \right) \mathbb{E}[F(h_{\widehat{\boldsymbol{\theta}}_{t-1}}, \widehat{S}_t \mid a_{\phi^*(\widehat{\boldsymbol{\theta}}_{t-1}, \widehat{S}_t)})] \quad (78)$$

Using the $\nabla_{\max}$-Lipschitzness of $F$, we get

$$|F(h_{\widehat{\boldsymbol{\theta}}_t}, S \mid a_{\phi^*(\widehat{\boldsymbol{\theta}}_t, S)}) - F(h_{\widehat{\boldsymbol{\theta}}_{t-1}}, S \mid a_{\phi^*(\widehat{\boldsymbol{\theta}}_{t-1}, S)})| \leq \nabla_{\max} \|\widehat{\boldsymbol{\theta}}_t - \widehat{\boldsymbol{\theta}}_{t-1}\| \quad (79)$$

$$= \nabla_{\max} \eta \|\nabla_{\widehat{\boldsymbol{\theta}}_{t-1}} F(h_{\widehat{\boldsymbol{\theta}}_{t-1}}, S \mid a_{\phi^*(\widehat{\boldsymbol{\theta}}_{t-1}, S)})\| \quad (80)$$

$$\leq \nabla_{\max}^2 \eta \quad (81)$$

Now, using the Lipschitz condition, from inequality (81) we get that for $t$ and $S$

$$F(h_{\widehat{\boldsymbol{\theta}}_t}, S \mid a_{\phi^*(\widehat{\boldsymbol{\theta}}_t, S)}) \leq F(h_{\widehat{\boldsymbol{\theta}}_{t-1}}, S \mid a_{\phi^*(\widehat{\boldsymbol{\theta}}_{t-1}, S)}) + \eta \nabla_{\max}^2 \quad (82)$$

$$\leq \left( \frac{\kappa}{\kappa - e^{-\gamma^*} - \delta} \right) \mathbb{E}[F(h_{\widehat{\boldsymbol{\theta}}_{t-1}}, \widehat{S}_t \mid a_{\phi^*(\widehat{\boldsymbol{\theta}}_{t-1}, \widehat{S}_t)})] + \eta \nabla_{\max}^2 \quad (83)$$

$$\implies \left(1 - \frac{e^{-\gamma^*} + \delta}{\kappa}\right) F(h_{\widehat{\boldsymbol{\theta}}_t}, S \mid a_{\boldsymbol{\phi}^*(\widehat{\boldsymbol{\theta}}_t, S)}) \leq \mathbb{E}[F(h_{\widehat{\boldsymbol{\theta}}_{t-1}}, \widehat{S}_t \mid a_{\boldsymbol{\phi}^*(\widehat{\boldsymbol{\theta}}_{t-1}, \widehat{S}_t)})] + \left(1 - \frac{e^{-\gamma^*} + \delta}{\kappa}\right) \eta \nabla_{\max}^2$$

(84)

Here, inequality (83) is obtained by using inequality (78). Subtracting $\mathbb{E}[F(h_{\widehat{\boldsymbol{\theta}}_t}, \widehat{S}_t \mid a_{\boldsymbol{\phi}^*(\widehat{\boldsymbol{\theta}}_t, \widehat{S}_t)})]$ from both sides of inequality (84), we obtain:

$$\left(1 - \frac{e^{-\gamma^*} + \delta}{\kappa}\right) F(h_{\widehat{\boldsymbol{\theta}}_t}, S \mid a_{\boldsymbol{\phi}^*(\widehat{\boldsymbol{\theta}}_t, S)}) - \mathbb{E}[F(h_{\widehat{\boldsymbol{\theta}}_t}, \widehat{S}_t \mid a_{\boldsymbol{\phi}^*(\widehat{\boldsymbol{\theta}}_t, \widehat{S}_t)})]$$

(85)

$$\leq \mathbb{E}[F(h_{\widehat{\boldsymbol{\theta}}_{t-1}}, \widehat{S}_t \mid a_{\boldsymbol{\phi}^*(\widehat{\boldsymbol{\theta}}_{t-1}, \widehat{S}_t)}) - F(h_{\widehat{\boldsymbol{\theta}}_t}, \widehat{S}_t \mid a_{\boldsymbol{\phi}^*(\widehat{\boldsymbol{\theta}}_t, \widehat{S}_t)})] + \left(1 - \frac{e^{-\gamma^*} + \delta}{\kappa}\right) \eta \nabla_{\max}^2 \quad (86)$$

Using inequality (81), we conclude that the above quantity is less than

$$\eta \nabla_{\max}^2 + \left(1 - \frac{e^{-\gamma^*} + \delta}{\kappa}\right) \eta \nabla_{\max}^2 = \left(2 - \frac{e^{-\gamma^*} + \delta}{\kappa}\right) \eta \nabla_{\max}^2 \leq 2\eta \nabla_{\max}^2$$

(87)

This completes the proof of the lemma. $\qquad\square$

**Lemma 10.** *Suppose Assumption 1 and 2 hold. Let $\widehat{\nabla}_{\boldsymbol{\theta}} F(h_{\boldsymbol{\theta}}, S \mid a_{\boldsymbol{\phi}^*(\boldsymbol{\theta}, S)})$ denote the stochastic gradient of $F$ at $h_{\boldsymbol{\theta}}, S$ such that $\mathbb{E}[\widehat{\nabla}_{\boldsymbol{\theta}} F(h_{\boldsymbol{\theta}}, S \mid a_{\boldsymbol{\phi}^*(\boldsymbol{\theta}, S)})] = \nabla_{\boldsymbol{\theta}} F(h_{\boldsymbol{\theta}}, S \mid a_{\boldsymbol{\phi}^*(\boldsymbol{\theta}, S)})$, for all $\boldsymbol{\theta}, S$. Furthermore, suppose that for all $\boldsymbol{\theta}, S$, we have, $\|\widehat{\nabla}_{\boldsymbol{\theta}} F(h_{\boldsymbol{\theta}}, S \mid a_{\boldsymbol{\phi}^*(\boldsymbol{\theta}, S)})\| \leq \nabla_{\max}$. For any $\boldsymbol{\theta}, S$, let $\boldsymbol{\theta}' = \boldsymbol{\theta} - \eta \widehat{\nabla}_{\boldsymbol{\theta}} F(h_{\boldsymbol{\theta}}, S \mid a_{\boldsymbol{\phi}^*(\boldsymbol{\theta}, S)})$. Then, for any $\tilde{\boldsymbol{\theta}} \in \Theta$*

$$\mathbb{E}[F(h_{\boldsymbol{\theta}'}, S \mid a_{\boldsymbol{\phi}^*(\boldsymbol{\theta}', S)}) \mid \boldsymbol{\theta}] - F(h_{\tilde{\boldsymbol{\theta}}}, S \mid a_{\boldsymbol{\phi}^*(\tilde{\boldsymbol{\theta}}, S)}) \leq \frac{\eta}{2}(L\eta + 1)\nabla_{\max}^2 + \frac{1}{2\eta}\left[\|\boldsymbol{\theta} - \tilde{\boldsymbol{\theta}}\|^2 - \mathbb{E}[\|\boldsymbol{\theta}' - \tilde{\boldsymbol{\theta}}\|^2 \mid \boldsymbol{\theta}]\right]$$

*where the expectation is over the randomness in computing the stochastic gradient $\widehat{\nabla}_{\boldsymbol{\theta}} F(h_{\boldsymbol{\theta}}, S \mid a_{\boldsymbol{\phi}^*(\boldsymbol{\theta}, S)})$.*

*Proof.* In what follows, we fix $S$ and denote $F(h_{\boldsymbol{\theta}}, S \mid a_{\boldsymbol{\phi}^*(\boldsymbol{\theta}, S)}) \equiv F(h_{\boldsymbol{\theta}})$. We do this for brevity and succinctness. Using first $L$-smoothness and then $\boldsymbol{\theta}' = \boldsymbol{\theta} - \eta \widehat{\nabla}_{\boldsymbol{\theta}} F(h_{\boldsymbol{\theta}}, S \mid a_{\boldsymbol{\phi}^*(\boldsymbol{\theta}, S)})$, we obtain:

$$F(h_{\boldsymbol{\theta}'}) \leq F(h_{\boldsymbol{\theta}}) + \langle \nabla_{\boldsymbol{\theta}} F(h_{\boldsymbol{\theta}}), \boldsymbol{\theta}' - \boldsymbol{\theta} \rangle + \frac{L}{2}\|\boldsymbol{\theta}' - \boldsymbol{\theta}\|^2$$

(88)

$$\implies F(h_{\boldsymbol{\theta}'}) - F(h_{\boldsymbol{\theta}}) \leq -\eta \langle \nabla_{\boldsymbol{\theta}} F(h_{\boldsymbol{\theta}}), \widehat{\nabla}_{\boldsymbol{\theta}} F(h_{\boldsymbol{\theta}}) \rangle + \frac{L\eta^2}{2}\|\widehat{\nabla}_{\boldsymbol{\theta}} F(h_{\boldsymbol{\theta}})\|^2$$

(89)

Taking expectations on both sides over the randomness in computing SGD, we obtain:

$$\mathbb{E}[F(h_{\boldsymbol{\theta}'}) - F(h_{\boldsymbol{\theta}}) \mid \boldsymbol{\theta}] \leq -\eta \mathbb{E}[\langle \nabla_{\boldsymbol{\theta}} F(h_{\boldsymbol{\theta}}), \widehat{\nabla}_{\boldsymbol{\theta}} F(h_{\boldsymbol{\theta}}) \rangle \mid \boldsymbol{\theta}] + \frac{L\eta^2}{2}\mathbb{E}[\|\widehat{\nabla}_{\boldsymbol{\theta}} F(h_{\boldsymbol{\theta}})\|^2 \mid \boldsymbol{\theta}] \quad (90)$$

$$\implies \mathbb{E}[F(h_{\boldsymbol{\theta}'}) \mid \boldsymbol{\theta}] - F(h_{\boldsymbol{\theta}}) \leq -\eta \langle \nabla_{\boldsymbol{\theta}} F(h_{\boldsymbol{\theta}}), \mathbb{E}[\widehat{\nabla}_{\boldsymbol{\theta}} F(h_{\boldsymbol{\theta}}) \mid \boldsymbol{\theta}] \rangle + \frac{L\eta^2}{2}\mathbb{E}[\|\widehat{\nabla}_{\boldsymbol{\theta}} F(h_{\boldsymbol{\theta}})\|^2 \mid \boldsymbol{\theta}] \quad (91)$$

$$= -\eta \|\nabla_{\boldsymbol{\theta}} F(h_{\boldsymbol{\theta}})\|^2 + \frac{L\eta^2}{2}\mathbb{E}[\|\widehat{\nabla}_{\boldsymbol{\theta}} F(h_{\boldsymbol{\theta}})\|^2 \mid \boldsymbol{\theta}] \quad (92)$$

Since $F$ is convex in $\tilde{\theta}$, we have:

$$F(h_{\boldsymbol{\theta}}) - F(h_{\tilde{\boldsymbol{\theta}}}) \leq \langle \nabla_{\boldsymbol{\theta}} F(h_{\boldsymbol{\theta}}), \boldsymbol{\theta} - \tilde{\boldsymbol{\theta}} \rangle$$

(93)

Adding the inequality (93) and inequality (92), we obtain:

$$\mathbb{E}[F(h_{\boldsymbol{\theta}'}) \mid \boldsymbol{\theta}] - F(h_{\tilde{\boldsymbol{\theta}}}) \leq -\eta \|\nabla_{\boldsymbol{\theta}} F(h_{\boldsymbol{\theta}})\|^2 + \frac{L\eta^2}{2}\mathbb{E}[\|\widehat{\nabla}_{\boldsymbol{\theta}} F(h_{\boldsymbol{\theta}})\|^2 \mid \boldsymbol{\theta}] + \langle \nabla_{\boldsymbol{\theta}} F(h_{\boldsymbol{\theta}}), \boldsymbol{\theta} - \tilde{\boldsymbol{\theta}} \rangle \quad (94)$$

We derive a value for $\langle \nabla_{\boldsymbol{\theta}} F(h_{\boldsymbol{\theta}}), \boldsymbol{\theta} - \tilde{\boldsymbol{\theta}} \rangle$ using the following:

$$\|\boldsymbol{\theta}' - \tilde{\boldsymbol{\theta}}\|^2 = \|\boldsymbol{\theta}' - \boldsymbol{\theta} + \boldsymbol{\theta} - \tilde{\boldsymbol{\theta}}\|^2$$

(95)

$$= \|\boldsymbol{\theta}' - \boldsymbol{\theta}\|^2 + \|\boldsymbol{\theta} - \tilde{\boldsymbol{\theta}}\|^2 + 2\langle \boldsymbol{\theta}' - \boldsymbol{\theta}, \boldsymbol{\theta} - \tilde{\boldsymbol{\theta}} \rangle$$

(96)

$$= \| - \eta \widehat{\nabla}_{\boldsymbol{\theta}} F(h_{\boldsymbol{\theta}}) \|^2 + \| \boldsymbol{\theta} - \tilde{\boldsymbol{\theta}} \|^2 + 2 \langle -\eta \widehat{\nabla}_{\boldsymbol{\theta}} F(h_{\boldsymbol{\theta}}), \boldsymbol{\theta} - \tilde{\boldsymbol{\theta}} \rangle \tag{97}$$

$$= \eta^2 \| \widehat{\nabla}_{\boldsymbol{\theta}} F(h_{\boldsymbol{\theta}}) \|^2 + \| \boldsymbol{\theta} - \tilde{\boldsymbol{\theta}} \|^2 - 2\eta \langle \widehat{\nabla}_{\boldsymbol{\theta}} F(h_{\boldsymbol{\theta}}), \boldsymbol{\theta} - \tilde{\boldsymbol{\theta}} \rangle \tag{98}$$

Taking expectation on both sides, given $\boldsymbol{\theta}$ (noting that $\tilde{\boldsymbol{\theta}}$ does not depend on $\boldsymbol{\theta}$), we get

$$\mathbb{E}\left[ \| \boldsymbol{\theta}' - \tilde{\boldsymbol{\theta}} \|^2 \,|\, \boldsymbol{\theta} \right] - \| \boldsymbol{\theta} - \tilde{\boldsymbol{\theta}} \|^2 = \eta^2 \mathbb{E}\left[ \| \widehat{\nabla}_{\boldsymbol{\theta}} F(h_{\boldsymbol{\theta}}) \|^2 \,|\, \boldsymbol{\theta} \right] - 2\eta \mathbb{E}\left[ \langle \widehat{\nabla}_{\boldsymbol{\theta}} F(h_{\boldsymbol{\theta}}), \boldsymbol{\theta} - \tilde{\boldsymbol{\theta}} \rangle | \boldsymbol{\theta} \right] \tag{99}$$

$$= \eta^2 \mathbb{E}[\| \widehat{\nabla}_{\boldsymbol{\theta}} F(h_{\boldsymbol{\theta}}) \|^2 \,|\, \boldsymbol{\theta}] - 2\eta \langle \nabla_{\boldsymbol{\theta}} F(h_{\boldsymbol{\theta}}), \boldsymbol{\theta} - \tilde{\boldsymbol{\theta}} \rangle \tag{100}$$

Rearranging, we obtain:

$$\langle \nabla_{\boldsymbol{\theta}} F(h_{\boldsymbol{\theta}}), \boldsymbol{\theta} - \tilde{\boldsymbol{\theta}} \rangle = \frac{\eta}{2} \mathbb{E}[\| \widehat{\nabla}_{\boldsymbol{\theta}} F(h_{\boldsymbol{\theta}}) \|^2 \,|\, \boldsymbol{\theta}] - \frac{1}{2\eta} \left[ \mathbb{E}[\| \boldsymbol{\theta}' - \tilde{\boldsymbol{\theta}} \|^2 \,|\, \boldsymbol{\theta}] - \| \boldsymbol{\theta} - \tilde{\boldsymbol{\theta}} \|^2 \right] \tag{101}$$

Plugging inequality (101) back into the inequality (94), we get

$$\mathbb{E}[F(h_{\boldsymbol{\theta}'})|\boldsymbol{\theta}] - F(h_{\tilde{\boldsymbol{\theta}}})$$

$$\leq \frac{\eta}{2}(L\eta + 1)\mathbb{E}[\| \widehat{\nabla}_{\boldsymbol{\theta}} F(h_{\boldsymbol{\theta}}) \|^2 \,|\, \boldsymbol{\theta}] - \eta \| \nabla_{\boldsymbol{\theta}} F(h_{\boldsymbol{\theta}}) \|^2 + \frac{1}{2\eta}\left[ \| \boldsymbol{\theta} - \tilde{\boldsymbol{\theta}} \|^2 - \mathbb{E}[\| \boldsymbol{\theta}' - \tilde{\boldsymbol{\theta}} \|^2 \,|\, \boldsymbol{\theta}] \right]$$

$$\leq \frac{\eta}{2}(L\eta + 1)\nabla^2_{\max} + \frac{1}{2\eta}\left[ \| \boldsymbol{\theta} - \tilde{\boldsymbol{\theta}} \|^2 - \mathbb{E}[\| \boldsymbol{\theta}' - \tilde{\boldsymbol{\theta}} \|^2 \,|\, \boldsymbol{\theta}] \right] \tag{102}$$

$$\square$$

**Lemma 11.** *Suppose the assumptions of Lemma 10 hold. Furthermore, let $\widehat{\boldsymbol{\theta}}_t, \widehat{S}_t$ denote the iterates of our algorithm. Then, for any $\tilde{\boldsymbol{\theta}} \in \Theta$,*

$$\sum_{t=1}^{T} \left[ \mathbb{E}[F(h_{\widehat{\boldsymbol{\theta}}_t}, \widehat{S}_t \,|\, a_{\boldsymbol{\phi}^*(\widehat{\boldsymbol{\theta}}_t, \widehat{S}_t)})] - \mathbb{E}[F(h_{\tilde{\boldsymbol{\theta}}}, \widehat{S}_t \,|\, a_{\boldsymbol{\phi}^*(\tilde{\boldsymbol{\theta}}, \widehat{S}_t)})] \right] \leq 2T\eta \nabla^2_{\max} + \frac{2\theta^2_{\max}}{\eta} \tag{103}$$

*Here the expectation is taken w.r.t. the total randomness of the algorithm. This includes the randomness in stochastic gradient descent as well as stochastic distorted greedy.*

*Proof.* Putting $\boldsymbol{\theta}' = \widehat{\boldsymbol{\theta}}_t, \boldsymbol{\theta} = \widehat{\boldsymbol{\theta}}_{t-1}$ and $S = \widehat{S}_{t-1}$, in the statement of Lemma 10 we obtain: that for any $\tilde{\boldsymbol{\theta}} \in \Theta$

$$\mathbb{E}[F(h_{\widehat{\boldsymbol{\theta}}_t}, \widehat{S}_{t-1} \,|\, a_{\boldsymbol{\phi}^*(\widehat{\boldsymbol{\theta}}_t, \widehat{S}_{t-1})})|\widehat{\boldsymbol{\theta}}_{t-1}] - F(h_{\tilde{\boldsymbol{\theta}}}, \widehat{S}_{t-1} \,|\, a_{\boldsymbol{\phi}^*(\tilde{\boldsymbol{\theta}}, \widehat{S}_{t-1})})$$

$$\leq \frac{\eta}{2}(L\eta + 1)\nabla^2_{\max} + \frac{1}{2\eta}\left[ \| \widehat{\boldsymbol{\theta}}_{t-1} - \tilde{\boldsymbol{\theta}} \|^2 - \mathbb{E}\left[ \| \widehat{\boldsymbol{\theta}}_t - \tilde{\boldsymbol{\theta}} \|^2 \,|\, \widehat{\boldsymbol{\theta}}_{t-1} \right] \right] \tag{104}$$

We now take the expectation w.r.t. $\widehat{\boldsymbol{\theta}}_0, \widehat{\boldsymbol{\theta}}_1, \ldots, \widehat{\boldsymbol{\theta}}_t$. Let $\widehat{\boldsymbol{\theta}}_{0:\tau} := \widehat{\boldsymbol{\theta}}_0, \ldots, \widehat{\boldsymbol{\theta}}_{\tau}$. Using law of total expectation and noting that given $\widehat{\boldsymbol{\theta}}_{t-1}$, $\widehat{\boldsymbol{\theta}}_t$ is independent of $\widehat{\boldsymbol{\theta}}_{\tau}$ for $\tau \neq t-1$, we get

$$\mathbb{E}_{\widehat{\boldsymbol{\theta}}_{0:t}} \left[ F(h_{\widehat{\boldsymbol{\theta}}_t}, \widehat{S}_{t-1} \,|\, a_{\boldsymbol{\phi}^*(\widehat{\boldsymbol{\theta}}_t, \widehat{S}_{t-1})}) \right] = \mathbb{E}_{\widehat{\boldsymbol{\theta}}_{0:t-1}} \left[ \mathbb{E}_{\widehat{\boldsymbol{\theta}}_{0:t}|\widehat{\boldsymbol{\theta}}_{0:t-1}} \left[ F(h_{\widehat{\boldsymbol{\theta}}_t}, \widehat{S}_{t-1} \,|\, a_{\boldsymbol{\phi}^*(\widehat{\boldsymbol{\theta}}_t, \widehat{S}_{t-1})})|\widehat{\boldsymbol{\theta}}_0, \ldots, \widehat{\boldsymbol{\theta}}_{t-1} \right] \right]$$
$$\tag{105}$$

$$= \mathbb{E}_{\widehat{\boldsymbol{\theta}}_{0:t-1}} \left[ \mathbb{E}_{\widehat{\boldsymbol{\theta}}_t|\widehat{\boldsymbol{\theta}}_{0:t-1}} \left[ F(h_{\widehat{\boldsymbol{\theta}}_t}, \widehat{S}_{t-1} \,|\, a_{\boldsymbol{\phi}^*(\widehat{\boldsymbol{\theta}}_t, \widehat{S}_{t-1})})|\widehat{\boldsymbol{\theta}}_0, \ldots, \widehat{\boldsymbol{\theta}}_{t-1} \right] \right]$$
$$\tag{106}$$

$$= \mathbb{E}_{\widehat{\boldsymbol{\theta}}_{0:t-1}} \left[ \mathbb{E}_{\widehat{\boldsymbol{\theta}}_t|\widehat{\boldsymbol{\theta}}_{t-1}} \left[ F(h_{\widehat{\boldsymbol{\theta}}_t}, \widehat{S}_{t-1} \,|\, a_{\boldsymbol{\phi}^*(\widehat{\boldsymbol{\theta}}_t, \widehat{S}_{t-1})})|\widehat{\boldsymbol{\theta}}_{t-1} \right] \right] \tag{107}$$

Finally, we note that $\mathbb{E}_{\widehat{\boldsymbol{\theta}}_{0:T}} [F(h_{\widehat{\boldsymbol{\theta}}_t}, \widehat{S}_{t-1} \,|\, a_{\boldsymbol{\phi}^*(\widehat{\boldsymbol{\theta}}_t, \widehat{S}_{t-1})})] = \mathbb{E}_{\widehat{\boldsymbol{\theta}}_{0:t}} [F(h_{\widehat{\boldsymbol{\theta}}_t}, \widehat{S}_{t-1} \,|\, a_{\boldsymbol{\phi}^*(\widehat{\boldsymbol{\theta}}_t, \widehat{S}_{t-1})}))]$ as later iterations cannot impact previous iterations. We finally take the expectation w.r.t the randomness in stochastic distorted greedy as well. Thus, we obtain:

$$\mathbb{E}[F(h_{\widehat{\boldsymbol{\theta}}_t}, \widehat{S}_{t-1} \,|\, a_{\boldsymbol{\phi}^*(\widehat{\boldsymbol{\theta}}_t, \widehat{S}_{t-1})})] - \mathbb{E}[F(h_{\tilde{\boldsymbol{\theta}}}, \widehat{S}_{t-1} \,|\, a_{\boldsymbol{\phi}^*(\tilde{\boldsymbol{\theta}}, \widehat{S}_{t-1})})]$$
$$\leq \frac{\eta}{2}(L\eta + 1)\nabla_{\max}^2 + \frac{1}{2\eta}\left[\mathbb{E}[\|\widehat{\boldsymbol{\theta}}_{t-1} - \tilde{\boldsymbol{\theta}}\|^2] - \mathbb{E}[\|\widehat{\boldsymbol{\theta}}_t - \tilde{\boldsymbol{\theta}}\|^2]\right]$$

where $\mathbb{E}$ now denotes the expectation w.r.t. the randomness in the entire procedure (stochastic gradient descent and stochastic distorted greedy). Using Lipschitzness of $F$, we get that

$$|F(h_{\widehat{\boldsymbol{\theta}}_t}, \widehat{S}_{t-1} \,|\, a_{\boldsymbol{\phi}^*(\widehat{\boldsymbol{\theta}}_t, \widehat{S}_{t-1})}) - F(h_{\widehat{\boldsymbol{\theta}}_{t-1}}, \widehat{S}_{t-1} \,|\, a_{\boldsymbol{\phi}^*(\widehat{\boldsymbol{\theta}}_{t-1}, \widehat{S}_{t-1})})| \leq \nabla_{\max}\|\widehat{\boldsymbol{\theta}}_t - \widehat{\boldsymbol{\theta}}_{t-1}\|$$
$$F(h_{\widehat{\boldsymbol{\theta}}_{t-1}}, \widehat{S}_{t-1} \,|\, a_{\boldsymbol{\phi}^*(\widehat{\boldsymbol{\theta}}_{t-1}, \widehat{S}_{t-1})}) \leq F(h_{\widehat{\boldsymbol{\theta}}_t}, \widehat{S}_{t-1} \,|\, a_{\boldsymbol{\phi}^*(\widehat{\boldsymbol{\theta}}_t, \widehat{S}_{t-1})}) + \eta\nabla_{\max}^2$$

Plugging this in the equation above, we get

$$\mathbb{E}[F(h_{\widehat{\boldsymbol{\theta}}_{t-1}}, \widehat{S}_{t-1} \,|\, a_{\boldsymbol{\phi}^*(\widehat{\boldsymbol{\theta}}_{t-1}, \widehat{S}_{t-1})})] - F(h_{\tilde{\boldsymbol{\theta}}}, \widehat{S}_{t-1} \,|\, a_{\boldsymbol{\phi}^*(\tilde{\boldsymbol{\theta}}, \widehat{S}_{t-1})})$$
$$\leq \eta\nabla_{\max}^2 + \frac{\eta}{2}(L\eta + 1)\nabla_{\max}^2 + \frac{1}{2\eta}\left[\mathbb{E}[\|\widehat{\boldsymbol{\theta}}_{t-1} - \tilde{\boldsymbol{\theta}}\|^2] - \mathbb{E}[\|\widehat{\boldsymbol{\theta}}_t - \tilde{\boldsymbol{\theta}}\|^2]\right]$$

Setting $\eta < 1/L$ and simplifying gives us the following,

$$\mathbb{E}[F(h_{\widehat{\boldsymbol{\theta}}_t}, \widehat{S}_t \,|\, a_{\boldsymbol{\phi}^*(\widehat{\boldsymbol{\theta}}_t, \widehat{S}_t)})] - \mathbb{E}[F(h_{\tilde{\boldsymbol{\theta}}}, \widehat{S}_t \,|\, a_{\boldsymbol{\phi}^*(\tilde{\boldsymbol{\theta}}, \widehat{S}_t)})]$$
$$\leq 2\eta\nabla_{\max}^2 + + \frac{1}{2\eta}\left[\mathbb{E}[\|\widehat{\boldsymbol{\theta}}_t - \tilde{\boldsymbol{\theta}}\|^2] - \mathbb{E}[\|\widehat{\boldsymbol{\theta}}_{t+1} - \tilde{\boldsymbol{\theta}}\|^2]\right]$$

Summing over all $T$ gives us a telescoping sum on the right. Simplifying, we obtain:

$$\sum_{t=1}^{T}\left[\mathbb{E}[F(h_{\widehat{\boldsymbol{\theta}}_t}, \widehat{S}_t \,|\, a_{\boldsymbol{\phi}^*(\widehat{\boldsymbol{\theta}}_t, \widehat{S}_t)})] - \mathbb{E}[F(h_{\tilde{\boldsymbol{\theta}}}, \widehat{S}_t \,|\, a_{\boldsymbol{\phi}^*(\tilde{\boldsymbol{\theta}}, \widehat{S}_t)})]\right] \leq 2T\eta\nabla_{\max}^2 + \frac{1}{2\eta}\mathbb{E}[\|\widehat{\boldsymbol{\theta}}_1 - \tilde{\boldsymbol{\theta}}\|^2]$$
$$\tag{108}$$

We give an upper bound on $\|\widehat{\boldsymbol{\theta}}_1 - \tilde{\boldsymbol{\theta}}\|^2$ as follows

$$\|\widehat{\boldsymbol{\theta}}_1 - \tilde{\boldsymbol{\theta}}\|^2 \leq \left[\|\widehat{\boldsymbol{\theta}}_1\| + \|\tilde{\boldsymbol{\theta}}\|\right]^2 \leq (2\theta_{\max})^2 = 4\theta_{\max}^2 \tag{109}$$

Substituting upper bound from inequality (109) in inequality (108) gives us the statement of the Lemma:

$$\sum_{t=1}^{T}\left[\mathbb{E}[F(h_{\widehat{\boldsymbol{\theta}}_t}, \widehat{S}_t \,|\, a_{\boldsymbol{\phi}^*(\widehat{\boldsymbol{\theta}}_t, \widehat{S}_t)})] - \mathbb{E}[F(h_{\tilde{\boldsymbol{\theta}}}, \widehat{S}_t \,|\, a_{\boldsymbol{\phi}^*(\tilde{\boldsymbol{\theta}}, \widehat{S}_t)})]\right] \leq 2T\eta\nabla_{\max}^2 + \frac{2\theta_{\max}^2}{\eta} \tag{110}$$

$$\square$$

### F.1 AUXILIARY LEMMAS FOR THEOREM 4

**Lemma 12.** *($k$-SGD guarantee for a fixed $S$) Suppose Assumption 1 and 2 hold and $F$ is a non-convex function that satisfies the PL condition. Fix $S$ and suppose $\boldsymbol{\theta}^{(1)}, \ldots, \boldsymbol{\theta}^{(k)}$ are obtained using $k$-step stochastic gradient descent, for the fixed $S$, starting from $\boldsymbol{\theta}^{(0)}$. Then, for any $\tilde{\boldsymbol{\theta}} \in \Theta$, we have,*

$$\mathbb{E}[F(h_{\boldsymbol{\theta}^{(k)}}, S \,|\, a_{\boldsymbol{\phi}^*(\boldsymbol{\theta}^{(k)}, S)}) \,|\, \boldsymbol{\theta}^{(0)}, \ldots, \boldsymbol{\theta}^{(k-1)}] - F(h_{\tilde{\boldsymbol{\theta}}}, S \,|\, a_{\boldsymbol{\phi}^*(\tilde{\boldsymbol{\theta}}, S)}) \leq \frac{4L\theta_{\max}^2(1 - \eta\sigma)^k}{\sigma} + \frac{L\eta\nabla_{\max}^2}{2\sigma}$$

*where the expectation is w.r.t. randomness in computing the stochastic gradient.*

*Proof.* Since $S$ is fixed, we drop the second argument and denote $F_S(h_{\boldsymbol{\theta}}, S \,|\, a_{\boldsymbol{\phi}^*(\boldsymbol{\theta}, S)}) \equiv F_S(h_{\boldsymbol{\theta}})$ for succinctness and brevity. For $i = 0, \ldots, k-1$, we have, $\boldsymbol{\theta}^{(i+1)} = \boldsymbol{\theta}^{(i)} - \eta\widehat{\nabla}_{\boldsymbol{\theta}^{(i)}}F_S(h_{\boldsymbol{\theta}^{(i)}})$. Using $L$-smoothness, we obtain:

$$F_S(h_{\boldsymbol{\theta}^{(k)}}) \leq F_S(h_{\boldsymbol{\theta}^{(k-1)}}) + \langle\nabla_{\boldsymbol{\theta}^{(i)}}F_S(h_{\boldsymbol{\theta}^{(i)}}), \boldsymbol{\theta}^{(k)} - \boldsymbol{\theta}^{(k-1)}\rangle + \frac{L}{2}\|\boldsymbol{\theta}^{(k)} - \boldsymbol{\theta}^{(k-1)}\|^2$$

Taking expectation (given $\boldsymbol{\theta}^{(k-1)}$) on both sides w.r.t. randomness in stochastic gradient descent, we obtain:

$$\mathbb{E}[F_S(h_{\boldsymbol{\theta}^{(k)}}) \,|\, \boldsymbol{\theta}^{(k-1)}] \leq F_S(h_{\boldsymbol{\theta}^{(k-1)}}) - \eta \big\langle \nabla_{\boldsymbol{\theta}^{(k-1)}} F_S(h_{\boldsymbol{\theta}^{(k-1)}}), \mathbb{E}[\widehat{\nabla}_{\boldsymbol{\theta}^{(k-1)}} F_S(h_{\boldsymbol{\theta}^{(k-1)}})] \big\rangle \quad (111)$$

$$+ \frac{L\eta^2}{2} \mathbb{E}[\|\widehat{\nabla}_{\boldsymbol{\theta}^{(k-1)}} F_S(h_{\boldsymbol{\theta}^{(i)}})\|^2 \,|\, ] \quad (112)$$

$$\leq F_S(h_{\boldsymbol{\theta}^{(k-1)}}) - \eta \|\nabla_{\boldsymbol{\theta}^{(k-1)}} F_S(h_{\boldsymbol{\theta}^{(k-1)}})\|^2 + \frac{L\eta^2}{2} \nabla_{\max}^2 \quad (113)$$

The PL condition implies that,

$$\|\nabla_{\boldsymbol{\theta}^{(k-1)}} F_S(h_{\boldsymbol{\theta}^{(k-1)}})\|^2 \geq \sigma [F_S(h_{\boldsymbol{\theta}^{(k-1)}}) - F_S(h_{\boldsymbol{\theta}_S^*})] \quad (114)$$

where $\boldsymbol{\theta}_S^* = \min_{\boldsymbol{\theta}'} F(h_{\boldsymbol{\theta}'}, S \,|\, a_{\boldsymbol{\phi}^*(\boldsymbol{\theta}', S)})$. Substituting this in the inequality (113), we obtain:

$$\mathbb{E}[F_S(h_{\boldsymbol{\theta}^{(k)}}) \,|\, \boldsymbol{\theta}^{(k-1)}] \leq F_S(h_{\boldsymbol{\theta}^{(k-1)}}) - \eta\sigma [F_S(h_{\boldsymbol{\theta}^{(k-1)}}) - F_S(h_{\boldsymbol{\theta}_S^*})] + \frac{L\eta^2 \nabla_{\max}^2}{2} \quad (115)$$

Subtracting $F_S(h_{\boldsymbol{\theta}_S^*})$ from both sides and simplifying gives us,

$$\mathbb{E}[F_S(h_{\boldsymbol{\theta}^{(k)}}) \,|\, \boldsymbol{\theta}^{(k-1)}] - F_S(h_{\boldsymbol{\theta}_S^*}) \leq (1 - \eta\sigma)[F_S(h_{\boldsymbol{\theta}^{(k-1)}}) - F_S(h_{\boldsymbol{\theta}_S^*})] + \frac{L\eta^2 \nabla_{\max}^2}{2} \quad (116)$$

Now, we take expectation given $\boldsymbol{\theta}^{(k-2)}$ and get,

$$\mathbb{E}[F_S(h_{\boldsymbol{\theta}^{(k)}}) \,|\, \boldsymbol{\theta}^{(k-1)}, \boldsymbol{\theta}^{(k-2)}] - F_S(h_{\boldsymbol{\theta}_S^*})$$
$$\leq (1 - \eta\sigma)[\mathbb{E}[F_S(h_{\boldsymbol{\theta}^{(k-1)}}) \,|\, \boldsymbol{\theta}^{(k-2)}] - F_S(h_{\boldsymbol{\theta}_S^*})] + \frac{L\eta^2 \nabla_{\max}^2}{2} \quad (117)$$

Note that $\mathbb{E}[F_S(h_{\boldsymbol{\theta}^{(k)}}) \,|\, \boldsymbol{\theta}^{(k-1)}, \boldsymbol{\theta}^{(k-2)}] = \mathbb{E}[F_S(h_{\boldsymbol{\theta}^{(k)}}) \,|\, \boldsymbol{\theta}^{(k-1)}]$ because $\boldsymbol{\theta}^{(k)}$, is conditionally independent of $\boldsymbol{\theta}^{(k-2)}$ when $\boldsymbol{\theta}^{(k-1)}$ is given. Moreover, the term $\mathbb{E}[F_S(h_{\boldsymbol{\theta}^{(k-1)}}) \,|\, \boldsymbol{\theta}^{(k-2)}] - F_S(h_{\boldsymbol{\theta}_S^*})$ can be simplified in the same manner as above to get,

$$\mathbb{E}[F_S(h_{\boldsymbol{\theta}^{(k-1)}}) \,|\, \boldsymbol{\theta}^{(k-2)}] - F_S(h_{\boldsymbol{\theta}_S^*}) \leq (1 - \eta\sigma)[\mathbb{E}[F_S(h_{\boldsymbol{\theta}^{(k-2)}}) \,|\, \boldsymbol{\theta}^{(k-3)}] - F_S(h_{\boldsymbol{\theta}_S^*})] + \frac{L\eta^2 \nabla_{\max}^2}{2}$$
$$(118)$$

Let $\boldsymbol{\theta}^{(0):(\tau)}$ denote $\boldsymbol{\theta}^{(0)}, \ldots, \boldsymbol{\theta}^{(\tau)}$. Then, repeating the above procedure and simplifying yields,

$$\mathbb{E}_{\boldsymbol{\theta}^{(0):(k-1)}}[F_S(h_{\boldsymbol{\theta}^{(k-1)}})] - F_S(h_{\boldsymbol{\theta}_S^*})$$

$$\leq (1 - \eta\sigma)^k [\mathbb{E}[F_S(h_{\boldsymbol{\theta}^{(0)}})] - F_S(h_{\boldsymbol{\theta}_S^*})] + \frac{L\eta^2 \nabla_{\max}^2}{2} \sum_{\tau=1}^{k} (1 - \eta\sigma)^\tau \quad (119)$$

$$\leq (1 - \eta\sigma)^k [F_S(h_{\boldsymbol{\theta}^{(0)}}) - F_S(h_{\boldsymbol{\theta}_S^*})] + \frac{L\eta^2 \nabla_{\max}^2}{2} \frac{1}{\eta\sigma} \quad (120)$$

$$\leq \frac{(1 - \eta\sigma)^k}{\sigma} \|\nabla_{\boldsymbol{\theta}^{(0)}} F_S(h_{\boldsymbol{\theta}^{(0)}})\|^2 + \frac{L\eta \nabla_{\max}^2}{2\sigma} \quad (121)$$

$$\leq \frac{L(1 - \eta\sigma)^k}{\sigma} \|\boldsymbol{\theta}^{(0)} - \boldsymbol{\theta}_S^*\|^2 + \frac{L\eta \nabla_{\max}^2}{2\sigma} \quad (122)$$

$$\leq \frac{4L\theta_{\max}^2 (1 - \eta\sigma)^k}{\sigma} + \frac{L\eta \nabla_{\max}^2}{2\sigma} \quad (123)$$

where, in inequality (121) we have used the PL condition and in inequality (122) we have used the fact that $\|\nabla_{\boldsymbol{\theta}^{(0)}} F_S(h_{\boldsymbol{\theta}^{(0)}})\| = \|\nabla_{\boldsymbol{\theta}^{(0)}} F_S(h_{\boldsymbol{\theta}^{(0)}}) - \nabla_{\boldsymbol{\theta}_S^*} F_S(h_{\boldsymbol{\theta}_S^*})\| \leq L\|\boldsymbol{\theta}^{(0)} - \boldsymbol{\theta}_S^*\|$ (using Lipschitz gradients).

Finally, we note that by the definition of $\boldsymbol{\theta}_S^*$, $F_S(h_{\tilde{\boldsymbol{\theta}}}) \geq F_S(h_{\boldsymbol{\theta}_S^*})$. Thus, we replace $F_S(h_{\tilde{\boldsymbol{\theta}}})$ by $F_S(h_{\boldsymbol{\theta}_S^*})$ in the LHS to get the statement of the lemma. This completes the proof of the lemma. $\quad\square$

**Lemma 13.** *($k$−SGD guarantee) Suppose Assumption 1 and 2 hold and $F$ is a non-convex function that satisfies the PL condition. Let $\widehat{\boldsymbol{\theta}}_t, \widehat{S}_t$ denote the iterates of our algorithm and suppose* method *= $k$-SGD is used. Then, for any $\tilde{\boldsymbol{\theta}} \in \Theta$*

$$\mathbb{E}[F(h_{\widehat{\boldsymbol{\theta}}_{t-1}}, \widehat{S}_{t-1} \,|\, a_{\boldsymbol{\phi}^*(\widehat{\boldsymbol{\theta}}_{t-1}, \widehat{S}_{t-1})})] - \mathbb{E}[F(h_{\tilde{\boldsymbol{\theta}}}, \widehat{S}_{t-1} \,|\, a_{\boldsymbol{\phi}^*(\tilde{\boldsymbol{\theta}}, \widehat{S}_{t-1})})]$$
$$\leq 2k\eta\nabla_{\max}^2 + \frac{4L\theta_{\max}^2(1-\eta\sigma)^k}{\sigma} + \frac{L\eta\nabla_{\max}^2}{2\sigma} \quad (124)$$

*where the expectation is taken w.r.t. the total randomness of the algorithm (stochastic gradient descent and stochastic distorted greedy).*

*Proof.* In our algorithm with method = $k$-SGD $\widehat{\boldsymbol{\theta}}_t$ is derived from $\widehat{\boldsymbol{\theta}}_{t-1}$ using $k$ steps of SGD with fixed set $\widehat{S}_{t-1}$. In this case, $\widehat{\boldsymbol{\theta}}_{t-1} = \boldsymbol{\theta}^{(0)}$, $\widehat{\boldsymbol{\theta}}_t = \boldsymbol{\theta}^{(k)}$, $S = \widehat{S}_{t-1}$ and $\boldsymbol{\theta}^{(1)}, \ldots, \boldsymbol{\theta}^{(k-1)}$, denote the intermediate $k$-SGD iterates. In this case, Lemma 12 holds and we obtain:

$$\mathbb{E}[F(h_{\widehat{\boldsymbol{\theta}}_t}, \widehat{S}_{t-1} \,|\, a_{\boldsymbol{\phi}^*(\widehat{\boldsymbol{\theta}}_t, \widehat{S}_{t-1})}) \,|\, \widehat{\boldsymbol{\theta}}_{t-1}, \boldsymbol{\theta}^{(1)}, \ldots, \boldsymbol{\theta}^{(k-1)}] - F(h_{\tilde{\boldsymbol{\theta}}}, \widehat{S}_{t-1} \,|\, a_{\boldsymbol{\phi}^*(\tilde{\boldsymbol{\theta}}, \widehat{S}_{t-1})})$$
$$\leq \frac{4L\theta_{\max}^2(1-\eta\sigma)^k}{\sigma} + \frac{L\eta\nabla_{\max}^2}{2\sigma} \quad (125)$$

Taking total expectation on both sides, w.r.t. $\boldsymbol{\theta}^{(1)}, \ldots, \boldsymbol{\theta}^{(k-1)}$, using law of total expectation and observing that the other terms in the equation are independent of these random variables, we obtain:

$$\mathbb{E}[F(h_{\widehat{\boldsymbol{\theta}}_t}, \widehat{S}_{t-1} \,|\, a_{\boldsymbol{\phi}^*(\widehat{\boldsymbol{\theta}}_t, \widehat{S}_{t-1})}) \,|\, \widehat{\boldsymbol{\theta}}_{t-1}] - F(h_{\tilde{\boldsymbol{\theta}}}, \widehat{S}_{t-1} \,|\, a_{\boldsymbol{\phi}^*(\tilde{\boldsymbol{\theta}}, \widehat{S}_{t-1})}) \leq \frac{4L\theta_{\max}^2(1-\eta\sigma)^k}{\sigma} + \frac{L\eta\nabla_{\max}^2}{2\sigma}$$
$$(126)$$

Using Lipschitzness of $F$ from Assumption 1, we obtain:

$$|F(h_{\widehat{\boldsymbol{\theta}}_{t-1}}, \widehat{S}_{t-1} \,|\, a_{\boldsymbol{\phi}^*(\widehat{\boldsymbol{\theta}}_{t-1}, \widehat{S}_{t-1})}) - F(h_{\widehat{\boldsymbol{\theta}}_t}, \widehat{S}_{t-1} \,|\, a_{\boldsymbol{\phi}^*(\widehat{\boldsymbol{\theta}}_t, \widehat{S}_{t-1})})|$$
$$\leq \nabla_{\max}\|\widehat{\boldsymbol{\theta}}_{t-1} - \widehat{\boldsymbol{\theta}}_t\|$$
$$= \nabla_{\max}\|\widehat{\boldsymbol{\theta}}_{t-1} - \boldsymbol{\theta}^{(1)} + \boldsymbol{\theta}^{(1)} \ldots + \boldsymbol{\theta}^{(k-1)} - \widehat{\boldsymbol{\theta}}_t\|$$
$$\leq 2k\eta\nabla_{\max}^2 \quad (127)$$

Adding and subtracting $F(h_{\widehat{\boldsymbol{\theta}}_{t-1}}, \widehat{S}_{t-1} \,|\, a_{\boldsymbol{\phi}^*(\widehat{\boldsymbol{\theta}}_{t-1}, \widehat{S}_{t-1})})$ from inequality (126) gives us,

$$F(h_{\widehat{\boldsymbol{\theta}}_{t-1}}, \widehat{S}_{t-1} \,|\, a_{\boldsymbol{\phi}^*(\widehat{\boldsymbol{\theta}}_{t-1}, \widehat{S}_{t-1})}) - F(h_{\tilde{\boldsymbol{\theta}}}, \widehat{S}_{t-1} \,|\, a_{\boldsymbol{\phi}^*(\tilde{\boldsymbol{\theta}}, \widehat{S}_{t-1})})$$
$$\leq F(h_{\widehat{\boldsymbol{\theta}}_{t-1}}, \widehat{S}_{t-1} \,|\, a_{\boldsymbol{\phi}^*(\widehat{\boldsymbol{\theta}}_{t-1}, \widehat{S}_{t-1})}) - \mathbb{E}[F(h_{\widehat{\boldsymbol{\theta}}_{t-1}}, \widehat{S}_{t-1} \,|\, a_{\boldsymbol{\phi}^*(\widehat{\boldsymbol{\theta}}_{t-1}, \widehat{S}_{t-1})}) \,|\, \widehat{\boldsymbol{\theta}}_{t-1}]$$
$$+ \frac{4L\theta_{\max}^2(1-\eta\sigma)^k}{\sigma} + \frac{L\eta\nabla_{\max}^2}{2\sigma}$$

Now, using inequality (127), we obtain:

$$F(h_{\widehat{\boldsymbol{\theta}}_{t-1}}, \widehat{S}_{t-1} \,|\, a_{\boldsymbol{\phi}^*(\widehat{\boldsymbol{\theta}}_{t-1}, \widehat{S}_{t-1})}) - F(h_{\tilde{\boldsymbol{\theta}}}, \widehat{S}_{t-1} \,|\, a_{\boldsymbol{\phi}^*(\tilde{\boldsymbol{\theta}}, \widehat{S}_{t-1})})$$
$$\leq 2k\eta\nabla_{\max}^2 + \frac{4L\theta_{\max}^2(1-\eta\sigma)^k}{\sigma} + \frac{L\eta\nabla_{\max}^2}{2\sigma} \quad (128)$$

Finally, taking expectation w.r.t. the total randomness of the algorithm (stochastic gradient descent and stochastic distorted greedy), we obtain:

$$\mathbb{E}[F(h_{\widehat{\boldsymbol{\theta}}_{t-1}}, \widehat{S}_{t-1} \,|\, a_{\boldsymbol{\phi}^*(\widehat{\boldsymbol{\theta}}_{t-1}, \widehat{S}_{t-1})})] - \mathbb{E}[F(h_{\tilde{\boldsymbol{\theta}}}, \widehat{S}_{t-1} \,|\, a_{\boldsymbol{\phi}^*(\tilde{\boldsymbol{\theta}}, \widehat{S}_{t-1})})]$$
$$\leq 2k\eta\nabla_{\max}^2 + \frac{4L\theta_{\max}^2(1-\eta\sigma)^k}{\sigma} + \frac{L\eta\nabla_{\max}^2}{2\sigma} \quad (129)$$

$\square$

**Lemma 14.** *Suppose the assumptions of Lemma 8 and Theorem 2 hold. Let $\kappa = \mu C_{\min}/(\lambda\theta_{\max}^2 + \rho\ell_{\max} + \mu C_{\min})$, where the symbols are the same as defined in the statement of Lemma 8 and $\gamma^*$ is defined in Theorem 2.*

$$\left(1 - \frac{e^{-\gamma^*} + \delta}{\kappa}\right) F(h_{\widehat{\boldsymbol{\theta}}_t}, S \,|\, a_{\boldsymbol{\phi}^*(\widehat{\boldsymbol{\theta}}_t, S)}) - \mathbb{E}[F(h_{\widehat{\boldsymbol{\theta}}_t}, \widehat{S}_t \,|\, a_{\boldsymbol{\phi}^*(\widehat{\boldsymbol{\theta}}_t, \widehat{S}_t)})] \leq 4k\eta\nabla_{\max}^2 \quad (130)$$

*where the expectation is w.r.t. the randomness in Stochastic Distorted Greedy.*

*Proof.* To obtain our guarantees, we require $(1 - \frac{e^{-\gamma^*}+\delta}{\kappa}) \geq 0$. Substituting $\kappa = \mu C_{\min}/(\lambda\theta_{\max}^2 + \rho\ell_{\max} + \mu C_{\min})$ and simplifying gives us $\rho < [\mu C_{\min}((e^{-\gamma^*}+\delta)^{-1}-1) - \lambda\theta_{\max}^2]/\ell_{\max}$. To ensure that the this bound is positive, we require $\delta < (1 - e^{-\gamma^*})$.

In our algorithm, we obtain $\widehat{S}_t$ by applying Stochastic Distorted Greedy using $\widehat{\boldsymbol{\theta}}_{t-1}$. Therefore, by our characterization of $F$ as stated in Eq. (6) and using Lemma 7, we have that for any $S$,

$$F(h_{\widehat{\boldsymbol{\theta}}_{t-1}}, S \,|\, a_{\boldsymbol{\phi}^*(\widehat{\boldsymbol{\theta}}_{t-1}, S)}) \leq \left(\frac{\kappa}{\kappa - e^{-\gamma^*} - \delta}\right) \mathbb{E}[F(h_{\widehat{\boldsymbol{\theta}}_{t-1}}, \widehat{S}_t \,|\, a_{\boldsymbol{\phi}^*(\widehat{\boldsymbol{\theta}}_{t-1}, \widehat{S}_t)})] \tag{131}$$

Since $\widehat{\boldsymbol{\theta}}_t$ is derived from $\widehat{\boldsymbol{\theta}}_{t-1}$ using `method = k-SGD`, we cannot use Lipschitzness directly. Instead, we use inequality (127). Doing so, we get that for all $t, S$,

$$F(h_{\widehat{\boldsymbol{\theta}}_t}, S \,|\, a_{\boldsymbol{\phi}^*(\widehat{\boldsymbol{\theta}}_t, S)}) \leq F(h_{\widehat{\boldsymbol{\theta}}_{t-1}}, S \,|\, a_{\boldsymbol{\phi}^*(\widehat{\boldsymbol{\theta}}_{t-1}, S)}) + 2k\eta\nabla_{\max}^2 \tag{132}$$

$$\leq \left(\frac{\kappa}{\kappa - e^{-\gamma^*} - \delta}\right) \mathbb{E}[F(h_{\widehat{\boldsymbol{\theta}}_{t-1}}, \widehat{S}_t \,|\, a_{\boldsymbol{\phi}^*(\widehat{\boldsymbol{\theta}}_{t-1}, \widehat{S}_t)})] + 2k\eta\nabla_{\max}^2 \tag{133}$$

$$\left(1 - \frac{e^{-\gamma^*}+\delta}{\kappa}\right) F(h_{\widehat{\boldsymbol{\theta}}_t}, S \,|\, a_{\boldsymbol{\phi}^*(\widehat{\boldsymbol{\theta}}_t, S)}) \leq \mathbb{E}[F(h_{\widehat{\boldsymbol{\theta}}_{t-1}}, \widehat{S}_t \,|\, a_{\boldsymbol{\phi}^*(\widehat{\boldsymbol{\theta}}_{t-1}, \widehat{S}_t)})] + \left(1 - \frac{e^{-\gamma^*}+\delta}{\kappa}\right) 2k\eta\nabla_{\max}^2 \tag{134}$$

where in inequality (133) we use inequality (131). Subtracting $\mathbb{E}[F(h_{\widehat{\boldsymbol{\theta}}_t}, \widehat{S}_t \,|\, a_{\boldsymbol{\phi}^*(\widehat{\boldsymbol{\theta}}_t, \widehat{S}_t)})]$ from both sides, we obtain:

$$\left(1 - \frac{e^{-\gamma^*}+\delta}{\kappa}\right) F(h_{\widehat{\boldsymbol{\theta}}_t}, S \,|\, a_{\boldsymbol{\phi}^*(\widehat{\boldsymbol{\theta}}_t, S)}) - \mathbb{E}[F(h_{\widehat{\boldsymbol{\theta}}_t}, \widehat{S}_t \,|\, a_{\boldsymbol{\phi}^*(\widehat{\boldsymbol{\theta}}_t, \widehat{S}_t)})]$$

$$\leq \mathbb{E}[F(h_{\widehat{\boldsymbol{\theta}}_{t-1}}, \widehat{S}_t \,|\, a_{\boldsymbol{\phi}^*(\widehat{\boldsymbol{\theta}}_{t-1}, \widehat{S}_t)}) - F(h_{\widehat{\boldsymbol{\theta}}_t}, \widehat{S}_t \,|\, a_{\boldsymbol{\phi}^*(\widehat{\boldsymbol{\theta}}_t, \widehat{S}_t)})] + \left(1 - \frac{e^{-\gamma^*}+\delta}{\kappa}\right) 2k\eta\nabla_{\max}^2 \tag{135}$$

Using Lipschitzness of $F$ from Assumption 1, we obtain:

$$|F(h_{\widehat{\boldsymbol{\theta}}_{t-1}}, \widehat{S}_{t-1} \,|\, a_{\boldsymbol{\phi}^*(\widehat{\boldsymbol{\theta}}_{t-1}, \widehat{S}_{t-1})}) - F(h_{\widehat{\boldsymbol{\theta}}_t}, \widehat{S}_{t-1} \,|\, a_{\boldsymbol{\phi}^*(\widehat{\boldsymbol{\theta}}_t, \widehat{S}_{t-1})})|$$

$$\leq \nabla_{\max}\|\widehat{\boldsymbol{\theta}}_{t-1} - \widehat{\boldsymbol{\theta}}_t\|$$

$$= \nabla_{\max}\|\widehat{\boldsymbol{\theta}}_{t-1} - \boldsymbol{\theta}^{(1)} + \boldsymbol{\theta}^{(1)} \ldots + \boldsymbol{\theta}^{(k-1)} - \widehat{\boldsymbol{\theta}}_t\|$$

$$\leq 2k\eta\nabla_{\max}^2 \tag{136}$$

We use inequality (136) into the inequality (135) to have:

$$\left(1 - \frac{e^{-\gamma^*}+\delta}{\kappa}\right) F(h_{\widehat{\boldsymbol{\theta}}_t}, S \,|\, a_{\boldsymbol{\phi}^*(\widehat{\boldsymbol{\theta}}_t, S)}) - \mathbb{E}[F(h_{\widehat{\boldsymbol{\theta}}_t}, \widehat{S}_t \,|\, a_{\boldsymbol{\phi}^*(\widehat{\boldsymbol{\theta}}_t, \widehat{S}_t)})]$$

$$\leq 2k\eta\nabla_{\max}^2 + \left(1 - \frac{e^{-\gamma^*}+\delta}{\kappa}\right) 2k\eta\nabla_{\max}^2$$

$$= \left(2 - \frac{e^{-\gamma^*}+\delta}{\kappa}\right) 2k\eta\nabla_{\max}^2$$

$$\leq 4k\eta\nabla_{\max}^2 \tag{137}$$

$\square$

# G ADDITIONAL DETAILS ABOUT THE EXPERIMENTAL SETUP

## G.1 DATASET SPLIT INTO TRAIN, VALIDATION AND TEST

We split the datasets[1] into training, validation, and test set in the ratio of 4:1:1, 5:1:1 and 4:1:1 for CIFAR10, FMNIST and CIFAR100 respectively. For CIFAR10, we use $40000$ training examples, $10000$ validation examples and $10000$ test examples. For Fashion MNIST, we use $50000$ training examples, $10000$ validation examples, and $10000$ test examples. For CIFAR100, we use $40000$ training examples, $10000$ validation examples and $10000$ test examples. The same train, validation and test split it used for the baselines as well. In all cases, unless otherwise mentioned, we consider $|S| \leq b = 0.1|D|$ during training. Similarly, during test we use $10\%$ test instances for attack. The exact nature of drawing this $10\%$ test instances varies across experiments and is mentioned therein.

## G.2 DETAILS ABOUT THE BASELINES

In all baselines, we used ResNet18 architecture for CIFAR10 and ResNet9 for CIFAR100, with the last layers having 10 and 100 neurons respectively. For FMNIST, we use LeNet architecture. For all the methods (including ours) which use PGD attack during training, we keep the PGD attack parameters to be the same as ROGET ($a_\phi = $ PGD) (details in the following subsection). Similar to $\rho$ in our method, GAT, TRADES, Nu-AT and, MART also offer specific hyperparameters which can control the tradeoff between $\mathcal{A}_{\text{clean}}$ and $\mathcal{A}_{\text{robust}}$. For all methods (including Ours), we use PGD attack as the assumed adversarial perturbation method during hyperparameter selection on the validation set.

**GAT (Sriramanan et al., 2020).** We used the code from the official repository[2]. They provide two different codebases for CIFAR10 and MNIST. We used their MNIST code for experiments with FMNIST dataset. For default hyperparameter selection (Table 1), we refer to the official repository[3] in which they provide the value of $l2\_reg = 10$ for CIFAR10 and $l2\_reg = 15$ for MNIST (which we use for FMNIST). For worst-case hyperparameter selection (Table 2), we train GAT on a range of $l2\_reg$ values: $\{2.5, 5.0, 10.0, 15.0, 20.0, 30.0\}$ for CIFAR10 and FMNIST, and $\{2.0, 5.0, 10.0, 20.0, 30.0\}$ for CIFAR100. In their official code, they train GAT for 100 epochs on CIFAR10 and for 50 epochs on MNIST. Hence we train GAT for 100, 50 and 100 epochs on CIFAR10, FMNIST and CIFAR100 respectively.

**FBF (Wong et al., 2019).** For CIFAR10 we used the code from the official repository[4]. For FMNIST and CIFAR100, we implemented their code parallel to CIFAR10. The only changes (other than the architecture) were the mean and standard deviation that we computed for FMNIST and CIFAR100 separately. FBF does not have any tunable parameter and hence it does not undergo any hyperparameter selection. For CIFAR10 and CIFAR100, we train FBF for 80 epochs (as used in the official code for CIFAR10). For FMNIST, we train FBF for 10 epochs (as used in the official code for MNIST).

**TRADES (Zhang et al., 2019b).** We used the code from the official repository[5]. They provide two different codebases for CIFAR10 and MNIST. We used their MNIST code for experiments with FMNIST dataset. For default hyperparameter selection (Table 1), we refer to the official repository[6] in which they use $\beta = 6.0$ for CIFAR10 and $\beta = 1.0$ for MNIST (which we use for FMNIST). For worst-case hyperparameter selection (Table 2), we train TRADES on a range of $\beta$ values: $\{0.1, 0.2, 0.4, 0.6, 0.8, 2.0, 4.0, 6.0\}$ for CIFAR10, $\{0.4, 0.6, 0.8, 1.0, 2.0, 4.0, 6.0\}$ for FMNIST, and $\{1.0, 2.0, 4.0, 6.0, 8.0\}$ for CIFAR100. Optimizer, batch size and learning rate are same as those used in the official repository. We train TRADES for 120, 100, and 120 epochs on CIFAR10, FMNIST and, CIFAR100 respectively.

---

[1]We collect the datasets from https://www.cs.toronto.edu/ kriz/cifar.html (CIFAR-10), and https://www.kaggle.com/datasets/zalando-research/fashionmnist (FMNIST).

[2]https://github.com/val-iisc/GAMA-GAT/

[3]https://github.com/val-iisc/GAMA-GAT/

[4]https://github.com/locuslab/fast_adversarial/

[5]https://github.com/yaodongyu/TRADES/

[6]https://github.com/yaodongyu/TRADES/

**Nu-AT (Sriramanan et al., 2021).** We used the code from the official repository[7]. They only provide code for CIFAR10 hence for running it on FMNIST, we modify the PGD parameters to the one we used in FMNIST and the number of epochs to 100. For default hyperparameter selection (Table 1), we refer to the supplementary material of Nu-AT, in which the authors mention that "We use a $\lambda_{max}$ of 4.5 for CIFAR-10 on ResNet-18 architecture and 4 for WideResNet-34-10. For MNIST we use $\lambda_{max}$ of 1...". Hence we use $\lambda_{max}$=4.5 for CIFAR10 and $\lambda_{max}$=1.0 for FMNIST. For worst-case hyperparameter selection (Table 2), we train Nu-AT on a range of $\lambda_{max}$ values: {2.0, 2.5, 3.0, 3.5, 4.0, 4.5, 5.0} for CIFAR10 and FMNIST, and {1.0, 2.0, 4.5, 6.0, 8.0} for CIFAR100. Optimizer, batch size and learning rate are same as those used in the official repository. We train Nu-AT for 120, 100, and 120 epochs on CIFAR10, FMNIST and, CIFAR100 respectively.

**MART (Wang et al., 2019).** We used the code from the official repository[8]. They only provide code for CIFAR10 hence for running it on FMNIST, we modify the PGD parameters and the number of epochs. For default hyperparameter selection (Table 1), we refer to the official repository[9] in which they use $\beta = 5.0$ for CIFAR10. We were not able to find any mention of their hyperparameter values for MNIST either in their paper or their code. Hence using Figure 2(d) of their paper as reference, we trained MART on FMNIST for $\beta =$ {0.5, 1.0, 2.5, 5.0, 7.5, 10.0} and found that only $\beta = 1.0$ undergone effective training and gave $\mathcal{A}_{clean}$ above 50%. Hence we chose $\beta = 1.0$ as the default value for FMNIST. For worst-case hyperparameter selection (Table 2), we train MART on a range of $\beta$ values: {0.5, 1.0, 2.5, 5.0, 7.5, 10.0} for CIFAR10 and FMNIST, and {0.5, 1.0, 2.5, 5.0, 10.0} for CIFAR100. We train MART for 120, 100, and 120 epochs on CIFAR10, FMNIST and, CIFAR100 respectively.

**PGD-AT (Madry et al., 2017).** We could not find any official Pytorch implementation for PGD-AT. Therefore, we implemented it ourselves using the architecture mentioned above, for each dataset. We use SGD optimizer along with batch size of 128 was used for both datasets. For FMNIST, initial learning rate 0.01 and momentum 0.9 was used. For CIFAR10 and CIFAR100, we use initial learning rate of 0.1 and momentum 0.9. PGD-AT has no tunable parameter and hence it undergoes no hyperparameter selection. We train PGD-AT for 100 epochs on all three datasets.

**RFGSM-AT (Tramèr et al., 2018).** We could not find any official Pytorch implementation for RFGSM-AT. Therefore, we implemented it ourselves using the architecture mentioned above, for each dataset. RFGSM-AT also does not have any tunable parameter that controls the tradeoff between $\mathcal{A}_{robust}$ and $\mathcal{A}_{clean}$ but it has a parameter $\alpha$ which does affect its robust accuracy to a significant extent. We refer to their paper in which they use $\alpha = \epsilon/2$ for ImageNet and MNIST. Hence, for default hyperparameter selection (Table 1), we use $\alpha = \epsilon/2$ for CIFAR10 and FMNIST. This value however gives very low robust accuracy for all the attacks. Hence for worst-case hyperparameter selection (Table 2), we tune $\alpha$ such that it achieves $\mathcal{A}_{robust}$ above 40% on PGD attack which gives us the value of $\alpha = 0.05\epsilon$. Optimizer, batch size and learning rate are same as those used in the official repository. We use SGD optimizer along with batch size of 128 was used for both datasets. For FMNIST, initial learning rate 0.01 and momentum 0.9 was used. For CIFAR10 and CIFAR100, we use initial learning rate of 0.1 and momentum 0.9. We train RFGSM-AT for 100 epochs on all three datasets.

### G.3 Implementation Details of our method

**Hyperparameters of our method.** For default hyperparameter selection (Table 1), we simply use $\rho = 1$ for all the datasets. For worst-case hyperparameter selection (Table 2), we train ROGET ($a_\phi$ = PGD) on a range of $\rho$ values: {0.01, 0.05, 0.5, 1.0, 2.0, 5.0, 8.0, 10.0} for CIFAR10, {0.01, 0.1, 0.5, 1.0, 1.5, 2.0, 4.0, 8.0} for FMNIST, and {2.0, 5.0, 10.0, 20.0, 30.0} for CIFAR100. For ROGET ($a_\phi$ = AdvGAN), we train for $\rho$ values: {0.01, 0.25, 0.5, 1.0, 1.5, 2.0} for CIFAR10, {0.001, 0.005, 0.01, 0.1, 0.5, 1.0, 1.5, 2.0} for FMNIST and {0.005, 0.01, 0.25, 0.5, 1.0} for CIFAR100. Batch size was set to 128 in all the datasets. We use $k-$SGD to train our method. Moreover, we use early stopping as follows. While running multiple training epochs for a fixed $S$ we use early stopping to determine when to stop. In $k-$SGD, we stop training for the current $S$ if on the validation set, the attacked set accuracy, $\mathcal{A}_{robust}$, drops.

---

[7]https://github.com/val-iisc/NuAT/

[8]https://github.com/YisenWang/MART

[9]https://github.com/YisenWang/MART

**Details about $a_\phi$.** For ROGET ($a_\phi$ = PGD), we set $\epsilon = 0.031$, number of steps = 20 and step size = 0.007 for CIFAR10 and CIFAR100 while for FMNIST, we use $\epsilon = 0.3$, number of steps = 40 and step size = 0.01.

For ROGET ($a_\phi$ = AdvGAN), we used the Pytorch implementation available in an online repository[10]. The same architecture was used for all the datasets with only changes to the number of input channels. No changes to the generator or discriminator architecture were made besides the number of input channels which was set to 1 for FMNIST and 3 for CIFAR10 and CIFAR100. At the end of every iteration, we retrain AdvGAN on set $S$ output by stochastic distorted greedy algorithm, for 15 epochs. Retraining AdvGAN every time we add a point to the set $S$ in stochastic distorted greedy (SDG) algorithm increases the running time of SDG to such an extent that it becomes infeasible. For this reason, we instead choose to retrain AdvGAN after SDG outputs a set $S$, for the current iteration. This is equivalent to fine-tuning AdvGAN to attack the set $S$, output by SDG.

### G.4 DETAILS ABOUT ADVERSARIAL PERTURBATION

For PGD attack, we used same specifications used to train ROGET ($a_\phi$ = PGD). More specifically, we set $\epsilon = 0.031$, number of steps = 20 and step size = 0.007 for CIFAR10 and CIFAR100 while for FMNIST, we use $\epsilon = 0.3$, number of steps = 40 and step size = 0.01. For Auto Attack, we use the standard version which consists of untargeted APGD-CE, targeted APGD-DLR, targeted FAB and Square attacks each with the default parameters. For square attack, we set number of queries to 1000 and $\epsilon$ same as that for PGD attack above. For applying MI-FGSM in black box setting, we take a source model trained on the chosen dataset, get the perturbed sample by computing gradients using the source model and then test the method on this obtained perturbed sample. Transfer based black box attacks are weaker than white-box attacks. Hence we set the parameters of MI-FGSM to be stronger than those for PGD attack. We set $\epsilon = 0.2$, number of steps = 60, step size = 0.007 for CIFAR10 and CIFAR100 and $\epsilon = 0.305$, number of steps = 80, step size = 0.01 for FMNIST. For Square attack, we use $\ell_\infty$ norm with $\epsilon = 0.031$ for CIFAR10 and $\epsilon = 0.3$ for FMNIST.

### G.5 INFRASTRUCTURE DETAILS

We implement ROGET using Python 3.8 and PyTorch 1.10.1. The experiments were run on servers equipped with 2.9GHz CPUs, NVIDIA Quadro (48GB), NVIDIA RTX A6000 (48 GB), NVIDIA A40 (46 GB), NVIDIA Quadro RTX 8000 (49 GB) and NVIDIA TITAN RTX (24 GB) GPUs.

### G.6 LICENSE

We collect the datasets from https://www.cs.toronto.edu/~kriz/cifar.html (CIFAR10 and CIFAR100), and https://www.kaggle.com/datasets/zalando-research/fashionmnist (Fashion-MNIST). These sources allow the use of datasets for research purposes. Furthermore, we use the following publicly available repositories— https://github.com/val-iisc/GAMA-GAT/ (GAT), https://github.com/locuslab/fast_adversarial/ (FBF), https://github.com/yaodongyu/TRADES/ (TRADES), https://github.com/val-iisc/NuAT/ (Nu-AT), https://github.com/YisenWang/MART (MART) to implement the baselines, https://github.com/mathcbc/advGAN_pytorch (AdvGAN) to implement AdvGAN, and https://submodlib.readthedocs.io/en/latest/functions/facilityLocation.html to solve the Facility Location problem in one of our additional experiments mentioned in section H below.

---

[10]https://github.com/mathcbc/advGAN_pytorch

# H  ADDITIONAL EXPERIMENTS

## H.1  RESULTS ON LOSS BASED HYPERPARAMETER SELECTION TECHNIQUE ON CIFAR10 AND FMNIST

We also explore another hyperparameter selection technique in which the learner assumes a subset selection strategy of the adversary. More specifically, the learner assumes a distribution across validation samples which is inversely proportional to the loss of a classifier trained on clean samples. Based on this strategy an attack is simulated on the validation set and the hyperparameter giving the best overall accuracy is selected. We present the results for CIFAR10 and FMNIST in Table 7 For CIFAR10, we observe that our method achieves the highest overall accuracy for all attacks. For FMNIST, TRADES has a better overall accuracy on AA and Square attack by a small margin.

| | CIFAR10 | | | | | | | | | | |
|---|---|---|---|---|---|---|---|---|---|---|---|
| | | PGD | | AA | | Square | | MIFGSM | | AdvGAN | |
| | $\mathcal{A}_{\text{clean}}$ | $\mathcal{A}_{\text{robust}}$ | $\mathcal{A}$ | $\mathcal{A}_{\text{robust}}$ | $\mathcal{A}$ | $\mathcal{A}_{\text{robust}}$ | $\mathcal{A}$ | $\mathcal{A}_{\text{robust}}$ | $\mathcal{A}$ | $\mathcal{A}_{\text{robust}}$ | $\mathcal{A}$ |
| GAT | 77.55 | 63.26 | 76.12 | 49.73 | 74.77 | 60.07 | 75.80 | 52.91 | 75.09 | 81.07 | 77.91 |
| FBF | 74.92 | 72.04 | 74.64 | 31.10 | 70.54 | 31.59 | 70.59 | 36.09 | 71.04 | 37.22 | 71.15 |
| TRADES | 83.63 | 51.43 | 80.41 | 42.55 | 79.52 | 58.60 | 81.13 | 48.00 | 80.07 | 87.37 | 84.02 |
| Nu-AT | 84.66 | 54.64 | 81.66 | 45.79 | 80.77 | 61.34 | 82.33 | 47.41 | 80.93 | 88.18 | 85.01 |
| MART | 80.49 | 61.30 | 78.57 | 49.32 | 77.37 | 61.26 | 78.57 | 51.13 | 77.56 | 84.25 | 80.87 |
| PGD-AT | 83.36 | 57.33 | 80.76 | 48.50 | 79.88 | 62.71 | 81.30 | 50.00 | 80.03 | 87.13 | 83.74 |
| RFGSM-AT | 85.84 | 45.94 | 81.85 | 37.49 | 81.00 | 60.25 | 83.28 | 37.91 | 81.04 | 89.42 | 86.20 |
| ROGET (PGD) | 86.63 | 51.16 | 83.09 | 43.06 | 82.28 | 62.33 | 84.20 | 42.20 | 82.19 | 90.15 | 86.99 |
| ROGET (AdvGAN) | 87.86 | 47.97 | 83.87 | 39.97 | 83.07 | 62.21 | 85.29 | 40.27 | 83.10 | 91.53 | 88.22 |
| | FMNIST | | | | | | | | | | |
| | | PGD | | AA | | Square | | MIFGSM | | AdvGAN | |
| | $\mathcal{A}_{\text{clean}}$ | $\mathcal{A}_{\text{robust}}$ | $\mathcal{A}$ | $\mathcal{A}_{\text{robust}}$ | $\mathcal{A}$ | $\mathcal{A}_{\text{robust}}$ | $\mathcal{A}$ | $\mathcal{A}_{\text{robust}}$ | $\mathcal{A}$ | $\mathcal{A}_{\text{robust}}$ | $\mathcal{A}$ |
| GAT | 91.47 | 3.18 | 82.64 | 0.00 | 82.32 | 0.01 | 82.32 | 8.32 | 83.15 | 29.09 | 85.24 |
| FBF | 80.47 | 14.29 | 73.85 | 66.74 | 79.10 | 68.43 | 79.27 | 84.81 | 80.90 | 85.94 | 81.02 |
| TRADES | 86.85 | 62.90 | 84.45 | 53.37 | 83.50 | 57.37 | 83.90 | 81.43 | 86.30 | 70.42 | 85.20 |
| Nu-AT | 90.21 | 5.11 | 81.70 | 0.00 | 81.19 | 0.11 | 81.20 | 19.77 | 83.17 | 11.35 | 82.32 |
| MART | 78.94 | 77.72 | 78.82 | 59.73 | 77.02 | 61.46 | 77.19 | 79.81 | 79.03 | 70.10 | 78.06 |
| PGD-AT | 83.77 | 79.62 | 83.36 | 53.62 | 80.76 | 57.05 | 81.10 | 83.73 | 83.77 | 72.88 | 82.69 |
| RFGSM-AT | 89.46 | 13.49 | 81.86 | 0.00 | 80.51 | 0.02 | 80.51 | 24.88 | 83.00 | 21.85 | 82.70 |
| ROGET (PGD) | 87.01 | 63.24 | 84.63 | 49.63 | 83.27 | 54.29 | 83.74 | 80.92 | 86.40 | 69.16 | 85.22 |
| ROGET (AdvGAN) | 87.08 | 49.62 | 83.33 | 47.32 | 83.10 | 51.99 | 83.57 | 82.49 | 86.62 | 73.44 | 85.71 |

Table 7: Performance comparison under *Loss based hyperparameter selection*. Here, the adversary adopts uncertainty based subset selection to perform attack, where the true subset chosen for attack $S^{\text{latent}}$ consists of top 10% test instances in terms of the uncertainty of a classifier trained on all the clean examples. Numbers in green (yellow) indicate the best (second best) performers.

## H.2 EVALUATION ON LABEL BASED SUBSET SELECTION STRATEGY

We present the complete set of results for label based subset selection strategy on CIFAR10 in Table 8 and 9. Note that we use the worst case hyperparameter setting for all the methods. We observe that our method achieves the highest overall accuracy for all classes and for all attacks.

| | PGD | | | | | | | | | |
|---|---|---|---|---|---|---|---|---|---|---|
| | Airplane | Automobile | Bird | Cat | Deer | Dog | Frog | Horse | Ship | Truck |
| GAT | 76.37 | 76.17 | 76.71 | 76.36 | 75.46 | 76.69 | 75.48 | 76.18 | 75.71 | 76.19 |
| FBF | 74.43 | 74.66 | 74.59 | 74.96 | 74.54 | 74.51 | 74.40 | 74.91 | 74.48 | 74.64 |
| TRADES | 80.92 | 81.30 | 80.25 | 80.06 | 78.85 | 81.12 | 79.45 | 81.03 | 80.44 | 81.04 |
| Nu-AT | 82.12 | 82.63 | 81.50 | 80.96 | 80.22 | 81.94 | 80.71 | 82.36 | 81.61 | 82.36 |
| MART | 78.49 | 79.02 | 78.68 | 78.45 | 78.15 | 78.91 | 78.09 | 78.99 | 77.96 | 78.73 |
| PGD-AT | 81.11 | 81.53 | 80.53 | 80.88 | 79.17 | 80.98 | 80.17 | 81.03 | 80.97 | 81.03 |
| RFGSM-AT | 82.45 | 83.19 | 81.72 | 81.63 | 80.48 | 82.08 | 80.39 | 82.31 | 82.06 | 82.33 |
| ROGET ($a_\phi$ = PGD) | 83.76 | 84.68 | 82.94 | 82.84 | 80.73 | 83.63 | 81.38 | 83.53 | 83.16 | 84.23 |
| ROGET ($a_\phi$ = AdvGAN) | 84.98 | 86.02 | 83.41 | 83.70 | 81.32 | 84.30 | 82.01 | 84.40 | 83.86 | 84.87 |
| | AA | | | | | | | | | |
| | Airplane | Automobile | Bird | Cat | Deer | Dog | Frog | Horse | Ship | Truck |
| GAT | 75.19 | 75.41 | 74.76 | 73.76 | 72.92 | 75.06 | 74.20 | 75.21 | 74.85 | 75.28 |
| FBF | 72.39 | 66.43 | 74.21 | 71.38 | 72.71 | 71.77 | 68.54 | 69.59 | 73.29 | 67.11 |
| TRADES | 80.17 | 80.79 | 78.75 | 78.31 | 77.47 | 79.52 | 78.41 | 80.14 | 79.94 | 80.44 |
| Nu-AT | 81.32 | 82.06 | 79.97 | 79.36 | 78.76 | 80.54 | 79.71 | 81.62 | 81.00 | 81.68 |
| MART | 77.69 | 78.43 | 76.91 | 76.31 | 75.57 | 77.34 | 76.85 | 78.01 | 77.17 | 78.02 |
| PGD-AT | 80.38 | 80.89 | 79.15 | 79.08 | 77.83 | 79.60 | 79.15 | 80.31 | 80.28 | 80.45 |
| RFGSM-AT | 81.80 | 82.58 | 80.37 | 80.08 | 79.19 | 80.84 | 79.54 | 81.53 | 81.42 | 81.46 |
| ROGET ($a_\phi$ = PGD) | 82.97 | 84.19 | 81.48 | 81.07 | 79.73 | 82.05 | 80.67 | 82.79 | 82.49 | 83.68 |
| ROGET ($a_\phi$ = AdvGAN) | 84.24 | 85.63 | 82.01 | 81.96 | 80.45 | 82.93 | 81.14 | 83.71 | 83.19 | 84.03 |

Table 8: $\mathcal{A}$ on label based subset selection strategy on CIFAR10 under white box (PGD, AA) attacks. Here, the attacked subset selection is based on the uncertainty of a vanilla classifier ($h_v$) on the test samples. For all the methods, we perform worst-case hyperparameter selection.

| | Square | | | | | | | | | |
|---|---|---|---|---|---|---|---|---|---|---|
| | Airplane | Automobile | Bird | Cat | Deer | Dog | Frog | Horse | Ship | Truck |
| GAT | 76.06 | 76.39 | 75.42 | 74.80 | 74.15 | 75.98 | 75.48 | 76.20 | 75.98 | 76.27 |
| FBF | 72.44 | 66.44 | 74.34 | 71.44 | 72.75 | 71.80 | 68.59 | 69.65 | 73.35 | 67.12 |
| TRADES | 81.39 | 82.23 | 80.26 | 79.44 | 79.16 | 80.90 | 80.60 | 81.70 | 81.65 | 82.05 |
| Nu-AT | 82.78 | 83.28 | 81.52 | 80.63 | 80.31 | 81.83 | 81.67 | 83.06 | 82.81 | 83.23 |
| MART | 78.78 | 79.36 | 78.01 | 77.33 | 76.84 | 78.40 | 78.19 | 79.10 | 78.75 | 79.32 |
| PGD-AT | 81.57 | 82.02 | 80.39 | 80.01 | 79.62 | 80.90 | 81.04 | 81.71 | 81.79 | 81.86 |
| RFGSM-AT | 83.49 | 84.39 | 82.15 | 81.74 | 81.64 | 82.59 | 83.07 | 83.51 | 83.95 | 83.90 |
| ROGET ($a_\phi$ = PGD) | 84.42 | 85.57 | 83.14 | 82.55 | 81.96 | 83.51 | 83.47 | 84.67 | 84.77 | 85.47 |
| ROGET ($a_\phi$ = AdvGAN) | 85.92 | 87.01 | 83.88 | 83.61 | 82.96 | 84.70 | 84.00 | 85.95 | 85.79 | 86.33 |
| | MIFGSM | | | | | | | | | |
| | Airplane | Automobile | Bird | Cat | Deer | Dog | Frog | Horse | Ship | Truck |
| GAT | 74.43 | 74.64 | 75.33 | 74.58 | 73.03 | 75.40 | 77.23 | 75.76 | 74.23 | 76.42 |
| FBF | 72.65 | 66.42 | 76.14 | 71.73 | 74.71 | 71.92 | 68.81 | 69.81 | 73.52 | 67.10 |
| TRADES | 79.60 | 79.97 | 79.49 | 79.12 | 78.26 | 79.50 | 82.37 | 81.42 | 78.63 | 82.33 |
| Nu-AT | 80.19 | 80.56 | 80.91 | 79.74 | 79.67 | 80.62 | 83.33 | 82.36 | 79.25 | 82.75 |
| MART | 76.98 | 77.26 | 77.35 | 76.94 | 76.23 | 77.40 | 80.54 | 78.29 | 75.78 | 78.98 |
| PGD-AT | 79.72 | 79.90 | 80.99 | 79.34 | 78.32 | 78.80 | 82.72 | 80.49 | 78.69 | 81.36 |
| RFGSM-AT | 80.35 | 80.58 | 81.72 | 80.60 | 79.79 | 79.99 | 84.34 | 81.72 | 80.28 | 81.74 |
| ROGET ($a_\phi$ = PGD) | 81.27 | 82.37 | 82.56 | 81.42 | 80.35 | 80.97 | 85.20 | 83.04 | 81.20 | 83.88 |
| ROGET ($a_\phi$ = AdvGAN) | 82.53 | 83.57 | 83.58 | 82.21 | 80.93 | 81.87 | 85.59 | 84.31 | 81.88 | 84.64 |

Table 9: $\mathcal{A}$ on label based subset selection strategy on CIFAR10 under black box (Square, MIFGSM) attacks. Here, the attacked subset selection is based on the uncertainty of a vanilla classifier ($h_v$) on the test samples. For all the methods, we perform worst-case hyperparameter selection.

### H.3 DISCLOSING THE SUBSET SELECTION STRATEGY TO THE BASELINES

In this experiment, we reveal the advsersary's subset selection strategy (uncertainty based) to the baselines during hyperparameter selection. We select the hyperparameter of the baseline which has the best overall accuracy on the validation set using the revealed subset selection strategy. The results are presented in Table 10. Comparing with results in Table 2, we see that GAT, TRADES, Nu-AT and MART have improved *i.e.*, all the baselines which had a tunable hyperparameter have become better. More importantly, our method still achieves the best overall accuracy across all attacks except PGD.

| CIFAR10 | | PGD | | AA | | Square | | MIFGSM | | AdvGAN | |
|---|---|---|---|---|---|---|---|---|---|---|---|
| | $\mathcal{A}_{\text{clean}}$ | $\mathcal{A}_{\text{robust}}$ | $\mathcal{A}$ | $\mathcal{A}_{\text{robust}}$ | $\mathcal{A}$ | $\mathcal{A}_{\text{robust}}$ | $\mathcal{A}$ | $\mathcal{A}_{\text{robust}}$ | $\mathcal{A}$ | $\mathcal{A}_{\text{robust}}$ | $\mathcal{A}$ |
| GAT | 86.18 | 35.68 | 81.13 | 11.80 | 78.74 | 48.92 | 82.45 | 38.47 | 81.41 | 89.66 | 86.52 |
| FBF | 74.92 | 72.04 | 74.64 | 31.10 | 70.54 | 31.59 | 70.59 | 36.09 | 71.04 | 37.22 | 71.15 |
| TRADES | 84.70 | 50.98 | 81.33 | 39.58 | 80.19 | 60.29 | 82.26 | 41.97 | 80.43 | 88.27 | 85.06 |
| Nu-AT | 87.88 | 47.54 | 83.85 | 18.87 | 80.98 | 53.01 | 84.40 | 39.15 | 83.01 | 91.28 | 88.22 |
| MART | 82.12 | 60.29 | 79.94 | 48.73 | 78.78 | 61.92 | 80.10 | 51.21 | 79.03 | 85.42 | 82.46 |
| PGD-AT | 83.36 | 57.33 | 80.76 | 48.50 | 79.88 | 62.71 | 81.30 | 50.00 | 80.03 | 87.13 | 83.74 |
| RFGSM-AT | 85.84 | 45.94 | 81.85 | 37.49 | 81.00 | 60.25 | 83.28 | 37.91 | 81.04 | 89.42 | 86.20 |
| ROGET (PGD) | 87.40 | 49.91 | 83.65 | 42.14 | 82.88 | 62.45 | 84.91 | 41.53 | 82.82 | 91.12 | 87.78 |
| ROGET (AdvGAN) | 88.23 | 44.24 | 83.83 | 36.48 | 83.05 | 61.01 | 85.51 | 37.42 | 83.15 | 91.96 | 88.60 |

Table 10: Performance under *revealed hyperparameter selection*, where for the baselines, we select the hyperparameters which would maximize the overall accuracy using the adversary's true subset selection strategy (uncertainty based). Numbers in green (yellow) indicate the best (second best) performers.

### H.4 WORST CASE OVERALL ACCURACY

In this experiment choose $R = 10000$ subsets $\{S_j\}_{j=1}^{R}$ uniformly at random from $D_{\text{Test}}$ and report the minimum $\mathcal{A}$ along with the corresponding $\mathcal{A}_{\text{clean}}$ and $\mathcal{A}_{\text{robust}}$. We use default hyperparameter selection and report the results for CIFAR10 and FMNIST in Table 11. We make the following observations: (1) our method achieves the best min $\mathcal{A}$ for PGD, AA, and MIFGSM attacks on CIFAR10 and PGD, Square and MIFGSM attacks on FMNIST. (2) There is no clear winner among the baselines. RFGSM-AT has a good $\mathcal{A}$ on CIFAR10 but poor $\mathcal{A}_{\text{robust}}$. For FMNIST, GAT achieves the highest $\mathcal{A}$ for AA. However it is purely because of its $\mathcal{A}_{\text{clean}}$ as it has 0% $\mathcal{A}_{\text{robust}}$.

| | CIFAR10 | | | | | | | | | | | |
|---|---|---|---|---|---|---|---|---|---|---|---|---|
| | PGD | | | AA | | | Square | | | MIFGSM | | |
| | $\mathcal{A}_{\text{clean}}$ | $\mathcal{A}_{\text{robust}}$ | min $\mathcal{A}$ | $\mathcal{A}_{\text{clean}}$ | $\mathcal{A}_{\text{robust}}$ | min $\mathcal{A}$ | $\mathcal{A}_{\text{clean}}$ | $\mathcal{A}_{\text{robust}}$ | min $\mathcal{A}$ | $\mathcal{A}_{\text{clean}}$ | $\mathcal{A}_{\text{robust}}$ | min $\mathcal{A}$ |
| GAT | 78.96 | 50.60 | 76.12 | 78.96 | 38.70 | 74.93 | 78.96 | 51.70 | 76.23 | 78.98 | 42.90 | 75.37 |
| FBF | 75.02 | 65.80 | 74.10 | 75.18 | 25.20 | 70.18 | 75.18 | 25.70 | 70.23 | 75.09 | 30.70 | 70.65 |
| TRADES | 80.28 | 56.90 | 77.94 | 80.28 | 46.10 | 76.86 | 80.28 | 56.40 | 77.89 | 80.53 | 46.80 | 77.16 |
| Nu-AT | 83.30 | 49.30 | 79.90 | 83.49 | 37.20 | 78.86 | 83.36 | 53.30 | 80.35 | 83.40 | 42.60 | 79.32 |
| MART | 81.58 | 54.90 | 78.91 | 81.57 | 42.50 | 77.66 | 81.57 | 54.90 | 78.90 | 81.47 | 46.80 | 78.00 |
| PGD-AT | 83.56 | 47.70 | 79.97 | 83.64 | 38.80 | 79.16 | 83.66 | 52.90 | 80.58 | 83.60 | 42.20 | 79.46 |
| RFGSM-AT | 89.19 | 26.70 | 82.94 | 89.39 | 18.10 | 82.26 | 89.19 | 48.00 | 85.07 | 89.14 | 18.70 | 82.10 |
| ROGET ($a_\phi$ = PGD) | 85.46 | 47.30 | 81.64 | 85.58 | 37.60 | 80.78 | 85.63 | 52.60 | 82.33 | 85.40 | 41.00 | 80.96 |
| ROGET ($a_\phi$ = AdvGAN) | 88.12 | 39.40 | 83.25 | 88.09 | 31.40 | 82.42 | 88.37 | 49.20 | 84.45 | 87.98 | 33.30 | 82.51 |
| | FMNIST | | | | | | | | | | | |
| | PGD | | | AA | | | Square | | | MIFGSM | | |
| | $\mathcal{A}_{\text{clean}}$ | $\mathcal{A}_{\text{robust}}$ | min $\mathcal{A}$ | $\mathcal{A}_{\text{clean}}$ | $\mathcal{A}_{\text{robust}}$ | min $\mathcal{A}$ | $\mathcal{A}_{\text{clean}}$ | $\mathcal{A}_{\text{robust}}$ | min $\mathcal{A}$ | $\mathcal{A}_{\text{clean}}$ | $\mathcal{A}_{\text{robust}}$ | min $\mathcal{A}$ |
| GAT | 91.54 | 3.30 | 82.72 | 91.54 | 0.00 | 82.39 | 91.54 | 0.00 | 82.39 | 91.57 | 3.60 | 82.77 |
| FBF | 80.68 | 13.60 | 73.97 | 80.97 | 52.70 | 78.14 | 80.97 | 54.20 | 78.29 | 80.73 | 75.60 | 80.22 |
| TRADES | 86.23 | 54.40 | 83.05 | 86.64 | 37.90 | 81.77 | 86.37 | 44.60 | 82.19 | 86.53 | 69.60 | 84.84 |
| Nu-AT | 90.56 | 4.20 | 81.92 | 90.52 | 0.00 | 81.47 | 90.54 | 0.00 | 81.49 | 90.74 | 10.50 | 82.72 |
| MART | 79.63 | 65.70 | 78.24 | 79.48 | 47.30 | 76.26 | 79.48 | 48.80 | 76.41 | 79.71 | 68.00 | 78.54 |
| PGD-AT | 84.17 | 68.80 | 82.63 | 84.32 | 37.80 | 79.67 | 84.32 | 41.50 | 80.04 | 84.39 | 73.20 | 83.27 |
| RFGSM-AT | 89.33 | 9.80 | 81.38 | 89.24 | 0.00 | 80.32 | 89.24 | 0.00 | 80.32 | 89.34 | 20.70 | 82.48 |
| ROGET ($a_\phi$ = PGD) | 87.28 | 49.30 | 83.48 | 87.34 | 34.70 | 82.08 | 87.34 | 39.50 | 82.56 | 87.46 | 69.40 | 85.65 |
| ROGET ($a_\phi$ = AdvGAN) | 88.06 | 26.90 | 81.94 | 88.30 | 23.10 | 81.78 | 88.23 | 29.20 | 82.33 | 88.38 | 68.20 | 86.36 |

Table 11: Min $\mathcal{A}$ across $R = 10000$ random subsets along with the corresponding $\mathcal{A}_{\text{clean}}$ and $\mathcal{A}_{\text{robust}}$ for CIFAR10 and FMNIST. For all the methods, we use default hyperparameter selection.

## H.5 TRADE OFF BETWEEN $\mathcal{A}_{\text{clean}}$ AND $\mathcal{A}_{\text{robust}}$

In this experiment, we evaluate various hyperparameters of the baselines and plot their $\mathcal{A}_{\text{clean}}$ vs $\mathcal{A}_{\text{robust}}$ in Table 12 and 13. Each point on the plot represents a hyperparameter of the method. Here the adversary uses uncertainty based subset selection where the true subset chosen for attack $S^{\text{latent}}$ consists of top $10\%$ test instances in terms of the uncertainty of a classifier trained on all the clean examples. We observe that our method forms a pareto optimal front for all the attacks and hence achieves a better trade off between $\mathcal{A}_{\text{clean}}$ and $\mathcal{A}_{\text{robust}}$. We also notice that the baselines do not follow a regular trend and show high sensitivity with their tunable parameter. In that aspect, our method is relatively stable with respect to $\rho$.

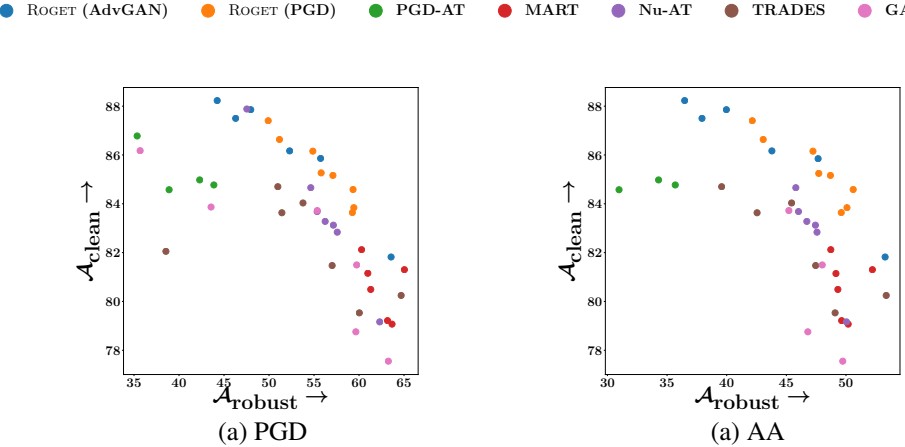

Figure 12: Trade off between $\mathcal{A}_{\text{clean}}$ vs $\mathcal{A}_{\text{robust}}$ for all methods for whitebox attacks (PGD, AA) on CIFAR10. Here, the adversary adopts uncertainty based subset selection to perform attack, where the true subset chosen for attack $S^{\text{latent}}$ consists of top $10\%$ test instances in terms of the uncertainty of a classifier trained on all the clean examples.

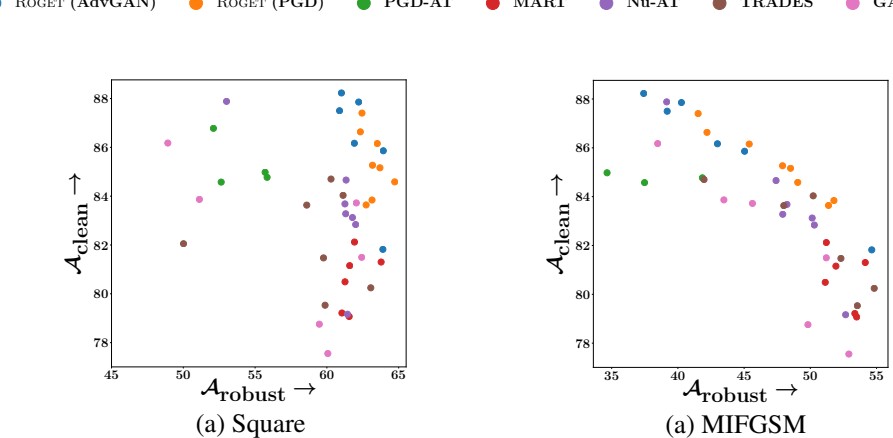

Figure 13: Trade off between $\mathcal{A}_{\text{clean}}$ vs $\mathcal{A}_{\text{robust}}$ for all methods for blackbox attacks (Square, MIFGSM) on CIFAR10. Here, the adversary adopts uncertainty based subset selection to perform attack, where the true subset chosen for attack $S^{\text{latent}}$ consists of top $10\%$ test instances in terms of the uncertainty of a classifier trained on all the clean examples.

## H.6 Variation of $\mathcal{A}_{\text{robust}}$, $\mathcal{A}_{\text{clean}}$ and $\mathcal{A}$ vs $\rho$

On CIFAR10 dataset, we try out different values of $\rho$ for two variants of our methods, *i.e.*, ROGET ($a_\phi = $ PGD) and ROGET ($a_\phi = $ AdvGAN). We probe the variation of $\mathcal{A}_{\text{clean}}$, $\mathcal{A}_{\text{robust}}$ and $\mathcal{A}$ vs $\rho$ for two attacks— AA (standard) and black-box MI-FGSM. The plot for ROGET ($a_\phi = $ PGD) is shown in Figure 14. For each value of $\rho$, we test using 5 random attack-clean (1:9) splits on the test set and report the mean accuracy with standard deviation. The plot for ROGET ($a_\phi = $ AdvGAN) is shown in Figure 15. We make the following observations: (1) For white box attack (AA), the standard deviation from the mean is $\pm 1.07\%$ across all values of $\rho$ and across all models. In case of black box MI-FGSM attack, the standard deviation rises to $\pm 3.34\%$. (2) For ROGET ($a_\phi = $ PGD), $\mathcal{A}_{\text{robust}}$ accuracy decreases as $\rho$ increases (except $\rho = 8$). (3) For ROGET ($a_\phi = $ AdvGAN), we observe that $\mathcal{A}_{\text{robust}}$ rises slightly before decreasing as $\rho$ increases.

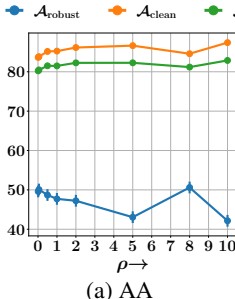
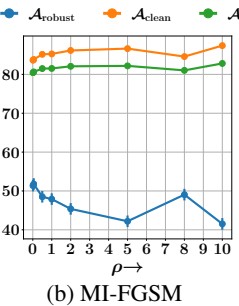

(a) AA                 (b) MI-FGSM

Figure 14: Variation of $\mathcal{A}_{\text{robust}}$, $\mathcal{A}_{\text{clean}}$ and $\mathcal{A}$ vs $\rho$ of ROGET ($a_\phi = $ PGD) on CIFAR10 dataset for two attacks *viz.*, Auto Attack and MI-FGSM. For each value of $\rho$, we test using uncertainty based subset selection and report the mean accuracy with standard deviation. We show the mean of $\mathcal{A}_{\text{robust}}$, $\mathcal{A}_{\text{clean}}$ and $\mathcal{A}$, along with error bars to show the standard deviation.

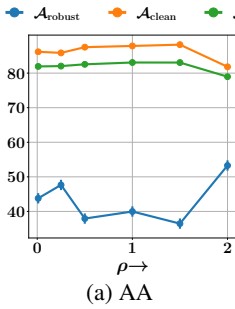
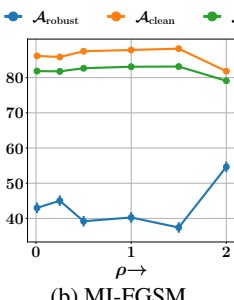

(a) AA                 (b) MI-FGSM

Figure 15: Variation of $\mathcal{A}_{\text{robust}}$, $\mathcal{A}_{\text{clean}}$ and $\mathcal{A}$ vs $\rho$ of ROGET ($a_\phi = $ AdvGAN) on CIFAR10 dataset for two attacks *viz.*, Auto Attack and MI-FGSM. For each value of $\rho$, we test using uncertainty based subset selection and report the mean accuracy with standard deviation. We show the mean of $\mathcal{A}_{\text{robust}}$, $\mathcal{A}_{\text{clean}}$ and $\mathcal{A}$, along with error bars to show the standard deviation

### H.7 RUN-TIME AND MEMORY ANALYSIS

In this section, we present details about the training time and the maximum GPU memory required for all methods while training on CIFAR10. Note that the time mentioned for our method includes both, the time taken for a gradient step and a stochastic greedy step. Here, the size of the attacked subset was $|S| = 0.1|D|$, where $D$ denotes the training set. The results are presented in Table 16.

| **CIFAR10** | Time(s)/epoch | Max GPU Memory (GB) |
|---|---|---|
| GAT | 86.80 | 2.0 |
| FBF | 87.36 | 2.7 |
| TRADES | 578.61 | 4.0 |
| Nu-AT | 74.29 | 3.2 |
| MART | 284.52 | 2.8 |
| PGD-AT | 290.18 | 2.7 |
| RFGSM-AT | 39.07 | 2.7 |
| ROGET (PGD) | 479.05 | 5.1 |
| ROGET (AdvGAN) | 68.89 | 7.7 |

Table 16: Run-time and memory analysis for all the methods on CIFAR10.

### H.8 ATTACK ON THE MOST VULNERABLE SET

In this experiment, we take the final set $\widehat{S}_T$ from our algorithm for CIFAR-10 and find $b = 0.1|D|$ number of samples with highest loss. Then, we compute the corresponding nearest samples from the test set to get the most vulnerable test set. We attack on this set for all the methods and show the results in Table 17. We observe that our method makes a good trade-off between accuracy and robustness. Although RFGSM-AT is best in terms of overall accuracy, its robustness is very poor.

| | $\mathcal{A}_{\text{clean}}$ | $\mathcal{A}_{\text{robust}}$ | $\mathcal{A}$ |
|---|---|---|---|
| TRADES | 80.1 | 24.4 | 74.5 |
| Nu-AT | 41.3 | 0.0 | 37.2 |
| MART | 78.1 | 24.1 | 72.7 |
| PGD-AT | 82.6 | 23.2 | 76.7 |
| RFGSM-AT | 91.0 | 6.6 | 82.5 |
| ROGET ($a_\phi = $ PGD) | 88.1 | 19.0 | 81.2 |

Table 17: Performance under AA (white-box) on the most vulnerable test set of CIFAR-10 under the worst case hyperparameter selection, when $S^{\text{latent}}$ was chosen using uncertainty based subset selection strategy.

### H.9 WORST-CASE HYPERPARAMETER SETTING FOR FMNIST

We present the results of worst-case hyperparameter selection for FMNIST in Table 18. Here, we achieve the best overall accuracy $\mathcal{A}$ for three attacks and second-best overall accuracy for the rest.

| **FMNIST** | | PGD | | AA | | Square | | MIFGSM | | AdvGAN | |
|---|---|---|---|---|---|---|---|---|---|---|---|
| | $\mathcal{A}_{\text{clean}}$ | $\mathcal{A}_{\text{robust}}$ | $\mathcal{A}$ | $\mathcal{A}_{\text{robust}}$ | $\mathcal{A}$ | $\mathcal{A}_{\text{robust}}$ | $\mathcal{A}$ | $\mathcal{A}_{\text{robust}}$ | $\mathcal{A}$ | $\mathcal{A}_{\text{robust}}$ | $\mathcal{A}$ |
| GAT | 91.47 | 3.18 | 82.64 | 0.00 | 82.32 | 0.01 | 82.32 | 8.32 | 83.15 | 29.09 | 85.24 |
| FBF | 80.47 | 14.29 | 73.85 | 66.74 | 79.10 | 68.43 | 79.27 | 84.81 | 80.90 | 85.94 | 81.02 |
| TRADES | 86.85 | 62.90 | 84.45 | 53.37 | 83.50 | 57.37 | 83.90 | 81.43 | 86.30 | 70.42 | 85.20 |
| Nu-AT | 90.37 | 5.92 | 81.92 | 0.00 | 81.33 | 0.07 | 81.34 | 20.26 | 83.36 | 14.01 | 82.73 |
| MART | 78.94 | 77.72 | 78.82 | 59.73 | 77.02 | 61.46 | 77.19 | 79.81 | 79.03 | 70.10 | 78.06 |
| PGD-AT | 83.77 | 79.62 | 83.36 | 53.62 | 80.76 | 57.05 | 81.10 | 83.73 | 83.77 | 72.88 | 82.69 |
| RFGSM-AT | 89.46 | 13.49 | 81.86 | 0.00 | 80.51 | 0.02 | 80.51 | 24.88 | 83.00 | 21.85 | 82.70 |
| ROGET (PGD) | 87.01 | 63.24 | 84.63 | 49.63 | 83.27 | 54.29 | 83.74 | 80.92 | 86.40 | 69.16 | 85.22 |
| ROGET (AdvGAN) | 87.08 | 49.62 | 83.33 | 47.32 | 83.10 | 51.99 | 83.57 | 82.49 | 86.62 | 73.44 | 85.71 |

Table 18: Performance comparison under *worst case hyperparameter setting* for two white box attacks (PGD and AA) and three black box attacks (Square, MIFGSM, and AdvGAN) on FMNIST. Numbers in green (yellow) indicate the best (second best) performers.

### H.10 EXPERIMENTS ON CIFAR100

Here, we present results on CIFAR-100 dataset. Here, the model is Resnet-9. In default hyperparameter setting, we use $\rho = 0.5$ for ROGET ($a_\phi = $ PGD) and $\rho = 2.0$ for ROGET ($a_\phi = $ AdvGAN) . For the baselines, we choose the same default hyperparameters as in CIFAR10. The results can be seen in Table 19. We also report the results of worst-case hyperparameter setting in Table 20. We observe that our method outperforms all the baselines in terms of clean, robust (except on PGD attack), and overall accuracies.

| CIFAR100 | | PGD | | AA | | Square | | MIFGSM | | AdvGAN | |
|---|---|---|---|---|---|---|---|---|---|---|---|
| | $\mathcal{A}_{\text{clean}}$ | $\mathcal{A}_{\text{robust}}$ | $\mathcal{A}$ | $\mathcal{A}_{\text{robust}}$ | $\mathcal{A}$ | $\mathcal{A}_{\text{robust}}$ | $\mathcal{A}$ | $\mathcal{A}_{\text{robust}}$ | $\mathcal{A}$ | $\mathcal{A}_{\text{robust}}$ | $\mathcal{A}$ |
| GAT | 45.05 | 43.37 | 44.88 | 21.50 | 42.69 | 24.03 | 42.95 | 20.75 | 42.62 | 31.48 | 43.69 |
| TRADES | 43.71 | 47.56 | 44.10 | 32.50 | 42.59 | 35.04 | 42.84 | 35.44 | 42.88 | 43.91 | 43.73 |
| Nu-AT | 32.64 | 33.15 | 32.70 | 8.03 | 30.37 | 24.84 | 31.86 | 8.24 | 30.39 | 15.72 | 31.14 |
| MART | 33.66 | 32.49 | 33.55 | 17.41 | 32.06 | 19.72 | 32.27 | 22.89 | 32.59 | 26.45 | 32.94 |
| RFGSM-AT | 44.55 | 41.66 | 44.26 | 9.20 | 41.02 | 10.34 | 41.13 | 8.10 | 40.91 | 14.71 | 41.57 |
| ROGET (PGD, $\rho = 0.5$) | 51.65 | 47.65 | 51.25 | 40.75 | 50.56 | 45.52 | 51.04 | 44.03 | 50.89 | 57.57 | 52.24 |
| ROGET (AdvGAN, $\rho = 2.0$) | 53.29 | 45.02 | 52.46 | 39.76 | 51.92 | 44.93 | 52.45 | 41.65 | 52.11 | 57.52 | 53.70 |

Table 19: Performance comparison under *default hyperparameter setting* on CIFAR100. We report (percentage) (i) accuracy on the clean examples $\mathcal{A}_{\text{clean}}$, (ii) robustness to the adversarial perturbations $\mathcal{A}_{\text{robust}}$ and (iii) overall accuracy $\mathcal{A}$. Here, the adversary adopts uncertainty-based subset selection to perform attack, where the true subset chosen for attack $S^{\text{latent}}$ consists of top $10\%$ test instances in terms of the uncertainty of a classifier trained on all the clean examples.

| CIFAR100 | | PGD | | AA | | Square | | MIFGSM | | AdvGAN | |
|---|---|---|---|---|---|---|---|---|---|---|---|
| | $\mathcal{A}_{\text{clean}}$ | $\mathcal{A}_{\text{robust}}$ | $\mathcal{A}$ | $\mathcal{A}_{\text{robust}}$ | $\mathcal{A}$ | $\mathcal{A}_{\text{robust}}$ | $\mathcal{A}$ | $\mathcal{A}_{\text{robust}}$ | $\mathcal{A}$ | $\mathcal{A}_{\text{robust}}$ | $\mathcal{A}$ |
| GAT | 47.23 | 44.11 | 46.92 | 18.54 | 44.36 | 21.56 | 44.66 | 16.63 | 44.17 | 28.50 | 45.36 |
| TRADES | 49.37 | 47.95 | 49.23 | 26.54 | 47.09 | 29.13 | 47.34 | 25.40 | 46.97 | 38.00 | 48.23 |
| Nu-AT | 32.64 | 33.15 | 32.70 | 8.03 | 30.37 | 24.84 | 31.86 | 8.24 | 30.39 | 15.72 | 31.14 |
| MART | 33.66 | 32.49 | 33.55 | 17.41 | 32.06 | 19.72 | 32.27 | 22.89 | 32.59 | 26.45 | 32.94 |
| RFGSM-AT | 44.55 | 41.66 | 44.26 | 9.20 | 41.02 | 10.34 | 41.13 | 8.10 | 40.91 | 14.71 | 41.57 |
| ROGET (PGD) | 52.71 | 46.89 | 52.13 | 40.77 | 51.52 | 45.29 | 51.97 | 43.60 | 51.80 | 57.20 | 53.16 |
| ROGET (AdvGAN) | 53.29 | 45.02 | 52.46 | 39.76 | 51.92 | 44.93 | 52.45 | 41.65 | 52.11 | 57.52 | 53.70 |

Table 20: Performance comparison under *worst-case hyperparameter setting* on CIFAR100. We report (percentage) (i) accuracy on the clean examples $\mathcal{A}_{\text{clean}}$, (ii) robustness to the adversarial perturbations $\mathcal{A}_{\text{robust}}$ and (iii) overall accuracy $\mathcal{A}$. Here, the adversary adopts uncertainty-based subset selection to perform attack, where the true subset chosen for attack $S^{\text{latent}}$ consists of top $10\%$ test instances in terms of the uncertainty of a classifier trained on all the clean examples.

### H.11 COMPARISON OF ROBUST ACCURACY SUBJECT TO A MINIMUM OVERALL ACCURACY

Here, we produce more results related to Table 21. Specifically, we first tune the hyperparameters of all the methods to ensure that the overall accuracy of all methods reaches a given threshold and then compare their robustness. If $P$ indicates the hyperparameters, then we find $\max_P \mathcal{A}_{\text{robust}}(P)$ such that $\mathcal{A}(P) \geq a$ for some given $a$. We present the results of CIFAR10 for $a = 0.81$ and FMNIST for $a = 0.83$ in Table 21 for different attacks. For CIFAR10, we find that ROGET ($a_\phi = $ PGD) is the best performer in terms of robust accuracy and ROGET ($a_\phi = $ AdvGAN) is the best performer in terms of overall accuracy (except for MIFGSM attack). Moreover, ROGET ($a_\phi = $ AdvGAN) is the second-best performer in terms of robust accuracy. For FMNIST, ROGET ($a_\phi = $ AdvGAN) achieves the highest robust accuracy $\mathcal{A}_{\text{robust}}$ (except for PGD attack).

| CIFAR10 | PGD | | AA | | Square | | MIFGSM | | AdvGAN | |
|---|---|---|---|---|---|---|---|---|---|---|
| | $\mathcal{A}_{\text{robust}}$ | $\mathcal{A}$ | $\mathcal{A}_{\text{robust}}$ | $\mathcal{A}$ | $\mathcal{A}_{\text{robust}}$ | $\mathcal{A}$ | $\mathcal{A}_{\text{robust}}$ | $\mathcal{A}$ | $\mathcal{A}_{\text{robust}}$ | $\mathcal{A}$ |
| GAT | 35.68 | 81.13 | – | – | 62.05 | 81.56 | 38.47 | 81.41 | 89.66 | 86.52 |
| FBF | – | – | – | – | – | – | – | – | – | – |
| TRADES | 53.78 | 81.01 | – | – | 61.13 | 81.74 | 0.04 | 81.90 | 88.27 | 85.06 |
| Nu-AT | 54.64 | 81.66 | – | – | 61.34 | 82.33 | 39.15 | 83.01 | 91.28 | 88.22 |
| MART | – | – | – | – | – | – | – | – | 85.42 | 82.46 |
| PGD-AT | 35.35 | 81.64 | – | – | 55.84 | 81.88 | – | – | 92.66 | 88.82 |
| RFGSM-AT | 45.94 | 81.85 | 37.49 | 81.00 | 60.25 | 83.28 | 37.91 | 81.04 | 89.42 | 86.20 |
| ROGET (PGD) | 59.42 | 81.40 | 50.60 | 81.19 | 64.73 | 82.60 | 49.05 | 81.03 | 91.12 | 87.78 |
| ROGET (AdvGAN) | 55.73 | 82.85 | 47.66 | 82.03 | 63.93 | 83.67 | 45.04 | 81.78 | 91.96 | 88.60 |

| FMNIST | PGD | | AA | | Square | | MIFGSM | | AdvGAN | |
|---|---|---|---|---|---|---|---|---|---|---|
| | $\mathcal{A}_{\text{robust}}$ | $\mathcal{A}$ | $\mathcal{A}_{\text{robust}}$ | $\mathcal{A}$ | $\mathcal{A}_{\text{robust}}$ | $\mathcal{A}$ | $\mathcal{A}_{\text{robust}}$ | $\mathcal{A}$ | $\mathcal{A}_{\text{robust}}$ | $\mathcal{A}$ |
| GAT | – | – | – | – | – | – | 9.43 | 83.07 | 29.09 | 85.24 |
| FBF | – | – | – | – | – | – | – | – | – | – |
| TRADES | 75.30 | 83.35 | 53.37 | 83.50 | 57.37 | 83.90 | 82.05 | 87.32 | 70.60 | 86.18 |
| Nu-AT | – | – | – | – | – | – | 20.26 | 83.36 | – | – |
| MART | – | – | – | – | – | – | – | – | – | – |
| PGD-AT | 70.50 | 89.66 | 4.42 | 83.05 | 37.74 | 86.38 | 80.55 | 90.66 | 54.32 | 88.04 |
| RFGSM-AT | – | – | – | – | – | – | – | – | – | – |
| ROGET (PGD) | 64.10 | 84.62 | 49.67 | 83.25 | 54.29 | 83.74 | 81.59 | 86.99 | 69.47 | 85.78 |
| ROGET (AdvGAN) | 72.61 | 84.87 | 54.86 | 83.10 | 58.19 | 83.43 | 83.97 | 86.01 | 75.00 | 85.28 |

Table 21: Performance comparison for two white-box attacks (PGD and AA) and three black-box attacks (Square, MIFGSM, and AdvGAN). We report (percentage) (i) accuracy on the clean examples $\mathcal{A}_{\text{clean}}$, (ii) robustness to the adversarial perturbations $\mathcal{A}_{\text{robust}}$ and (iii) overall accuracy $\mathcal{A}$. Here, we apply a threshold on $\mathcal{A}$ and then compare the best possible $\mathcal{A}_{\text{robust}}$ for all the methods. Also, the adversary adopts uncertainty-based subset selection to perform the attack, where the true subset chosen for attack $S^{\text{latent}}$ consists of top $10\%$ test instances in terms of the uncertainty of a classifier trained on all the clean examples. Numbers in green (yellow) indicate the best (second best) performers. "–" indicates that we could not find the hyperparameter that satisfies the condition applied on $\mathcal{A}$

## H.12   PERFORMANCE VARIATION WITH $|S^{\text{LATENT}}|$

Here, we train both the variants of ROGET using $b = 0.1|D_{\text{Tr}}|$ and evaluate using different number of instances $|S^{\text{latent}}|$ perturbed during test. We already reported the results for CIFAR10 in the main. In Figure 22, we report the results for FMNIST, which show that ROGET ($a_{\phi}$ = AdvGAN) and ROGET ($a_{\phi}$ = PGD) outperform the baselines at the smaller values of $|S^{\text{latent}}|$. For larger value of $|S^{\text{latent}}|$, our methods are only being outperformed by the PGD-AT.

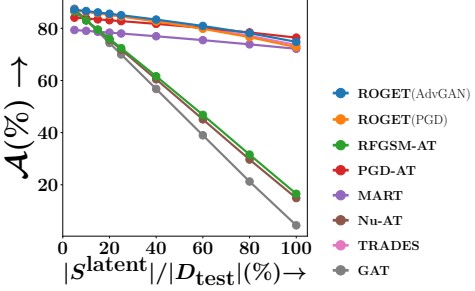

Figure 22: $\mathcal{A}$ vs. $|S^{\text{latent}}|$ for FMNIST

In the experiments, *i.e.*, in Figures 5 and 22, we kept the value of $\rho$ same across different values of $|S^{\text{latent}}|$, set by worst case hyperparameter selection with $b = 0.1|D_{\text{tr}}|$ We observed that our method performed better for smaller values of $|S^{\text{latent}}|$. Next, we adjust $\rho$ for different $|S^{\text{latent}}|$ with $b = \frac{|S^{\text{latent}}|}{|D_{\text{test}}|}|D_{\text{tr}}|$, *i.e.*, we provide a little information about the proportion of instances that are going to be attacked. Figure 23 summarizes the results which show that ROGET ($a_{\phi}$ = AdvGAN) outperforms the baselines for a wide range of the size of attacked test instances.

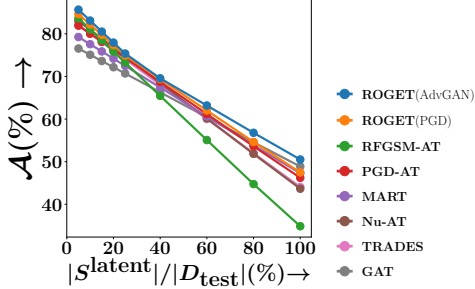

Figure 23: $\mathcal{A}$ vs. $|S^{\text{latent}}|$ for CIFAR10 after tuning $\rho$ based on the size of number of instances being attacked.

## H.13 PLUGGING OTHER MODELS INTO OUR FRAMEWORK

Here, we performed experiments where we plugged TRADES loss in the proposed algorithm. Results are as follows for CIFAR10 for PGD and Square attack.

|  | $\mathcal{A}_{\text{clean}}$ | $\mathcal{A}_{\text{robust}}$ | $\mathcal{A}$ | $\mathcal{A}_{\text{robust}}$ | $\mathcal{A}$ |
|---|---|---|---|---|---|
| TRADES | 80.25 | 64.68 | 78.69 | 63.06 | 78.53 |
| TRADES-Our | 84.50 | 56.08 | 81.62 | 63.06 | 82.32 |

Table 24: Effect of plugging Trades into our algorithm improves its performance.

We observe that plugging TRADES in our algorithm improves the performance of TRADES in terms of the overall accuracy.

## H.14 EXPERIMENTS WITH LOSS BASED ATTACK

Here, we performed the experiments where the adversary chooses instances based on the predicted accuracy. The following table summarizes the results for PGD and Square Attacks which show that our methods outperform the existing method in terms of the overall accuracy as well as robust accuracy.

| CIFAR10 |  | PGD | | Square | |
|---|---|---|---|---|---|
|  | $\mathcal{A}_{\text{clean}}$ | $\mathcal{A}_{\text{robust}}$ | $\mathcal{A}$ | $\mathcal{A}_{\text{robust}}$ | $\mathcal{A}$ |
| GAT | 78.85 | 46.33 | 75.60 | 37.32 | 74.70 |
| FBF | 76.49 | 54.62 | 74.30 | 21.65 | 71.01 |
| TRADES | 84.63 | 32.36 | 79.40 | 32.77 | 79.44 |
| Nu-AT | 85.62 | 34.00 | 80.46 | 36.35 | 80.70 |
| MART | 81.86 | 42.29 | 77.90 | 36.41 | 77.32 |
| PGD-AT | 84.92 | 37.07 | 80.14 | 34.54 | 79.88 |
| RFGSM-AT | 86.76 | 27.06 | 80.79 | 34.13 | 81.50 |
| ROGET(PGD) | 86.23 | 37.64 | 81.38 | 38.18 | 81.43 |
| ROGET(AdvGAN) | 86.75 | 33.98 | 81.47 | 38.24 | 81.90 |

Table 25: Performance on loss-based attack.

