# OpenReview forum: "Robust Training through Adversarially Selected Data Subsets"
_ICLR.cc/2023/Conference — Submitted to ICLR 2023_

### Official Review · Reviewer_iAPp · 2022-10-18

**Confidence:** 2
**Correctness:** 2
**Technical Novelty And Significance:** 3
**Empirical Novelty And Significance:** 3
**Recommendation:** 6

**Clarity, Quality, Novelty And Reproducibility:**

Clarity: The main concept of this paper is clearly written. The details of the theoretical derivations contain several mistakes as mentioned above.
Quality: The presentation quality is very good. The experimental results are clearly illustrated.
Novelty: The problem setting in which the adversary perturb a subset is novel and practically useful.
Reproducibility: The authors provide the full proofs and the codes for reproducing the experiments.

**Strength And Weaknesses:**

Strengths:
Since deep neural networks are practically used in our society, everyone acknowledges the importance of robust training of neural networks. However, as the authors say, the existing frameworks for robust training are too pessimistic, and often sacrifice its average performance for non-perturbed data. To overcome this issue, this paper considers a setting in which the adversary can perturb a small subset of data points. The proposed algorithm is an alternating method with greedy subset selection and loss minimization, which is not so surprising, but its performance is supported by theory.

Weaknesses:
Although the problem setting is novel and the algorithmic idea is nice, the theoretical and empirical results are not so outstanding.
On the theory side, the main theorem (Theorem 4) is not always meaningful. Theorem 2 states that $\lambda R(\theta)$ must be large to make the submodularity ratio $\gamma^*$ sufficiently large. On the other hand, if $\lambda R(\theta)$ is large, then $\kappa$ becomes small, and then $[1 - (e^{-\gamma^*}+\delta) / \kappa]^{-1}$ might be negative, which makes Theorem 4 meaningless. However, the problem is motivated by practice and I do not think the theoretical weakness is so problematic.
As the experimental results show, the proposed algorithm does not always outperform the existing algorithms. However, I think the authors sufficiently discuss the advantages of the proposed algorithm.
It would be better to revise the proofs since they contain small mistakes such as:
- $q \mu \beta \frac{|S \cup k|}{|S|}$ in the inequality (32) should be multiplied by $L_\phi$.
- $\sum_{i \in S \cup k}$ should be added before the last $\mu$ in (35).
- $q \mu \beta$ in the inequality (36) should be multiplied by $L_\phi$.
- $-\delta$ in the numerator should be replaced by $+\delta$ and $+\delta$ in the denominator should be replaced by $-\delta$ in the right-hand side of (63).

**Summary Of The Paper:**

This paper tackles the robust training problem of neural networks. In usual scenarios, the adversary chooses a single data point and perturbs it to be misclassified. On the other hand, the adversary in this study chooses a subset of data points and perturbs them to be misclassified. The goal is to minimize the average loss for all the perturbed and non-perturbed data points. The authors present an algorithm that alternately finds a subset to be perturbed via the greedy algorithm and minimizes the average loss via SGD. This algorithm is validated to be perform well both theoretically and empirically.

**Summary Of The Review:**

This paper proposes a new robust training framework and an algorithm with a theoretical performance guarantee. The concept is novel and practically useful, but the theoretical results are not so strong and the proposed algorithms do not always outperform the existing algorithms.

---

> ### Author Response · Authors · 2022-11-14
> **Response to Reviewer iAPp**
>
> We would like to thank the reviewer for the comments/suggestions, which would indeed improve our paper.
>
>
> > *Although the problem setting is novel and the algorithmic idea is nice, the theoretical and empirical results are not so outstanding. On the theory side, the main theorem (Theorem 4) is not always meaningful. Theorem 2 states that λR(θ) must be large to make the submodularity ratio γ  sufficiently large. On the other hand, if λR(θ)  is large, then κ becomes small, and then  [1−(exp(−γ) +δ)/κ]−1 might be negative, which makes Theorem 4 meaningless. However, the problem is motivated by practice and I do not think the theoretical weakness is so problematic.*
>
> We would like to emphasize that, we clearly made two assumptions under which the above approximation guarantee will never become negative.
>
> + $C_{min} > \lambda \theta_{max} ^2 (e^{-\gamma^*} +\delta) / (1-e^{-\gamma ^*} -\delta) $ (Assumption 2 (5), which ensures that adversarial network significantly perturbs a feature)
> + $\rho < [C_{min} ((e^{-\gamma^*} +\delta)^{-1}) -1) -\lambda \theta^2  _{max}] / \ell _{max}$. We stated this assumption in Theorem 4 ($\rho$ is non-trivial (i.e. >0) due to the first assumption).
>
> Under these stated conditions (At a high level, $C _{min}$ should be large and $\rho$ remains lower bounded by the above quantity), Theorem 4 will never become meaningless. Specifically, the second assumption directly gives us that $1- (e^{-\gamma^*} +\delta)/ \kappa >0$. The first assumption ensures that the upper bound of  $\rho$ never becomes negative.
>
> Of course, in other conditions, the result will not hold good. However,  here we focus on a non-convex function— in such cases, even in absence of adversarial perturbation, deriving convergence or generalization guarantee is already quite challenging and sometimes not plausible.  Thus, such guarantees become even more challenging in the presence of adversarial perturbation. As a result, our results require these necessary assumptions.
>
> We would also like to highlight that, as the reviewer correctly pointed out, $\lambda$ plays a crucial role in the approximation guarantee. This is because— the objective in Eq (1) is not monotone and we resorted to expressing it as a difference between  Eq 4 and Eq 5. This requires us to introduce $\lambda R(\theta)$ . However,  a very recent work (https://arxiv.org/abs/2202.03051) has been published in Neurips 2022, which shows that greedy like algorithms admit some approximation guarantee for submodular functions which are not always monotone. Using this method, we can compute a partial monotonicity ratio of the objective in Eq (1) and then use it to derive an approximation guarantee of our method, without introducing $\lambda R(\theta)$. Consequently, it would be independent of $\lambda$.
>
>
>
> > *As the experimental results show, the proposed algorithm does not always outperform the existing algorithms.*
>
> We elaborated it in detail in response to reviewer H4So *Table 1,2: the proposed method is not the best performer in terms of perturbed robustness.*
>
> > *Small mistakes*
>
> We are really thankful to the reviewer for catching the mistakes. We rectified them in the revised version.

---

### Official Review · Reviewer_H4So · 2022-10-24

**Confidence:** 4
**Correctness:** 3
**Technical Novelty And Significance:** 3
**Empirical Novelty And Significance:** 2
**Recommendation:** 5

**Clarity, Quality, Novelty And Reproducibility:**

The selected-subset robustness problem is important to the community and the proposed method looks new to me.

**Strength And Weaknesses:**

Strength:
- The paper is clearly motivated.
- The paper is well written and easy to follow.

Comments:
- I am a bit worried about the empirical significance.
   1) As in both Table 1 and Table 2, the proposed method is not the best performer in terms of perturbed robustness.
   2) As in Table 18-20, on CIFAR-100 the proposed method, although achieved second best in most cases, the gap to the best performer is pretty large around 5% in terms of overall accuracy. I have concerns about the scalability issue, that is the proposed method only perform well on small datasets? Does it have the ability of scaling to harder dataset, e.g., Tiny-ImageNet?
- I would suggest the authors to provide more related work. I checked the appendix, however, I didn't see the related work of worst-class problem and its application in adversarial robustness. The authors are also suggested to compare with those in Table 3.
- Fig 5 shows the percentage of instances chosen for perturbation, from 5% to 25%, I would suggest to extend axis all the way to 100%, to clearly show the transition power point of the proposed method. If possible, comparing more figures over different datasets would be even more interesting.

**Summary Of The Paper:**

Defending against all perturbed instances is the target of the most current defense mechanisms, and yet in practice only a subset of instances might be selected by the attacker. This paper aims at a new defense mechanism to minimize the worst case loss across all subsets. To solve this optimization problem, the authors design a group of algorithms that admits approximation guarantees for both convex and non-convex (under Polyak-Lojasiewicz condition) objectives. The paper also gives some theoretical analysis of the proposed method. In experiments, the paper shows that the proposed method outperforms SOTA methods on both FMNIST and CIFAR-10.

**Summary Of The Review:**

The paper is well-motivated. However, I have concerns on the empirical significance, which makes the paper marginally below the threshold.

---

> ### Author Response · Authors · 2022-11-14
> **Response to Reviewer H4So (Part-1/2)**
>
> We would thank the reviewer for the suggestions, which would improve the paper.
>
> > *Table 1,2: the proposed method is not the best performer in terms of perturbed robustness.*
>
> Apparently, yes, the proposed method is not the best performer in terms of the perturbed robustness as per the results presented in the paper. This is because of two reasons:
>
> (1) The baselines were aggressively optimized for robust accuracy (they incorporate the attack on each and every instance) and often not the overall accuracy when the number of instances attacked is low.  In contrast, our goal is to maximize overall accuracy.  Note that robust accuracy can always be increased by sacrificing overall accuracy. Existing baselines do exactly this in the context of our problem when a small number of instances are attacked. Thus, the robust accuracy of our method looks suboptimal because the baselines provide an excessively optimized robust accuracy at the cost of a drop in overall accuracy.
>
> (2) In Table-1, we *did NOT fine-tune* the hyperparameter $\rho$ for our method and set $\rho=1$ as the default value. This is because, to keep a realistic and technically sound setup, we make no assumption about the adversary’s subset selection strategy as well as the attack method. Such a setup renders cross-validation of $\rho$ very tricky. Hence, we set $\rho=1$ which we believe is the most natural option for a default specification to a potential end user, without any information about the adversary. In contrast, the default values of the hyperparameters prescribed by the baselines are set by fine-tuning. Consequently, they gave our method a stern (and perhaps significantly unfair) competition.
>
> We pondered about selecting $\rho$ similar to the baselines, to provide our method a fair platform to compare with the default setup of the baselines. However, we found that the baselines *only prescribed the values of the default hyperparameters* in their codes/papers; but, we could not find out a clear methodology behind setting up these default values in their codes/papers. As a result, we were unable to tune our hyperparameter $\rho$ in a way similar to the baselines. Therefore, we set $\rho=1$, which we felt was a natural option for default specification from an end-user perspective.
>
> Due to these two reasons, Tables 1 and 2 report better robust accuracy for the baselines. However, note that even after the above-mentioned relaxed hyperparameter choice, our method outperforms the baselines in terms of the overall accuracy in Table 1.
>
>
> Now, note that our method does provide better robust accuracy as well as better overall accuracy if we tune  $\rho$. In fact, Figs 12 and 13 (App H.5) show that our method provides a better trade-off between clean accuracy and robustness.  Thus we were aware that there are values of the $\rho$ which would better showcase our method in terms of robust accuracy. However, we preferred to show results with $\rho=1$ rather than just highlighting the best possible result.  In a scenario with no knowledge about the adversary and no clear consensus/ recipe for setting hyperparameter values by the baselines, we believed it is the natural choice of $\rho$ for a realistic and sound setup.
>
> But we acknowledge the reviewer’s concerns and clarify them through existing as well as more experiments as follows:
>
> In the following (1–4), we illustrate results that show we also perform well in terms of robust accuracy.
>
> (1)    Figs 12 and 13 (App H.5) contain a tradeoff plot between clean and robust accuracy. It shows that we provide a better trade-off for a wide range of robust accuracy.
>
> (2)  Most baselines allow some hyperparameters that tradeoff between clean and robust accuracy. Here we aim to compare the robust accuracy, subject to the condition that the overall accuracy for all methods crosses some threshold.  So, we first tune the hyperparameters of all the methods so that the overall accuracy of all methods reaches a given threshold and then compare their robustness.  If $H$ indicates the hyperparams, then we find $\max_{H} A _{robust}(H)$ such that $A(H) > a$ for some given $a$. Results are as follows (CIFAR10) for $a = 0.81$ (**bold** shows best, *italics* shows second best). Roget (PGD) is the best performer in terms of robust accuracy and Roget (AdvGAN) is the best performer in terms of overall accuracy.  Roget (AdvGAN) is the second-best performer in terms of robust accuracy.
>
> ||robust  (PGD)|total  (PGD)| \||robust (Square)|total (Square)
> |-|-|-|-|-|-
> GAT|35.68|81.13| \||62.05|81.56
> TRADES|53.78|81.01|\||61.13|81.74
> NUAT|54.64|81.66|\||61.34|82.33
> PGDAT|35.35|81.64|\||55.84|81.88
> RFGSMAT|45.94|*81.85*|\||60.25|*83.28*
> Our (PGD)|**59.42**|81.40|\||**64.73**|82.60
> Our (AdvGAN)|*55.73*|**82.85**|\||*63.93*|**83.67**
>
> FBF and MART are not in the table because we could not find hyperparams that satisfy $A(H)>a=0.81$. We put this result in the main (Tab 6). More results are in appendix (App. H.11)

---

> > ### Author Response · Authors · 2022-11-14
> > **Response to Reviewer H4So (Part-2/2)**
> >
> > [Contd from part-1]
> >
> > (3) in Appendix H.9, Table 17 in the *initial submission*, we showed results for AdvGAN attack. There, our method outperforms baselines in terms of the robust accuracy for CIFAR10 under worst-case hyperparameter selection. We quote the results here.
> >
> > ||Clear|Robust|Total
> > | -| -| -| -
> > GAT|77.55|81.08|77.91
> > FBF|74.92|37.22|71.15
> > TRADES|83.63|87.37|84.02
> > NuAT|84.66|88.18|85.01
> > MART|80.49|84.25|80.87
> > PGDAT|83.36|87.12|83.74
> > RFGSMAT|85.84|89.42|86.20
> > Our (PGD)|*86.63*|*90.15*|*86.99*
> > Our (AdvGAN)|**87.86**|**91.53**|**88.22**
> >
> > In the initial submission, we deferred them in the Appendix, as we preferred to show the results on popular attacks. Now, we included the results in Tab. 2.
> >
> > (4) During rebuttal, we performed more experiments where we tuned the default hyperparameter  $\rho$ : there, we observed significant improvement. Results are as follows for AdvGAN and Square attacks on CIFAR10.
> >
> > ||Robust (Square)|Total (Square)|Robust (AdvGAN)|Total (AdvGAN)
> > |-|-|-|-|-
> > GAT|59.47|76.83|82.04|79.08
> > FBF|31.59|70.59|37.22|71.15
> > TRADES|63.06|78.53|83.82|80.60
> > NuAT|61.78|80.99|86.82|83.49
> > MART|**63.78**|79.55|84.75|81.65
> > PGDAT|62.71|81.30|87.13|83.74
> > RFGSMAT|57.68|**85.96**|89.42|86.20
> > Our (PGD, $\rho$=1.0)|*63.18*|83.06|88.96|85.64
> > Our (AdvGAN, $\rho$=1.0)|62.21|85.29|*91.53*|*88.22*
> > Our (AdvGAN, $\rho$=0.01)|61.01|*85.51*|**91.96**|**88.60**
> >
> > For AdvGAN (Square), our method is best (second best) in terms of both robust and overall accuracy. In the previous draft, we had already put the results for the AdvGAN attack in the Appendix. We brought them in Tab-1.
> >
> > > *I didn't see the related work of the worst-class problem and its application in adversarial robustness. The authors are also suggested to compare with those in Table 3.*
> >
> > During rebuttal, we compared class-focused online learning (CFOL) proposed in “Revisiting adversarial training for the worst-performing class (https://openreview.net/forum?id=wkecshlYxI&referrer=%5BTMLR%5D(%2Fgroup%3Fid%3DTMLR) ).” This provides guarantees on the worst class loss.  We compared them in Table 3, which shows that our method outperforms this method.
> >
> > ||Airplane| Dog| Truck
> >  -| -| -| -|
> > GAT|75.19|75.06|75.28
> > FBF|72.39|71.77|67.11
> > TRADES|80.17|79.52|80.44
> > NuAT|81.32|80.54|81.68
> > MART|77.69|77.34|78.02
> > PGDAT|80.38|79.60|80.45
> > RFGSM|81.80|80.84|81.46
> > CFOL|70.10|69.61|70.63
> >  Roget, (PGD)|82.97|84.19|83.68
> >  Roget (AdvGAN)|84.24|85.63|84.03
> >
> > We collected more related works and discussed them in our related work section.
> >
> > > *As in Table 18-20, on CIFAR-100 the proposed method, .. the gap to the best performer is pretty large around 5% in terms of overall accuracy. [...] is the proposed method only perform well on small datasets? Does it have the ability of scaling to harder dataset, e.g., Tiny-ImageNet?*
> >
> > To the best of our knowledge, the baselines neither performed experiments on CIFAR100 nor prescribed any possible direction for choosing hyperparameters in general.  Thus, we struggled quite a bit to set up the model and the hyperparameters for all the methods including ours. During the rebuttal period, we found a better setting for both the model (Resnet-9)  and hyperparameters which improved the results of all the methods including the baselines.  Results are as follows for PGD and Square attacks (default hyperparameter tuning). We put it in the Appendix since we are still working on FBF and PGDAT (currently, we could not get the loss to converge). Once they're done, we can bring in the main. Since, no baseline reported results on CIFAR100, we had to work on their code to setup a new model on a new dataset.
> >
> > ||Clean| Robust  (PGD)| Total  (PGD)| Robust (Square)| Total (Square)
> > |-|-|-|-|-|-
> > GAT| 45.05| 43.37| 44.88|24.03|42.95
> > TRADES|43.71| *47.56*| 44.10| 35.04| 42.84
> > NUAT|32.64| 33.15| 32.70| 24.84| 31.86
> > MART|33.66| 32.49| 33.55| 19.72| 32.27
> > RFGSMAT|44.55| 41.66| 44.26| 10.34| 41.13
> > Our (PGD)|*51.65*|**47.65**| *51.25*| **45.52**| *51.04*
> > Our (AdvGAN)|**53.29**|45.02| **52.46**| *44.93*| **52.45**
> >
> > Our method outperforms all the baselines in terms of overall accuracy.  Appendix H.10 contains additional results. We have started experimenting on tiny imagenet. However, none of the baselines considered this hard dataset. Extending the model or hyperparameters from other datasets to this  dataset is naturally not working for any baselines. Thus, we  still need more time to find right model and hyperparameters for all the methods.
> >
> > > *Fig 5 [...] I would suggest to extend axis all the way to 100%, to clearly show the transition power point of the proposed method.*
> >
> > We extended the X axis to 100% in our figure, which shows that as long as the size of the attacked subset is less than ~30%, our method performs better. This happens when we keep the value of $\rho$ the same across different values of the attacked test size. Tuning $\rho$ using attacked set size leads to better results (Fig 23, App H.12).

---

### Official Review · Reviewer_gdCM · 2022-10-24

**Confidence:** 3
**Correctness:** 3
**Technical Novelty And Significance:** 4
**Empirical Novelty And Significance:** 4
**Recommendation:** 6

**Clarity, Quality, Novelty And Reproducibility:**

This paper is well-written and organized. The proposed method is theoretically motivated and novel. The authors provide the training details and the algorithm, which can help people to reproduce.

**Strength And Weaknesses:**

Strength
+ The new attack setting that adversarially selects a subset as an adversarial set is interesting and novel. The experimental results show that previous adversarial training methods are vulnerable to this adversarial subset attack. The proposed method can achieve the best performance even under the adaptive attack (subset selection with worst-case hyperparameter setup).
+ The proposed iterative greedy algorithm is well motivated by the theorem. The proposed method is compatible with variants of adversarial training methods.

Weaknesses (Questions)
- It could be better to show the results of the proposed method with variants of adversarial loss such as ROGET with TRADES loss to validate the compatibility of the proposed method.
- I have a concern regarding the adversarial subset attack. The adversary can choose to adversarially perturb those incorrectly-predicted data. This seems to be a realistic attack and could be viewed as the lower bound of the robust accuracy under the adversarial subset attack. It would be better to present the results of this type of attack.


**Summary Of The Paper:**

This paper considers a new type of adversary that chooses to only perturb a subset of data. The authors propose a defensive strategy that is formulated as a minimax problem that involves a worst-case subset selection procedure. This paper uses a greedy algorithm to realize the worst-case subset selection. The empirical results demonstrate the proposed method can gain the best overall accuracy on the selected adversarial data and unselected benign data in most scenarios.

**Summary Of The Review:**

This paper investigates a novel attack where the adversary only chooses to perturb a subset. The authors propose a defensive method that makes the model robust against adversarial subset attacks.  This paper is well-written and novel. Thus, I would like to accept this paper though I have some minor questions.

---

> ### Author Response · Authors · 2022-11-14
> **Response to reviewer gdCM**
>
> We would thank the reviewer for the suggestions, which would improve the paper.
>
> > *It could be better to show the results of the proposed method with variants of adversarial loss such as ROGET with TRADES loss to validate the compatibility of the proposed method.*
>
> During the rebuttal time, we performed experiments where we plugged TRADES loss in the proposed algorithm. Results are as follows for CIFAR10 for PGD and Square attack.
>
> || Clean| Robust  (PGD)| Total  (PGD)| Robust (Square)| Total (Square)
> |-|-|-|-|-|-
> TRADES|80.25| 64.68| 78.69| 63.06| 78.53
> TRADES-Our| 84.50| 56.08| **81.62**| 63.06| **82.32**
>
>
> We observe that plugging TRADES in our algorithm improves the performance of TRADES in terms of overall accuracy. We added this result in  Appendix H.13.
>
> > *I have a concern regarding the adversarial subset attack. The adversary can choose to adversarially perturb those incorrectly-predicted data. This seems to be a realistic attack and could be viewed as the lower bound of the robust accuracy under the adversarial subset attack. It would be better to present the results of this type of attack.*
>
>
> As the reviewer has suggested, we performed the experiments under the suggested subset selection strategy. The following table summarizes the results for PGD and Square Attacks which show that our methods outperform the existing method in terms of overall accuracy as well as robust accuracy.
>
> ||Clean|\||robust  (PGD)|total  (PGD)| \||robust (Square)|total (Square)
> |-|-|-|-|-|-|-|-
> GAT| 78.85| \|| 46.33| 75.60| \||  37.32| 74.70
> FBF| 76.49| \|| 54.62| 74.30| \||  21.65| 71.01
> TRADES| 84.63| \|| 32.36| 79.40| \||  32.77| 79.44
> NUAT| 85.62| \|| 34.00| 80.46| \||  36.35| 80.70
> MART| 81.86| \|| 42.29| 77.90| \||  36.41| 77.32
> PGDAT| 84.92| \|| *37.07*| 80.14| \||  34.54| 79.88
> RFGSMAT| 86.76| \|| 27.06| 80.79| \||  34.13| 81.50
> Our (PGD)| 86.23| \|| **37.64**| *81.38*| \||  *38.18*| *81.43*
> Our (AdvGAN)| 86.75| \|| 33.98| **81.47**| \||  **38.24**| **81.90**
>
> We added this result in  Appendix H.14.

---

### Official Review · Reviewer_SByk · 2022-10-28

**Confidence:** 3
**Correctness:** 2
**Technical Novelty And Significance:** 3
**Empirical Novelty And Significance:** 2
**Recommendation:** 5

**Clarity, Quality, Novelty And Reproducibility:**

The paper addresses an interesting setting and seems reproducible. The theoretical development of the work is nice, but empirical results are weak.

**Strength And Weaknesses:**

### Things I liked

- The paper addresses an interesting problem, clearly explains the difficult of the problem and proposes a solution with guarantees when certain assumptions hold.

- The authors conduct a reasonable set of experiments and provide logical choices of the baselines, attack methods, etc.

- The mix max over the hyper-parameter choice is nice little trick.

### Things that need clarification / improvement

- The empirical results are weak in several aspects.
  - The proposed method, ROGET, mostly outperforms other defenses (which are designed for the entire test-set) on clean test-data accuracy. It is at times the worst performing method of perturbed test data. There are also sentences like "Tuning ρ would easily improve the robustness, as shown in additional experiments in Appendix H" which surprises me on why would the authors not show their best results as part of the main paper?
  - If there is some knowledge about the true subset selection strategy at validation time, other methods (designed for all test attacks) outperforms ROGET.
  - As the amount of test data poisoned grows, the accuracy gains become minimal.

- Given the inner optimization is itself a difficult problem that admits an approximation, the authors should highlight the time/resource costs incurred by their defense vs. existing defenses. Beyond accuracy, this is an important metric to consider.

- There is no clear association between the assumptions made in proofs and the function used in practice. For example, is categorical-cross entropy a Polyak-Lojasiewicz Loss function? If not, the approximation guarantees do not hold here. This disrupts the flow of the paper somewhat.

**Summary Of The Paper:**

The paper considers a threat model where an attacker attacks a subset of the test data of a classification system. To develop a defense, the formulate the learning problem for the classier as a min-max optimization where the loss is minimized over the most adversarial subset of a particular cardinality. The authors show that the inner optimization that requires a subset selection (for a given $\theta$) us NP-hard. To solve this efficiently, they leverage a stochastic distorted greedy algorithm by (Harshaw et. al. 2019) and is able to give an approximation guarantee for the overall adversarial learning function for a particular class of loss functions. The authors then conduct a suite of experiments to showcase the effectiveness of their approach against existing defenses in the threat model they consider.

**Summary Of The Review:**

See above.

---

> ### Author Response · Authors · 2022-11-14
> **Response to Reviewer SByk (Part-1/2)**
>
> We thank the reviewer for the comments/suggestions, which would indeed improve our paper.
>
> > *ROGET at times the worst-performing method of perturbed test data. why would authors not show  best results as part of the main paper?*
>
> Roget is designed to maximize the *overall* accuracy (not clean accuracy) where a small subset of instances are attacked, rather than maximizing the robust accuracy alone. Tables 1 and 2 show that Roget is able to achieve its goal well.  But yes, as the reviewer correctly pointed out, we did not report the best result of Roget in the main, which is why the robust accuracy may look suboptimal in those tables.
> However, the robust accuracy of ours was modest, but not worst.
>
> Here, we explain why we did not put the best result and why in the reported cases, the robust accuracy is not good.
>
> (1) In Table-1, our goal was to provide a default hyperparameter setup for our method and compare it against the baselines at their default settings.  However, we found that the baselines *only prescribed the values of the default hyperparameters* in their codes/papers; but, we could not find out a clear methodology behind setting up these default values in their codes/papers. As a result, we could not tune our hyperparameter $\rho$ in a way similar to the baselines. We could have tuned $\rho$ in some other manner. However, we strictly ensured that the adversary remains unknown throughout training and validation. Such a condition has made any other protocol for hyperparameter tuning tricky.
>
> Due to these reasons, we preferred not to tune $\rho$ to set up its default value. Instead, we set $\rho=1$ which we believe is the most natural option as a default specification of $\rho$ to a potential end user, in absence of any knowledge about the adversary.  Note that, unlike our method, the hyperparameters prescribed by the baselines were set by them after fine-tuning. Thus, these baselines gave our method a stern (and perhaps unfair) competition and outperformed ours in terms of robust accuracy. On the other hand, despite such a relaxed hyperparameter choice, our method still outperforms the baselines in terms of overall accuracy, which is the primary metric our method is optimized for.
>
> We were aware that our relaxed hyperparameter choice in Table-1 might be unfair to our method itself in the presence of the tuned hyperparameters of the baselines.  We could have also used some other tuning strategy that is tailored to our setup (like min-max hyperparameter trick in the worst-case hyperparameter tuning), as our default strategy. But that might have been unfair to the baselines.
> Instead,  we decided to go with $\rho=1$, although we know that it may not showcase the best of our results.
>
> We would like to highlight that Figs 12 and 13 (Appendix H.5) contain the trade-off curve between clean accuracy and robust accuracy, which show that Roget has a better trade-off than the baselines. Thus we were aware that there are values of the $\rho$ which would better showcase our method in terms of robust accuracy. Still, we showed results with $\rho=1$ rather than highlighting the best result. In a scenario with no knowledge about the adversary and no clear consensus/recipe for setting hyperparameters by the baselines, we believe it is the natural choice of $\rho$ for a realistic and sound setup.
>
> (2) The baselines are aggressively optimized for robust accuracy (they incorporate the attack on each and every instance) and often not the overall accuracy when the number of instances attacked is low. In contrast, our goal is to maximize overall accuracy. Note that the robust accuracy can be increased by sacrificing overall accuracy. Existing baselines do exactly this in the context of our problem when a small number of instances are attacked. So, the robust accuracy of our method looks suboptimal because the baselines provide an excessively optimized robust accuracy at the cost of a drop in overall accuracy.
>
> The following (A-D) items show that we perform well even in terms of robustness.
>
> (A) Figs 12 and 13 (App H.5) already contain tradeoff curves between clean and robust accuracy. It shows that we provide a better tradeoff for a wide range of robust accuracies.
>
> (B) In CIFAR100 (App H.10), we consistently outperform the baselines in terms of both robust and overall accuracy. Results are as follows for PGD and Square attack for default hyperparameter setup.
>
> ||Robust (PGD)|Total (PGD)|Robust (Square)|Total (Square)
> |-|-|-|-|-
> GAT|43.37|44.88|24.03|42.95
> TRADES|*47.56*|44.10|35.04|42.84
> NUAT|33.15|32.70|24.84|31.86
> MART|32.49|33.55|19.72|32.27
> RFGSMAT|41.66|44.26|10.34|41.13
> Our(PGD)|**47.65**|*51.25*|**45.52**|*51.04*
> Our(AdvGAN)|45.02|**52.46**|*44.93*|**52.45**
>
> No baseline reported results on CIFAR100. We had to work on their code on a new dataset. As of now, we could not get the loss for FBF and PGDAT to converge. This is why, we put it in the Appendix. If the reviewer suggests, then we can bring it in the main.

---

> > ### Author Response · Authors · 2022-11-14
> > **Response to Reviewer SByk (Part-2/2)**
> >
> > [Contd. from Part -1]
> >
> > (C)  Most baselines allow some hyperparameters that trade-off between clean and robust accuracy. Thus, robustness can be improved by sacrificing overall accuracy. Hence, here we aim to compare the robust accuracy, subject to the condition that the overall accuracy for all methods crosses some threshold.  Specifically, we first tune the hyperparameters of all the methods to ensure that the overall accuracy of all methods reaches a given threshold and then compare their robustness.  If $H$ indicates the hyperparams, then we find $\max_{H} A_{robust}(H)$ such that $A  (H) \ge a$ for some given $a$. Results are as follows (CIFAR10) for $a = 0.81$ (**bold** shows best, *italics* shows second best). Roget (PGD) is the best performer in terms of robust accuracy and Roget (AdvGAN) is the best performer in terms of overall accuracy. Moreover, Roget (AdvGAN) is the second-best performer in terms of robust accuracy.
> >
> > ||robust  (PGD)|total  (PGD)| \||robust (Square)|total (Square)
> > |-|-|-|-|-|-
> > GAT|35.68|81.13| \||62.05|81.56
> > TRADES|53.78|81.01|\||61.13|81.74
> > NUAT|54.64|81.66|\||61.34|82.33
> > PGDAT|35.35|81.64|\||55.84|81.88
> > RFGSMAT|45.94|*81.85*|\||60.25|*83.28*
> > Our (PGD)|**59.42**|81.40|\||**64.73**|82.60
> > Our (AdvGAN)|*55.73*|**82.85**|\||*63.93*|**83.67**
> >
> > FBF and MART are not in the table because we could not find hyperparams that satisfy $A(H)>a=0.81$. We put this result in the main (Tab 6). More similar results are in the appendix (App. H.11)
> >
> > (D) We present results for CIFAR10 for Square and AdvGAN for different default \rho values.
> >
> > ||Robust (Square)|Total (Square)|Robust (AdvGAN)|Total (AdvGAN)
> > |-|-|-|-|-
> > GAT|59.47|76.83|82.04|79.08
> > FBF|31.59|70.59|37.22|71.15
> > TRADES|63.06|78.53|83.82|80.60
> > NuAT|61.78|80.99|86.82|83.49
> > MART|**63.78**|79.55|84.75|81.65
> > PGDAT|62.71|81.30|87.13|83.74
> > RFGSMAT|57.68|**85.96**|89.42|86.20
> > Our (PGD, $\rho$=1.0)|*63.18*|83.06|88.96|85.64
> > Our (AdvGAN, $\rho$=1.0)|62.21|85.29|*91.53*|*88.22*
> > Our (AdvGAN, $\rho$=0.01)|61.01|*85.51*|**91.96**|**88.60**
> >
> > For AdvGAN (Square), our method is best (second best) in terms of both robust and overall accuracy. In the previous draft, we had already put the results for the AdvGAN attack in the Appendix. We brought them in Tab-1.
> >
> >
> > > *if there is some knowledge about the true subset selection strategy, other methods outperforms ROGET.*
> >
> > In Tab 4, we reported numbers for *only* PGD attack. Here only Nu-AT outperforms ours, that too marginally (83.85 vs 83.83). But for other attacks (App H.3, Tab 10), our method outperforms others even when the knowledge of the true subset selection strategy is known. Results are as follows:
> >
> > || AA| Square
> > |-|-|-
> > GAT|78.74|82.45
> > FBF|70.54|70.59
> > TRADES|80.19|82.26
> > NuAT|80.98|84.40
> > MART|78.78| 80.10
> > PGDAT|79.88| 81.30
> > RFGSMAT|81.00| 83.28
> > Our (PGD)|*82.88*| *84.91*
> > Our (AdvGAN)|**83.05**| **85.51**
> >
> > In Tab 4, our key goal was to show that our approach gives *most* benefit when the subset selection strategy is unknown. But in many other cases, as showed in the above, our method outperforms others even when the subset selection strategy is known.
> >
> > > *As amount of test data poisoned grows, the accuracy gains become minimal.*
> >
> > As the test data poisoning grows, our loss function tends to become equal to the objective of usual robust learning problems where all the instances can be attacked. Therefore, the accuracy gain decreases.  If we change hyperparameters for different |S|, our method outperforms almost all baselines (Fig 23 in App H.12)
> >
> >
> > > *time/resource costs by their vs. existing defenses.*
> >
> > We present the results (also in App H.7) as follows:
> >
> > || Time (s)/ epoch|Max GPU Memory (GB)
> > |-|-|-
> > GAT|86.80|2.0
> > FBF| 87.36|2.7
> > TRADES| 578.61|4.0
> > NuAT|74.29|3.2
> > MART|284.52|2.8
> > PGD-AT|290.18|2.7
> > RFGSM-AT|39.07|2.7
> > Our (PGD)|479.05|5.1
> > Our (AdvGAN)|68.89|7.7
> >
> > PGD attack involves its own GD steps for each instance. Thus, it takes more time in our method (PGD). Our method (AdvGAN) generates attacks with a neural net. Here, this network size is much lower than no. of instances, and GD is performed on  NN params but not each instance. Thus,  our method (AdvGAN) is very fast. Though ours consume high memory, it fits well in a 12 GB GPU.
> >
> > > *is categorical-cross entropy a Polyak-Lojasiewicz Loss function?*
> >
> >  For a linear predictor: $P(y = k) =  \exp(w_k ^T x) /\sum_j \exp(w_j ^T x) $, CE loss is convex and hence, a PL function. Now, models like Resnet are heavily non-convex. Even for usual image classification without adversarial attack, any theoretical guarantee on convergence or generalization for such a model is extremely difficult. However, the approximation guarantee of the SDG method (Eq 7, Thm 3) still applies to Resnet-type architectures and thus the subset selection method still admits an approximation guarantee for each iteration.
> >
> > That said, we clearly acknowledged the limitation of our approximation guarantee (second para of Page 6), where we discuss the difficulty of the approximation guarantee.

---

### Decision · Program_Chairs · 2023-01-20

**Decision:**

Reject

**Justification For Why Not Higher Score:**

Please see above.

**Justification For Why Not Lower Score:**

NA

**Metareview: Summary, Strengths And Weaknesses:**

The authors introduce an interesting new setting in the context of adversarial robustness, in which the attacker is only allowed to perturb a subset of the data samples, but not all. This weakening of the attacker leaves the possibility of maintaining better overall accuracy but a challenge remains in that the subset to be selected by the attacker is a priori unknown. The authors cast this training problem as a min-max game involving worst-case subset selection along with optimization of model parameters, rendering the problem NP-hard. To tackle this, they first show that, for a given learner's model, the objective can be expressed as a difference between a weakly submodular and a modular function. They then use this property to propose an iterative approximation algorithm. Experiments show that the proposed method obtains better overall accuracy compared to several state-of-the-art defense methods for different adversarial subset selection techniques.

This is an interesting, novel setting; it is also very interesting to see that the inherent optimization can be cast into a combinatorial framework that comes with guarantees. That said, the reviewers found many aspects of the paper lacking, and have provided extensive feedback to the authors on how to improve their paper. I will list a few here:

1) The NP-hardness proof is wrong; the reduction should be inverted.
2) Baselines are indeed not optimized for the overall accuracy. However, baselines could be slightly modified targeting for the overall accuracy to make a fair comparison.
3) Theoretical assumptions are strong and could be weakened
4) Given attack examples are only 10% of the test set, ROGET seems to improve accuracy on the 90% clean dataset and appear better on overall accuracy (as we scale robust accuracy down in this calculation). This does not really showcase ROGET as a good defense; rather it capitalizes on the 90% clean examples to improve the overall accuracy.
5) ROGET may not provide a stable top-performance solution. When , if just looking at overall accuracy, as authors suggested, RFGSM-AT consistently outperforms ROGET(PGD) on CIFAR10, while TRADES outperforms ROGET(AdvGAN) on FMNIST over overall accuracy that considering white-box attack. Moreover, tuning rho is tricky and leads to more complicated cases. The best rho on different datasets changes in scale, e.g.,  on CIFAR10 and  on FMNIST, and the performance gap between different  settings, e.g.,  and  on FMNIST, making it less effective to provide a stable top-performance solution.
6)  SOTA architectures and attacks should be used (e.g., TRADES-trained WideResNet as architecture and AA as white-box attack).

**Summary Of Ac-Reviewer Meeting:**

Reviewers expressed many concerns, including ones highlighted above. They also added these online.